# Aridity-driven shift in biodiversity–soil multifunctionality relationships

Weigang Hu [1,6], Jinzhi Ran[1,6], Longwei Dong[1,6], Qiajun Du[1], Mingfei Ji[1], Shuran Yao[1], Yuan Sun[1], Chunmei Gong[2], Qingqing Hou[1], Haiyang Gong[1], Renfei Chen[1], Jingli Lu[1], Shubin Xie[1], Zhiqiang Wang[1], Heng Huang [1], Xiaowei Li[1], Junlan Xiong[1], Rui Xia[1], Maohong Wei[1], Dongmin Zhao[1], Yahui Zhang[1], Jinhui Li[1], Huixia Yang[1], Xiaoting Wang[1], Yan Deng[1], Ying Sun[1], Hailing Li[1], Liang Zhang[1], Qipeng Chu[1], Xinwei Li[1], Muhammad Aqeel[1], Abdul Manan[1], Muhammad Adnan Akram[1], Xianghan Liu[1], Rui Li[1], Fan Li[1], Chen Hou[3], Jianquan Liu [1], Jin-Sheng He [1], Lizhe An[1], Richard D. Bardgett [4✉], Bernhard Schmid [5✉] & Jianming Deng [1✉]

Relationships between biodiversity and multiple ecosystem functions (that is, ecosystem multifunctionality) are context-dependent. Both plant and soil microbial diversity have been reported to regulate ecosystem multifunctionality, but how their relative importance varies along environmental gradients remains poorly understood. Here, we relate plant and microbial diversity to soil multifunctionality across 130 dryland sites along a 4,000 km aridity gradient in northern China. Our results show a strong positive association between plant species richness and soil multifunctionality in less arid regions, whereas microbial diversity, in particular of fungi, is positively associated with multifunctionality in more arid regions. This shift in the relationships between plant or microbial diversity and soil multifunctionality occur at an aridity level of ~0.8, the boundary between semiarid and arid climates, which is predicted to advance geographically ~28% by the end of the current century. Our study highlights that biodiversity loss of plants and soil microorganisms may have especially strong consequences under low and high aridity conditions, respectively, which calls for climate-specific biodiversity conservation strategies to mitigate the effects of aridification.

[1] State Key Laboratory of Grassland Agro-Ecosystem, School of Life Sciences, Lanzhou University, Lanzhou, China. [2] College of Horticulture, Northwest A&F University, Yangling, China. [3] Department of Biological Science, Missouri University of Science and Technology, Rolla, MO, USA. [4] Department of Earth and Environmental Sciences, The University of Manchester, Manchester, UK. [5] Department of Geography, Remote Sensing Laboratories, University of Zurich, Zurich, Switzerland. [6] These authors contributed equally: Weigang Hu, Jinzhi Ran, Longwei Dong. ✉email: richard.bardgett@manchester.ac.uk; bernhard.schmid@ieu.uzh.ch; dengjm@lzu.edu.cn

Ecosystem functioning (fluxes of matter and energy between trophic levels, and nutrient stocks and transformation rates) are affected by the collective activities of producer, consumer, and decomposer communities[1–4]. Plants and soil microorganisms are two major drivers of ecosystem functioning, being the main producers and decomposers of terrestrial ecosystems, respectively[5–7]. Reflecting this, a number of studies have explored how both the plant and soil microbial diversity independently relate to particular ecosystem functions[8,9]. However, ecosystem functioning is inherently multifunctional[10], and awareness of this has led to a surge of studies exploring relationships between biodiversity and ecosystem multifunctionality, or the ability of an ecosystem to deliver multiple functions or services simultaneously[3,11–15].

So far, most research targeting biodiversity–multifunctionality relationships has explored the role of plant or soil microbial diversity at a single trophic level. However, there is increasing awareness of the influence and inter-dependence of different trophic groups on each other, and of the fundamental role played by feedbacks across trophic levels in regulating ecosystem multifunctionality[16–20]. Recent studies have shown that the effects of plant and soil microbial diversity on ecosystem multifunctionality can be synergistic and complementary[14,16,19,21], and that combining plant and soil microbial diversity can increase the predictive power of biodiversity–multifunctionality relationships[22,23]. For instance, Delgado-Baquerizo et al.[14] reported that soil microbial diversity positively related to ecosystem multifunctionality in global drylands, which indirectly mediated the positive association between plant species richness and multifunctionality. Similar cascading relationships between plant diversity, soil microbial diversity, and ecosystem multifunctionality were found across globally distributed biomes[21]. Furthermore, variation in plant and soil microbial diversity in combination accounted for more (22%) of the variation in ecosystem multifunctionality than did either component alone in alpine grasslands from the Qinghai-Tibetan Plateau[22]. Together, these findings suggest that plant and soil microbial diversity in combination may be powerful drivers of ecosystem multifunctionality. Moreover, given that biodiversity loss often occurs across trophic levels[18], a combined assessment of plant and soil microbial diversity is needed to better understand the potential consequences of species loss for ecosystem multifunctionality.

It is likely, however, that relationships between biodiversity and ecosystem multifunctionality also depend on the environmental context and therefore may change along environmental gradients[12,16,24,25]. For instance, plant species richness has been shown to enhance ecosystem multifunctionality in small-scale plant diversity-manipulation experiments[11,12,16,26] and large-scale studies of dryland and grassland ecosystems across different environmental conditions[13,22]. Furthermore, a meta-analysis of 94 experimental manipulations of plant species richness across aquatic and terrestrial habitats revealed that plant diversity sometimes has negative effects on ecosystem multifunctionality[27]. Likewise, positive effects of soil bacterial and fungal diversity on ecosystem multifunctionality have been reported in grassland, forest, and dryland ecosystems[14,15,22,28], negative effects of soil bacterial and saprophytic fungal richness have been reported in semiarid grassland and subtropical forest, respectively[17,23], whereas neutral effects of soil archaeal and bacterial richness have been reported in grassland and boreal forest[22,28]. However, empirical data are still lacking for the linkages among both the plant or soil microbial diversity and ecosystem multifunctionality along extended environmental gradients at large spatial scales. Importantly, although Jing et al.[22] reported that regional-scale change in climate could mediate the relationships between plant or soil microbial diversity and ecosystem multifunctionality, the extent to which these relationships vary, and whether their relative strength shifts along environmental gradients, remains largely untested. This limits our predictive understanding of the potential ecological consequences of biodiversity change of both the plants and soil microorganisms under different environmental conditions. This knowledge may be of particular importance if areas of conservation priority are to be identified and attempts to alleviate the effects of environmental change are made.

Given the context-dependency of biodiversity–multifunctionality relationships and that plants and soil microorganisms may have different roles in maintaining ecosystem multifunctionality[5–7,29], we hypothesize that the relationships between plant or soil microbial diversity and ecosystem multifunctionality may shift along an aridity gradient due to changes in the net effects of interactions across trophic levels. We predict that soil microbial diversity shows a stronger and positive association with ecosystem multifunctionality in more arid environments, whereas plant diversity exhibits a stronger and positive correlation with multifunctionality in less arid environments (Fig. 1). Specifically, we expect top-down effects of soil microbial decomposer diversity on ecosystem multifunctionality (via increasing organic matter decomposition and nutrient transformation) to be of more importance under more arid conditions[30–32]. Here plants are scarce and primary productivity and consequent resource inputs to soils are limited[33], which increases belowground competition for limiting resources such as water and nutrients[34–37], and thus enhances dependency on soil microbial decomposers for ecosystem functioning[14,21] (Fig. 1). In contrast, we expect bottom-up effects driven by the diversity of plant producers (via controlling resource inputs) to be of more importance under less arid conditions[33,38]. Here primary productivity is less restricted by water shortage and thus plant diversity–productivity[39–41] and –multifunctionality relationships are expected to develop to a greater extent (Fig. 1).

To test our hypothesis, we evaluate how relationships between plant or soil microbial diversity and soil multifunctionality vary along a broad aridity gradient across northern China. We focus on soil multifunctionality because of the importance of multiple soil functions in regulating global biogeochemical cycles[6,42]. To achieve this, we measure both the plant diversity (i.e., species richness) and soil microbial diversity [i.e., soil archaeal, bacterial, and fungal operational taxonomic unit (OTU) richness] together with seven soil variables [i.e., DNA concentration, soil organic carbon (C), total nitrogen (N), ammonium, nitrate, total phosphorus (P), and available P] related to nutrient pools that are representative of stocks of matter and energy at 130 dryland sites covering >3,500,000 km$^2$ (Fig. 2a). We measure stocks rather than process rates to assess ecosystem multifunctionality because stocks are better indicators of longer-term ecosystem functioning under natural conditions[2–4]. Together, these soil variables (hereafter functions) constitute a good proxy of nutrient cycling, climate regulation, and soil fertility[4,43] (see also Methods for further rationale of how these variables relate to soil or ecosystem functioning). The field sites are selected along a 4,000 km natural aridity [calculated as 1 – aridity index (AI, precipitation/potential evapotranspiration)] gradient and cover a large range of vegetation types (i.e., typical grassland, desert grassland, alpine grassland, and desert[44]), edaphic characteristics, and climatic conditions (i.e., dry-subhumid, semiarid, arid, and hyperarid regions) typical of drylands across northern China (Fig. 2a and Supplementary Table 1).

In addition to the field study, we manipulate soil water availability in a microcosm experiment to experimentally test for linkages between moisture content, microbial diversity, and soil multifunctionality by simulating differences in moisture conditions among our field sites (Supplementary Fig. 1; see also Methods for more details of experimental design). The purpose of

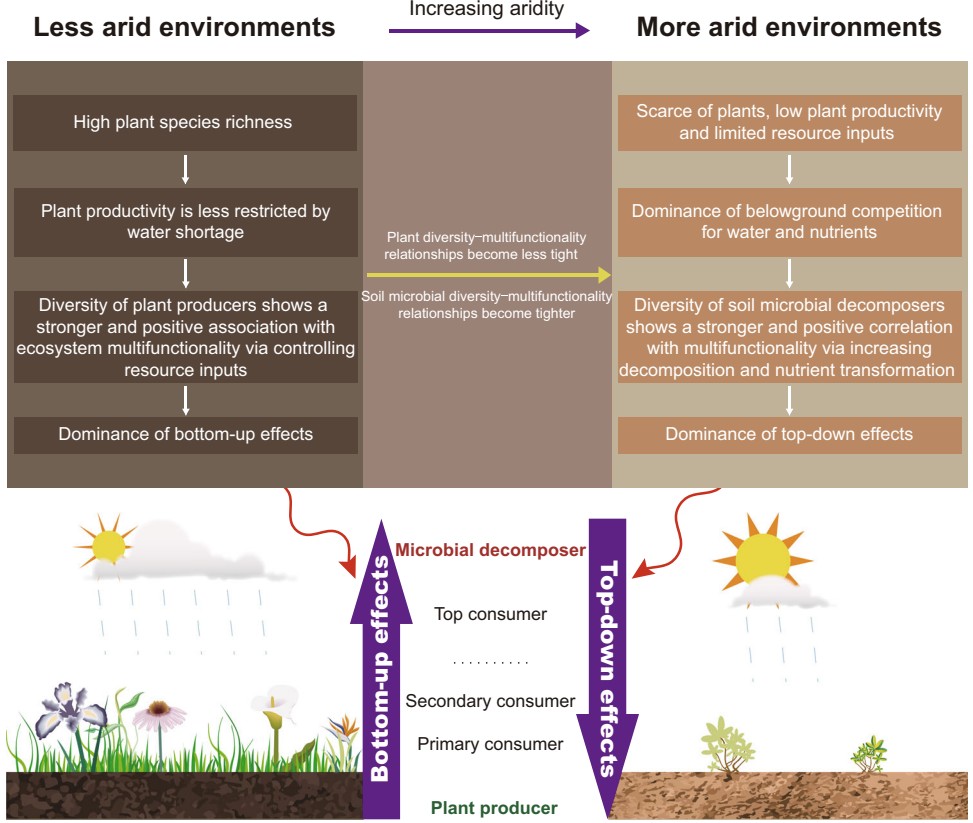

**Fig. 1 Aridity mediates the biodiversity-multifunctionality relationships.** Schematic diagram illustrating the hypothetical shift in relationships between plant or soil microbial diversity and ecosystem multifunctionality with increasing aridity.

the microcosm experiment is to complement the field study and thus experimentally underpin the potential changes in the relationship between soil microbial diversity and multifunctionality with increasing aridity in the absence of plants. Here we do not have the resources to take into account plant diversity and we assess soil multifunctionality by measuring process rates because the time span of the experiment is too short to detect changes in soil nutrient stocks. Our results demonstrate that the relationships of plant or microbial diversity with soil multifunctionality shift at an aridity level of ~0.8 under field conditions, with greater importance of microbial diversity, especially of fungi, observed in high-aridity sites. Our findings suggest that biodiversity conservation strategies should be adapted to the aridity conditions to alleviate negative impacts of aridification on soil multifunctionality. For instance, promoting higher plant or microbial diversity may represent an effective strategy to maintain soil functioning and service provisioning under low or high aridity conditions, respectively.

## Results

**Field study in drylands across northern China.** We first evaluated the responses of each of the individual soil functions and multifunctionality to aridity and identified the aridity levels at which these responses showed abrupt changes. All soil functions and multifunctionality responded in a nonlinear manner to increasing aridity (Fig. 2b–i and Supplementary Table 2). Among these, five individual soil functions and multifunctionality decreased nonlinearly with increasing aridity (Fig. 2b–e, h, i), whereas the two remaining soil functions, namely nitrate and total P, were less affected (Fig. 2f, g). A sharp decline in soil multifunctionality was detected at an aridity level of 0.59; this aridity value was lower than or equal to those found for any

individual soil functions (ranging from 0.59 to 0.96; Fig. 2b–i and Supplementary Figs. 2, 3), indicating that soil multifunctionality was more susceptible to increasing aridity. Similarly, strong and negative effects of aridity on soil multifunctionality were observed when using the multiple-threshold approach (Supplementary Fig. 4 and Supplementary Table 3).

We then fitted a linear mixed-effects model to evaluate the relationships between multiple biotic and abiotic factors and soil multifunctionality (Table 1). We found that plant species richness, belowground net primary productivity (BNPP), and soil clay content were positively, whereas aridity and soil pH were negatively associated with soil multifunctionality across the aridity gradient. The soil microbial diversity index (average of all components of soil microbial diversity metrics, i.e., archaeal, bacterial, and fungal richness; see also Methods) alone showed a weakly positive association with soil multifunctionality, but its interaction with aridity was significantly and positively correlated with soil multifunctionality. Conversely, the interaction term between plant species richness and aridity showed a negative association with soil multifunctionality. These results remained qualitatively consistent when fitting a simplified mixed-effects model to focus only on the links between aridity, biodiversity, and soil multifunctionality (Supplementary Fig. 5). We thus used the simplest model in further analyses for simplicity.

We next performed a moving-window analysis to evaluate the potential shifts in the relationships between plant or microbial diversity and soil multifunctionality with increasing aridity (Fig. 3). The positive relationship between plant species richness and soil multifunctionality weakened sharply at an aridity value of 0.78, and was neutral beyond that value (Fig. 3a,c and Supplementary Fig. 6a). Also, the interaction term between the aridity and plant species richness became less negative along the

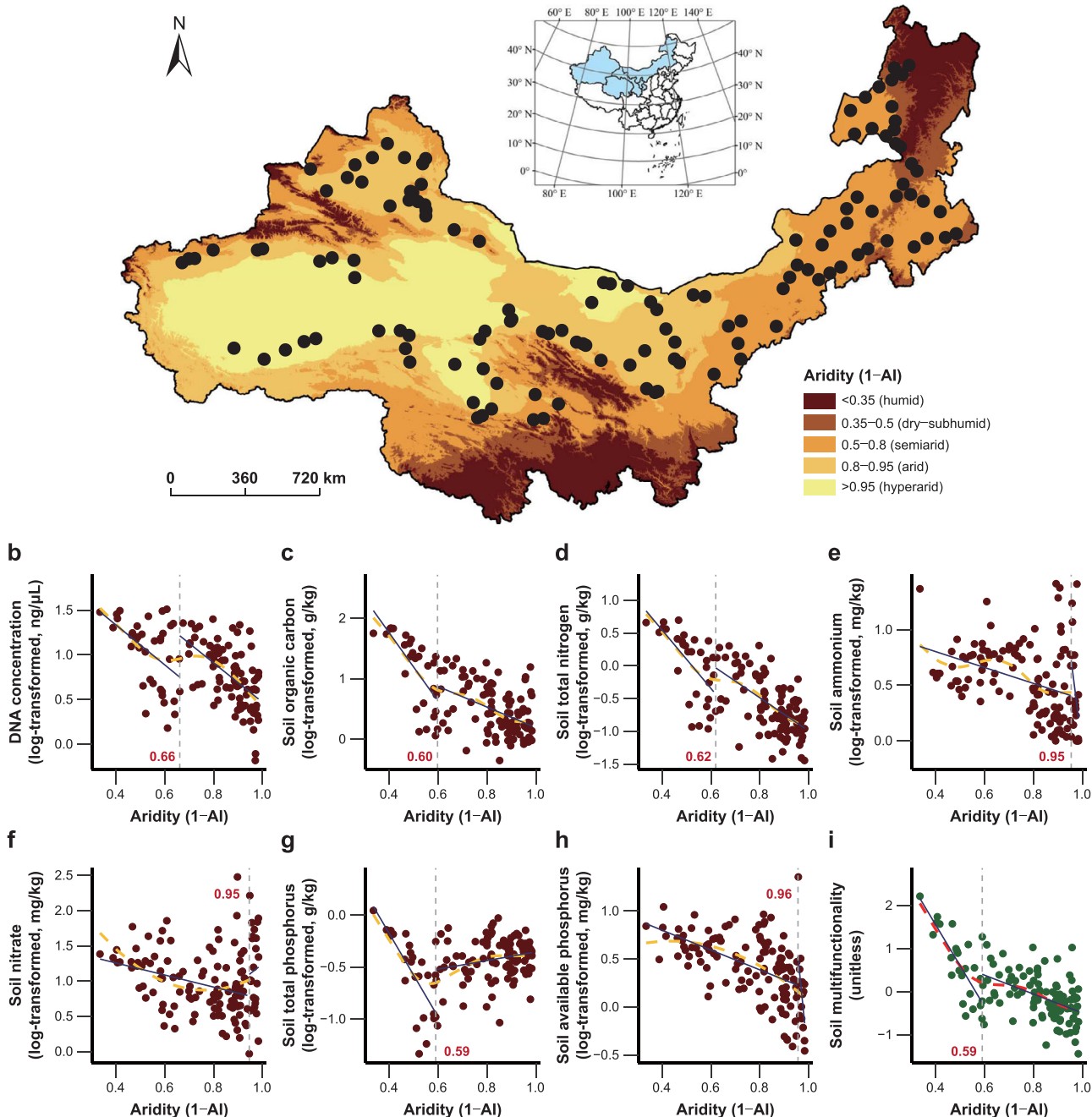

**Fig. 2 Geographical distribution of field sites and nonlinear responses of individual soil functions and multifunctionality to increasing aridity. a** Map of the 130 field sites in drylands across northern China. The sites represent the four different dryland subtypes (i.e., dry-subhumid, semiarid, arid, and hyperarid regions), which are defined with different aridity levels, as demonstrated in the legend. Inset map shows China with the study area colored in light blue. **b–i** Nonlinear responses of DNA concentration (**b**), soil organic carbon (**c**), total nitrogen (**d**), ammonium (**e**), nitrate (**f**), total phosphorus (**g**), available phosphorus (**h**), and soil multifunctionality (**i**) to aridity, and their respective aridity thresholds. The orange and red dashed lines indicate the nonlinear trends fitted by quadratic or generalized additive models (GAMs). The gray dashed lines and inset numbers in red represent the aridity thresholds identified. The blue solid lines denote the linear fits at both sides of each aridity threshold.

aridity gradient (Fig. 3b). These results suggest that plant species richness showed a stronger positive association with soil multi-functionality in less arid environments, which was further supported by the significant bootstrapped coefficients of both the plant species richness and its interaction with aridity under low aridity levels (Fig. 3e). At an aridity value of 0.81, however, an abrupt shift in the slope of the relationship between the soil

microbial diversity index and soil multifunctionality from negative to positive was observed, and the relationship finally changed to positive at the end of the aridity gradient (Fig. 3a, d and Supplementary Fig. 6b). The interaction term between the aridity and the soil microbial diversity index shifted across the aridity gradient from negative to positive (Fig. 3b), and it was statistically significant under high aridity conditions (Fig. 3e).

**Table 1 Linear mixed-effects model for the relationships between multiple biotic (BNPP, plant species richness, and the soil microbial diversity index) and abiotic (aridity, soil pH, and clay content) factors and soil multifunctionality with considering soil and vegetation types as random terms.**

| Term | df | ddf | MS | F | P | Estimate | VIF |
|---|---|---|---|---|---|---|---|
| Random terms are soil and vegetation types; Conditional $R^2$ 0.69; Marginal $R^2$ 0.59 | | | | | | | |
| Year | 1 | 37.5 | 3.30 | 9.22 | 0.004 | −0.11 | 2.09 |
| Plant species richness | 1 | 8.9 | 10.03 | 28.01 | <0.001 | 0.03 | 2.12 |
| Soil microbial diversity index | 1 | 103.1 | 0 | 0 | 0.996 | 0.21 | 2.45 |
| Aridity | 1 | 18.7 | 7.45 | 20.83 | 0.001 | −0.36 | 2.04 |
| BNPP | 1 | 49.1 | 1.88 | 5.25 | 0.026 | 0.05 | 1.10 |
| Soil pH | 1 | 112.6 | 3.50 | 9.77 | 0.002 | −0.26 | 1.65 |
| Soil clay content | 1 | 115.6 | 5.39 | 15.07 | <0.001 | 0.18 | 2.10 |
| Elevation | 1 | 109.5 | 1.24 | 3.45 | 0.066 | 0.23 | 2.27 |
| Latitude | 1 | 114.1 | 4.79 | 13.38 | <0.001 | 0.35 | 4.14 |
| Longitude | 1 | 114.6 | 0.99 | 2.77 | 0.098 | −0.19 | 1.72 |
| Plant species richness × Soil microbial diversity index | 1 | 109.1 | 0 | 0 | 0.994 | 0.20 | 4.47 |
| Aridity × Plant species richness | 1 | 104.2 | 1.36 | 3.79 | 0.054 | −0.13 | 1.76 |
| Aridity × Soil microbial diversity index | 1 | 115.2 | 2.03 | 5.67 | 0.019 | 0.26 | 4.10 |

*df* numerator degrees of freedom, *ddf* denominator degrees of freedom, *MS* mean squares, *F* variance ratio, *P* probability of type-I error (two-sided), *VIF* variance inflation factor.
Fixed terms are fitted sequentially (type-I sum of squares) as indicated in Eq. 2, and × denotes an interaction term. Marginal (variance explained by fixed terms) and conditional (variance explained by fixed and random terms) $R^2$ values are given. The term "Year" is first introduced into the model to eliminate the variation due to different sampling years. Latitude, longitude, and elevation of the field sites are included to account for the spatial structure of our dataset.

These results indicate that, in contrast to plant species richness, the soil microbial diversity index exhibited a stronger association with soil multifunctionality in more arid environments.

Given the clear shift in biodiversity–soil multifunctionality relationships detected at an aridity level of around 0.8 (Fig. 3), we further divided the study sites into two groups, namely sites with aridity <0.8 and >0.8, representing less and more arid regions respectively, to examine whether there was a significant linear relationship between each component of biodiversity and soil multifunctionality in less and more arid regions. Ordinary least-squares (OLS) regressions showed that soil multifunctionality was significantly and positively correlated with plant species richness, but significantly and negatively associated with soil archaeal and bacterial richness, and with the soil microbial diversity index in less arid regions (Fig. 4a–d). In contrast, soil multifunctionality exhibited a significant and positive association with richness of soil fungi and fungal saprotrophs, and with the soil microbial diversity index in more arid regions (Fig. 4b, e, f). These results were robust when accounting simultaneously for multiple biotic and abiotic factors by fitting linear mixed-effects models (Supplementary Table 4). Across the aridity gradient, however, soil multifunctionality was positively related to plant species richness, soil fungal richness, and richness of fungal saprotrophs, pathogens, and symbionts respectively, negatively related to soil archaeal richness, and not related to soil bacterial richness and the soil microbial diversity index (Fig. 4). Similar results were obtained when single soil functions (Supplementary Figs. 7–14) and alternative measurements of soil multifunctionality were analyzed (Supplementary Figs. 15, 16 and Supplementary Table 3).

As a complement to the analyses of bivariate correlations (Fig. 4 and Supplementary Table 4), we used structural equation models (SEMs) to infer the hypothesized direct and indirect relationships between aridity, soil pH, soil clay content, BNPP, plant and microbial diversity and soil multifunctionality (see an a priori model in Supplementary Fig. 17a), and to test whether different indirect pathways may drive the aridity–biodiversity–multifunctionality relationships in less and more arid regions (Fig. 5 and Supplementary Table 5). We hypothesized that aridity could influence soil multifunctionality directly and indirectly via affecting plant and microbial diversity. Previous experimental studies manipulating biodiversity have provided evidence that

both plant[11,12,16,26] and microbial diversity[9,15,45–47] have strong effects on multiple soil functions related to nutrient stocks, such as those evaluated here. Therefore, although our observational study is correlative in nature, and hence any causal hypotheses should be made with caution, we find it reasonable to assume that both the plant and microbial diversity can be drivers of soil multifunctionality. At the same time, we acknowledge that both the plant and microbial diversity may in turn respond to soil nutrients or other unmeasured factors, such as vegetation spatial patterns and plant–soil feedbacks[33,48–51]. Despite this, SEMs for less vs. more arid regions revealed that plant species richness and the soil microbial diversity index were directly and positively associated with soil multifunctionality in less and more arid regions, respectively. More importantly, although aridity did not affect soil multifunctionality directly in either of the two climatic regions, it did affect it indirectly and negatively via reduced plant species richness in less arid regions and via reduced the soil microbial diversity index in more arid regions (Fig. 5 and Supplementary Table 5).

To address the potential redundancy between total soil N and other individual soil functions (Supplementary Fig. 18a), and the fact that total soil P is more closely related to abiotic rather than biotic processes, we removed these two soil functions and then repeated the above analyses. Consistent results were found for the simplified version of soil multifunctionality including five soil functions (i.e., simplified soil multifunctionality) (Supplementary Figs. 19–28 and Supplementary Tables 2, 3, 6–8).

Finally, we provide a quantitative estimate of the future changes in areas with aridity crossing 0.8 in drylands across northern China according to current climatic forecasts by the Intergovernmental Panel on Climate Change's (IPCC's) Representative Concentration Pathways (RCP) 4.5 and 8.5 scenarios (Fig. 6). Our estimates showed that by 2100 these areas may expand by 11.5 and 28.3%, respectively, under RCP 4.5 and 8.5. Shrinking areas with aridity crossing 0.8 are located mainly on the southern slope of the Altai Mountains in northern Xinjiang, while expanding areas are distributed mostly in central and eastern Inner Mongolia.

**Microcosm experiment manipulating soil water availability.** To complement the field study and further confirm the potential

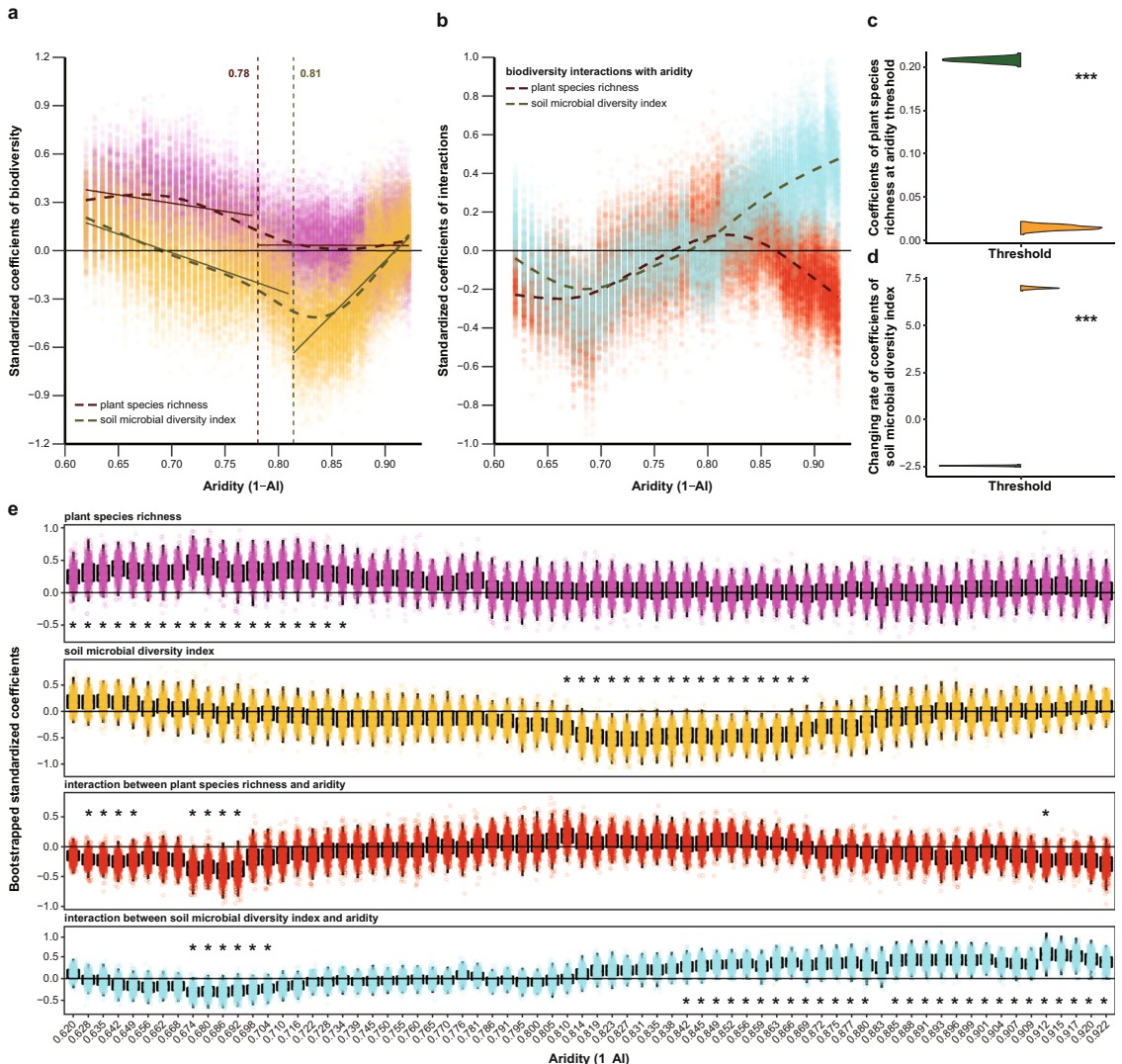

**Fig. 3 Nonlinear changes of relationships between biodiversity and its interactions with aridity and soil multifunctionality along aridity gradients. a, b**
Nonlinear changes of standardized coefficients of biodiversity (**a**) and the interactions between biodiversity and aridity (**b**) obtained from a linear mixed-effects model (Eq. 3) throughout a moving subset window of the field sites surveyed along aridity gradients. The dots indicate the bootstrapped coefficients of the fixed terms shown for each subset window. The dashed lines denote the nonlinear trend fitted by GAMs. In (**a**), the vertical dashed lines and inset numbers represent the aridity thresholds identified, and the solid lines represent the linear fits at both sides of each aridity threshold. **c, d** Violin diagrams show bootstrapped predicted values at the threshold (**c**) and bootstrapped slopes (**d**) of the two regressions existing at each side of the aridity threshold found for plant species richness and the soil microbial diversity index in (**a**), respectively (dark green for the regression before the threshold and orange for the regression after the threshold). Significant differences between before and after the threshold are determined using an unpaired two-sided Mann–Whitney $U$-test. Significance level is: ***$P < 0.001$. **e** Boxplots demonstrate the distribution of bootstrapped standardized coefficients corresponding to those in (**a, b**) for each subset window ($N = 500$ independent simulations). Boxplots show the median (center line), 25th and 75th percentiles of each distribution. Whiskers represent the minimum and maximum values that remain inferior 1.5 times the interquartile range below or above the distribution median. Asterisks indicate significant values of coefficients at 95% confidence intervals (one-sided $P \leq 0.05$).

changes in the soil microbial diversity–multifunctionality relationship with increasing aridity in the absence of plants, we evaluated the linkages between moisture content, microbial diversity, and soil multifunctionality by experimentally manipulating water availability in soil microcosms. Consistent with the findings of our field study (Supplementary Fig. 29), our microcosm experiment revealed a negative quadratic relationship between the soil moisture content and bacterial and fungal richness and the soil microbial diversity index (Supplementary Fig. 30). Furthermore, as in the field, our experimental study confirmed the role of declining soil water availability as a significant driver of reduced soil multifunctionality, except at the highest moisture level of 120% field capacity where soil microbial

activity may be suppressed by excess moisture (Supplementary Fig. 31a). Most importantly, we detected a stronger positive relationship between the soil microbial diversity (especially for bacteria and fungi) and multifunctionality below the moisture level of 20% field capacity (equivalent to moisture content of ~6.09%) (Supplementary Fig. 31b–d), which was similar to the soil moisture level of the lower boundary of arid regions (i.e., 0.8 aridity level) in our field study (Fig. 2a and Supplementary Fig. 1).

## Discussion
Our field study showed that increasing aridity reduced most individual soil functions related to nutrient cycling, climate

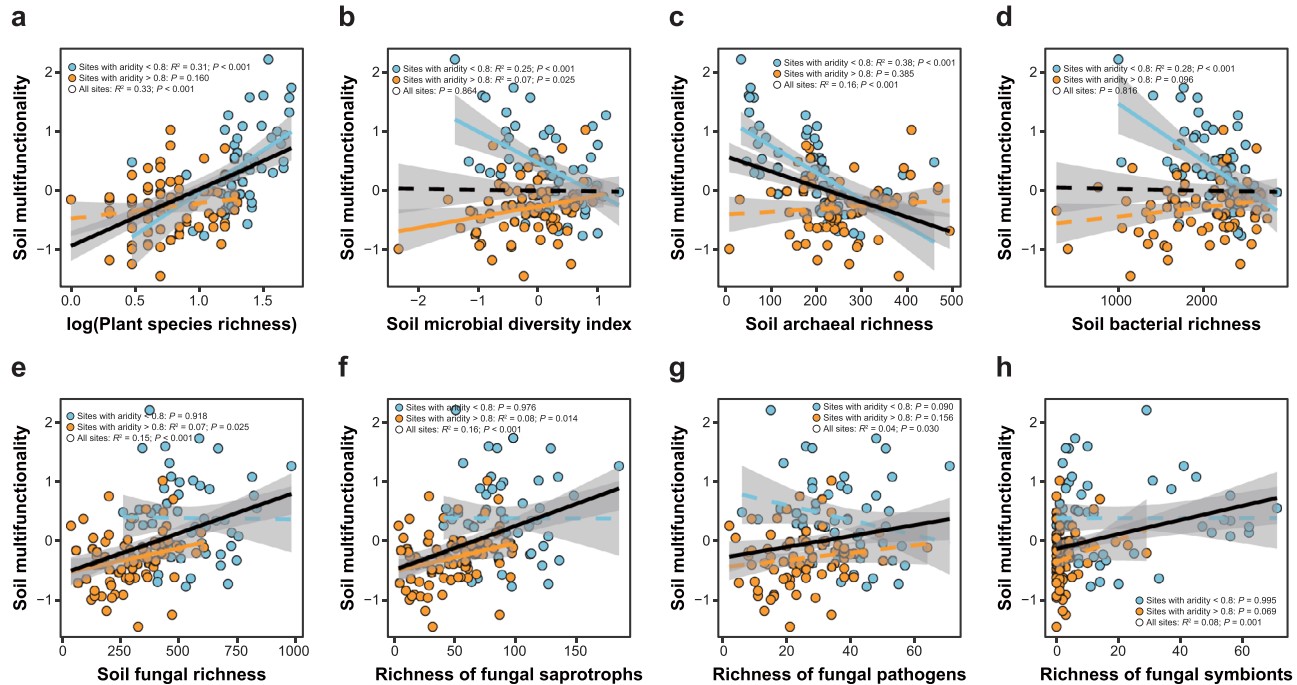

**Fig. 4 Relationships between plant or soil microbial diversity and soil multifunctionality. a–h** Relationships between log-transformed plant species richness (**a**), soil microbial diversity index (**b**), soil archaeal richness (**c**), soil bacterial richness (**d**), soil fungal richness (**e**), richness of fungal saprotrophs (**f**), richness of fungal pathogens (**g**), and richness of fungal symbionts (**h**) and soil multifunctionality at sites with aridity <0.8 ($N = 54$) and >0.8 ($N = 76$), as well as across all field sites ($N = 130$; the black lines). Lines represent the fitted linear OLS model. Solid and dashed lines denote statistically significant (two-sided $P \leq 0.05$) and nonsignificant (two-sided $P > 0.05$) relationships, respectively. Shaded areas denote the 95% confidence interval of the regression lines.

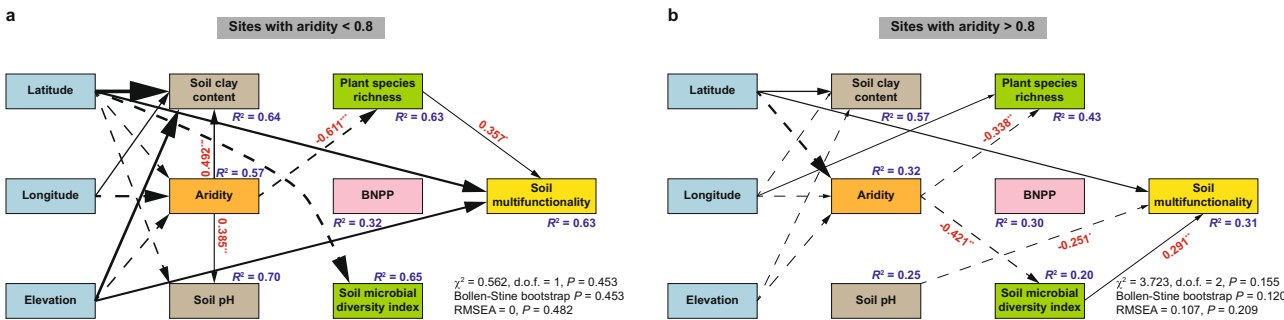

**Fig. 5 Structural equation models (SEMs) accounting for the hypothesized direct and indirect relationships between aridity, soil properties (pH and clay content), biodiversity (plant species richness and the soil microbial diversity index), BNPP, and soil multifunctionality. a, b** SEMs are shown for sites with aridity <0.8 (**a**; $N = 54$) and >0.8 (**b**; $N = 76$). Note that we only present significant relationships (two-sided $P < 0.05$) and their coefficients (numbers adjacent to arrows) for graphical simplicity. Latitude, longitude, and elevation of the field sites are included to account for the spatial structure of our dataset, and thus their coefficients are not included. A priori model including all hypothesized causal relationships is available in Supplementary Fig. 17a, and all the rest of coefficients and their significance levels are available in Supplementary Table 5. For the SEM of sites with aridity >0.8, we remove the relationship between soil pH and BNPP with a coefficient close to zero to improve its overall goodness-of-fit. Continuous and dashed arrows indicate positive and negative relationships, respectively. The thickness of the arrow is proportional to the magnitude of standardized path coefficients and indicative of the strength of the relationship. Asterisks indicate the significance level of each coefficient: *$P < 0.05$; **$P < 0.01$; ***$P < 0.001$. $R^2$ is the proportion of variance explained by the model. Goodness-of-fit statistics for each SEM are given (d.o.f. degrees of freedom, RMSEA root mean squared error of approximation).

regulation, and soil fertility, as well as soil multifunctionality. Compared with individual soil functions, soil multifunctionality was more sensitive to increasing aridity. This might be due to the correlative nature of individual soil functions, and reflect the similar underlying processes that responded negatively to increasing aridity. More importantly, aridity shifted the relationships between plant or microbial diversity and soil multifunctionality. Plant species richness was consistently and positively related to soil multifunctionality across the aridity gradient, and, as hypothesized, showed a stronger and more positive association with multifunctionality in less arid regions, whereas soil microbial diversity, in particular that of soil fungi, exhibited a stronger and positive association with multifunctionality in more arid regions. Our microcosm experiment, which complemented the field study by experimentally manipulating soil water availability, confirmed that declining soil moisture content was a major driver of reduced soil multifunctionality, and that, in the absence of plants, the relationship

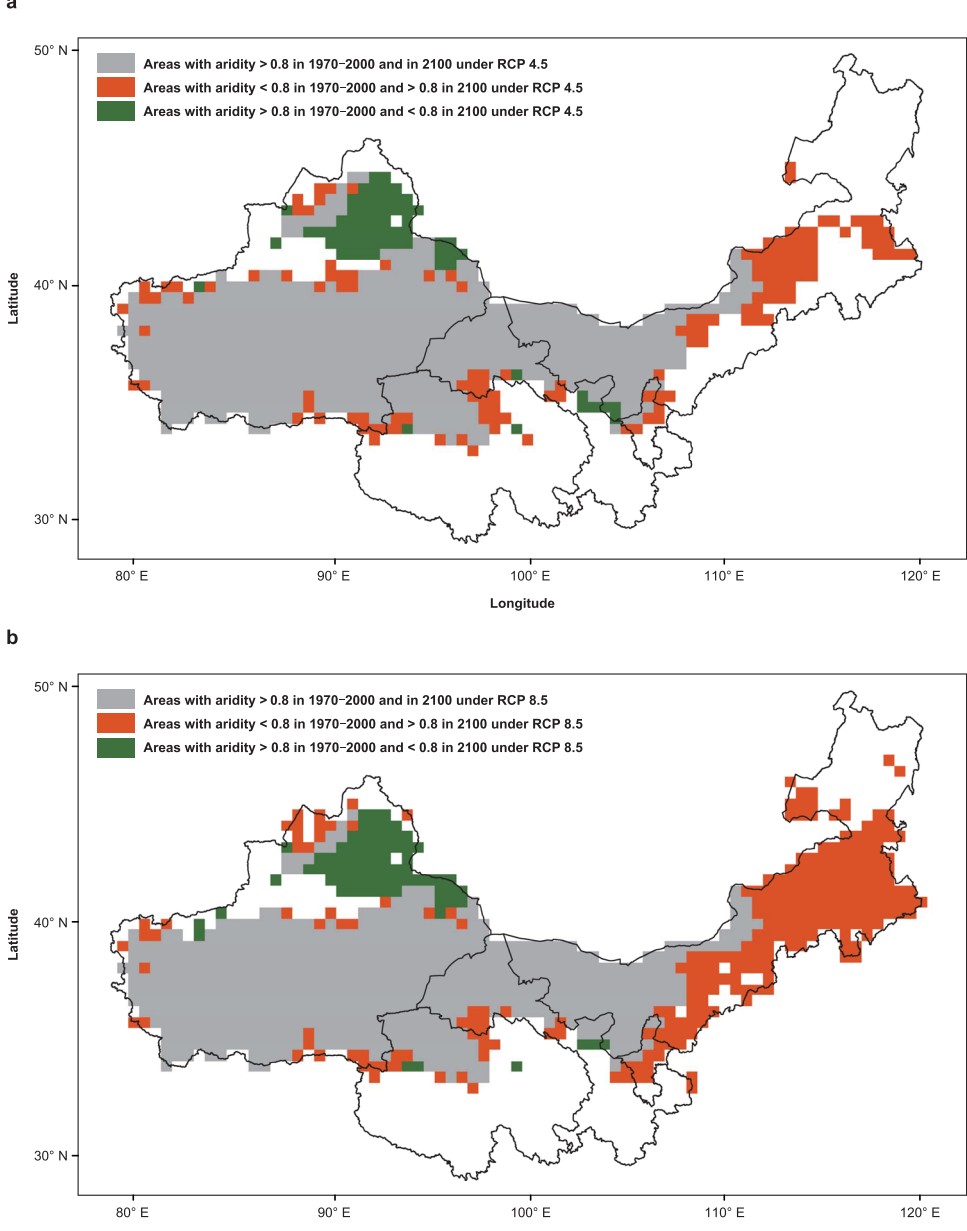

**Fig. 6 Predicted future changes in areas crossing 0.8 aridity level in drylands across northern China. a, b** Predictions of future changes in areas that will cross 0.8 aridity level are shown for between 1970–2000 and 2100 by the IPCC's RCP 4.5 (i.e., assuming saturated increase in $CO_2$ emissions; (**a**) and 8.5 (i.e., assuming sustained increase in $CO_2$ emissions; (**b**) scenarios in drylands across northern China, respectively. The blank areas are outside of the range considered for this study (i.e., areas that are dry-subhumid regions, semiarid regions, and non-drylands today).

between the soil microbial diversity and multifunctionality became stronger and positive at low levels of soil moisture. Furthermore, we found that the shift in the relationships between plant or microbial diversity and soil multifunctionality occurred at an aridity level of ~0.8 in the field, corresponding to a soil moisture content of ~6.09% in the microcosm experiment in the absence of plants, below which soil microbial diversity was more strongly related to multifunctionality.

The consistent and positive relationship between the plant species richness and soil multifunctionality across the aridity gradient aligned with previous reports[11–13,22,26] and can be explained by increased litter inputs into the soil due to increased net primary production or complementary resource use among species[52,53]. As predicted, the positive relationship between the plant diversity and soil multifunctionality became weaker from

less toward more arid regions. This can be, at least partially, attributed to a reduction in available biotope space for plant species with increasing aridity[54,55], decreasing the potential for niche complementarity among species, and thus declining resource inputs to soil[13,38,53]. Aridity thus acts as an environmental filter, selecting plant species with similar niches[56]. Therefore, plant diversity in less arid regions may promote soil functioning via complementary resource use and facilitation between the plant species[24,57], whereas competitive interactions could have contributed to the weakened relationship of plant diversity with soil multifunctionality in more arid regions[16,24]. Alternatively, the dominant species in plant communities often change with increasing aridity from diverse herbs to a few shrubs[58]. It is well known that the soil nutrient cycling or turnover rate is much slower for woody vegetation than for herbs,

which largely depends on soil microbial activities and diversity, particularly in more arid environments[5]. Therefore, plant diversity may promote resource availability and soil functioning via increased resource turnover in less arid environments, whereas slower resource turnover could weaken the positive relationship between plant diversity and soil multifunctionality under more arid conditions.

In contrast to plant diversity, soil microbial diversity was weakly correlated with soil multifunctionality across the aridity gradient. This reflects the fact that the relationship between the soil microbial diversity and multifunctionality varied among microbial taxa[22], suggesting that trade-offs may exist among the soil microbial diversity–multifunctionality relationship of different taxa[10]. For example, soil archaeal richness was negatively associated with soil multifunctionality, which could offset the positive association of soil fungal richness with multifunctionality. Furthermore, aridity may modify the relationship of soil microbial diversity with multifunctionality[22], as indicated by our finding that the slope of the relationship between the soil microbial diversity and multifunctionality shifted from negative to positive with increasing aridity. Consequently, different relationship patterns for less and more arid regions may have canceled out the overall relationship between soil microbial diversity and multifunctionality across the aridity gradient.

The unexpected negative association of soil microbial diversity with multifunctionality in less arid regions, combined with the strong and positive relationship between the plant diversity and multifunctionality in these regions, could be due to several reasons. First, increases in both the soil archaeal and bacterial richness, which were negatively correlated with soil multifunctionality in less arid regions, may reduce the dominance, and hence performance, of particular prokaryotic taxa with special functional significance (e.g., nutrient cycling contributed by ammonia-oxidizing archaea of the phylum *Thaumarchaeota*)[17,42], or may reduce the available resources for plant uptake via nutrient immobilization[59] and therefore slow down nutrient supply and resource recirculation in soils[14,21]. This could further decrease their positive interactions with other microorganisms based on syntrophy, or increase competitive strength toward other microorganisms and plants for nutrients, in turn leading to negative effects on plant productivity and associated soil functions[23,60,61]. Second, it is possible that the steeper decrease in plant diversity with increasing aridity was not compensated by the slower increase in soil microbial diversity on the less arid side of the gradient (see Supplementary Fig. 29), resulting in the greater importance of plant diversity than of soil microbial diversity, and presenting a net bottom-up effect of plant diversity in the regulation of soil multifunctionality in less arid regions[33–35,38].

As hypothesized, the relationship of soil microbial diversity with multifunctionality became tighter on the more arid side of the gradient, which was further confirmed by our microcosm experiment. In particular, soil multifunctionality under the drier conditions was consistently and positively associated with the diversity of soil fungi for both the field and microcosm study, which follows what has been found in boreal forests[28] and global drylands[14]. More interestingly, saprotrophic fungi were the only fungal guild whose diversity showed a significant and positive association with soil multifunctionality in more arid regions. These results reflected the dominance of top-down effects of soil microbial diversity in regulating multifunctionality by controlling resource outputs on the more arid side of the gradient, suggesting that soil multifunctionality depended mainly on microorganism-mediated decomposition and nutrient transformation under more arid conditions[30–33], and this is especially true for soil fungi. Fungi are generally considered to be crucial drivers of many ecosystem functions in drylands, including organic matter decomposition, C sequestration, C and N cycling, and exchange of resources between the soil, plants, and biocrust[14,32,48,62]. Such a "fungal loop" is favored by the environmental conditions typical of dryland ecosystems[30,62], which may result from better drought tolerance of fungi due to their extensive hyphal networks[14,63,64]. Saprotrophic fungi are the primary decomposers that promote decomposition, mineralization, and soil nutrient acquisition processes, as well as other above- and belowground functions linked to these processes[6,28]. For example, higher diversity of fungal saprotrophs boosts the rapid break down of organic matter from complex and recalcitrant polymers into simple and labile materials[65]. This process may contribute to multiple soil functions under infertile and more arid conditions where a large proportion of the primary productivity is returned to the soil as recalcitrant plant litter[5]. Similarly, diverse saprotrophic fungal communities may facilitate niche sharing among plant species and greater use of limiting nutrients by altering soil nutrient supply rates and resource partitioning, thereby increasing plant productivity and associated soil functions[23,60].

This study, which presents evidence on shifts in the biodiversity–soil multifunctionality relationships along a broad aridity gradient, has some limitations that should be addressed in future research. For instance, our field study measured soil functions that are representative of nutrient stocks but did not include variables related to soil process rates and aboveground processes. While nutrient stocks could be considered as indicators of longer-term net process rates that are too slow to be measured directly under natural conditions, the inclusion of actual process rates would better reflect the current status of ecosystem multifunctionality[2–4]. Therefore, focusing only on nutrient stocks could obscure the relationships between current biodiversity and soil multifunctionality. Acknowledging this, future studies should consider both nutrient pools and process rates to deepen our understanding of biodiversity–multifunctionality relationships. Furthermore, the DNA extraction method used to characterize soil microbial diversity focuses on the total microbial communities and fail to discriminate its active fraction, which may be related more closely to soil multifunctionality[66]. Despite this caveat, we expect the effects of the DNA extraction method to be minor because the studied dryland regions are characterized by high summer temperatures that fasten the degradation of relic DNA[67,68].

A number of recent studies show that aridification can lead to abrupt shifts in multiple ecosystem attributes[31,33,38,69,70]. Here, we identified a marked shift in the biodiversity–soil multifunctionality relationships at an aridity level of ~0.8 under field conditions in dryland ecosystems. Beyond this aridity level, soil microbial diversity had a stronger and positive association with soil multifunctionality, whereas below it, plant diversity showed a stronger and more positive association with multifunctionality. Our finding calls for climate-specific biodiversity conservation strategies to alleviate the negative impacts of aridification on soil functioning and service provisioning of dryland ecosystems. Our results also add support to the existence of a critical transition zone in biodiversity–soil multifunctionality relationships in drylands at the aridity level of ~0.8, —the boundary between the semiarid and arid climates. The geographical position of this boundary is highly vulnerable to aridification caused by land-use and climate change[33,71] (see also Fig. 6). Overall, these results suggest that the aridity level of 0.8 should be regarded as a critical ecological vulnerability zone that deserves strengthened protection and further research to avoid the negative impacts of human-induced global change.

## Methods

**Field survey and sampling**. Field data were collected from 130 study sites spanning a latitudinal gradient of 35.89−50.70° N and a longitudinal gradient of

76.62−122.41°E and covering five provinces across the temperate region in northern China (Xinjiang Autonomous Region, Qinghai Province, Gansu Province, Ningxia Autonomous Region, and Inner Mongolia Autonomous Region; Fig. 2a). Locations for the field study target natural drylands, delineated as areas with aridity level above 0.35 (ref. [30]), and represent a large aridity gradient including dry-subhumid ($N = 12$), semiarid ($N = 42$), arid ($N = 56$), and hyperarid ($N = 20$) regions (Fig. 2a), which are highly vulnerable to expected increases in aridity with human activity and climate change[33,71]. The aridity level of each site was calculated as 1 − AI, where AI is the ratio of precipitation to potential evapotranspiration[38]. We obtained AI from the Global Aridity Index and Potential Evapotranspiration Climate database (https://cgiarcsi.community/). The selection of the field sites aimed to minimize the potential impacts of human activity and other disturbances on soil, vegetation, and geomorphological characteristics based on the following three criteria: (i) sites were at least 1 km away from major roads and >50 km from human habitations; (ii) sites were under pristine or unmanaged conditions without visible signs of domestic animal grazing, grass/wood collection, engineering restoration plantings, and infrastructure construction; and (iii) the soil was dry without experiencing rainfall events for at least 3 days prior to sampling. Collectively, our field survey involved a wide range of the abiotic and biotic features of dryland ecosystems across northern China. These sites encompass the 14 soil types, i.e., arenosols, calcisols, cambisols, chernozems, fluvisols, gleysols, greyzems, gypsisols, kastanozems, leptosols, luvisols, phaeozems, solonchaks, and solonetz, and the four main vegetation types[44], i.e., typical grassland (dominated by *Stipa* spp., *Leymus* spp., *Cleistogenes* spp., and *Agropyron* spp.), desert grassland (dominated by *Stipa* spp., *Cleistogenes* spp., *Suaeda* spp., and *Artemisia* spp.), alpine grassland (dominated by *Stipa* spp., *Leymus* spp., *Carex* spp., and *Festuca* spp.), and desert (dominated by *Reaumuria* spp., *Salsola* spp., *Calligonum* spp., and *Nitraria* spp.). Elevation, mean annual temperature, and mean annual precipitation (1970–2000; https://www.worldclim.org/) of the sites varied from 204 to 3,570 m a.s.l. (mean, 1,294 m a.s.l.), from −4.3 to 12.8 °C (mean, 5.0 °C), and from 21 to 453 mm (mean, 195 mm), respectively (Supplementary Table 1).

Field sampling was conducted between June and September from 2015 to 2017 (each site was visited once over this period) following well-established standardized protocols as described in refs. [13,34]. In brief, three 30 m × 30 m quadrats were established at each site to represent the local vegetation and soil types that covered an area of no less than 10,000 m². The cover of perennial vegetation was estimated and all perennial plant species were listed by walking steadily along four 1.5 m × 30 m parallel transects (spaced 8 m apart) located within each quadrat using the belt transect method[72]. Site-level estimate for perennial plant cover was obtained by averaging the values measured in the 12 transects established. After vegetation survey, we located five 1 m × 1 m (for typical grassland, desert grassland, and alpine grassland) or five 5 m × 5 m (for desert) plots within each quadrat (at each corner and the center of the quadrat) to measure site-level plant aboveground and root biomass (g m⁻²). In each 1 m × 1 m plot, all grasses and dwarf shrubs were harvested to ground level for measurement of aboveground biomass. Five soil cores (7 cm diameter; 0–40 cm depth) per 1-m² plot were collected randomly, and the roots were removed using a 1-mm sieve and washed cleanly to measure root biomass. All shoot and root samples were dried to constant weight at 65 °C. In each 5 m × 5 m plot, we recorded the number of individuals per dominant shrub species and canopy cover and height of each individual, thereby estimating aboveground and root biomass according to the allometric models developed in previous studies that were conducted in the same regions as sampled here (see Supplementary Table 9 for details). Based on these measurements, we further estimated BNPP. However, BNPP is typically difficult to observe and measure, especially over large spatial scales and environmental gradients as in this study, because the root system is subject to simultaneous growth and turnover[73,74]. Across our survey areas, ~77–98% of the precipitation occurs between June and September (during the peak-growing season) corresponding to the period of the highest plant above- and belowground biomass[34,35,41,75]. Therefore, we argue that BNPP can be estimated approximately at each site by the following equation:

$$\frac{\text{Aboveground biomass}}{\text{Root biomass}} \cong \frac{\text{Aboveground net primary productivity (ANPP)}}{\text{BNPP}} \quad (1)$$

where both aboveground and root biomass are site-level measurements (g m⁻²). We used normalized difference vegetation index (NDVI) as a metric for ANPP as explained in recent studies in drylands[14,33,70]. NDVI data were obtained from the moderate resolution imaging spectroradiometer aboard NASA's Terra satellites (https://neo.sci.gsfc.nasa.gov/). We used the average NDVI values during our sampling dates as a proxy for ANPP at the site level as described in ref. [14].

Five soil cores (0–20 cm depth) per quadrat were then taken randomly under the canopies of the dominant perennial plant species and in bare areas devoid of perennial vegetation, respectively, and then were mixed as one sample for vegetation areas and the other sample for bare ground. When more than one dominant perennial plant species was observed, another three composite samples were collected under the canopies of co-dominant perennial plant species. All vegetation and soil surveys were carried out during the wet season (June to September) when biological activity and productivity are maximal; as such, we do not expect the different sampling times and years or seasonality to be a major factor influencing our conclusions. Collectively, 6–21 soil samples per site were collected, and in total 864 samples were taken and analyzed for each of the seven individual soil functions (see below) and multifunctionality. All soil functions

evaluated in the field study were calculated at site level by using a weighted average of the mean values observed in vegetated areas and bare ground by their respective cover[13,14,38]. After field sampling, the visible pieces of plant material were removed carefully from the soil, which was sieved and divided into three portions. The first portion was air-dried and used for soil organic C, total N, total P, available P, and pH analyses. The second portion was immediately mixed with 2 M KCl and stored at 4 °C for soil ammonium and nitrate analyses. The third portion was immediately frozen at –80 °C for assessing soil microbial diversity.

**Microcosm experiment**. In addition to the large-scale field study described above, we manipulated soil water availability in a microcosm experiment to evaluate the linkages between moisture content, soil microbial diversity, and multifunctionality. It is important to note that our intention is not to directly compare results between these two different approaches [i.e., in the field, measures of soil functions are related to nutrient pools, which we use to associate soil multifunctionality with both plant and soil microbial diversity, whereas in the microcosm experiment the measures of soil functions are related to process rates such as respiration rate and key enzyme activities (see below), which we use to associate soil multifunctionality with microbial diversity in the absence of plants]. Rather, by using an experimental microcosm approach, we aimed to complement the field study and thus further verify the potential increases in aridity to alter the relationship between soil microbial diversity and multifunctionality in the absence of plants. In parallel with the sampling protocols described above, we collected a greater mass of soil (c. 30 kg) under vegetation canopies from one site [i.e., Jingtai country (37.40°N, 104.26°E; Gansu Province, China)]. Soil type, mean annual temperature, mean annual precipitation, and aridity level (1970–2000; https://www.worldclim.org/) of the site is calcisols, 7.9 °C, 205 mm and 0.81, respectively. Following field sampling, the soil was stored immediately at 4 °C until subsequent processing in the laboratory.

In brief, a total of 30 experimental microcosms composed of 10 moisture levels with three replicates were established under sterile conditions in a closed incubation chamber (Supplementary Fig. 1a). Each microcosm was filled with 1 kg of soil. These microcosms were incubated at 18.5 °C [the annual mean land surface temperature (1981–2010) for the sampling site; http://data.cma.cn/en], and moisture contents were adjusted and artificially maintained at the ten levels respectively equivalent to 3, 5, 8, 10, 20, 40, 60, 80, 100, and 120% field capacity (27.6%) during the duration of the experiment for 30 days. The corresponding moisture content (%) measured at the end of the experiment varied from 2.03 ± 0.034 to 33.57 ± 1.94, which matched well with differences in moisture conditions among a subset of field soil samples ($N = 521$; Supplementary Fig. 1b). After incubation, the soil was removed from each microcosm; a portion of the soil was immediately frozen at –80 °C for molecular analysis, and the other fraction was air-dried, sieved, and stored at –20 °C for assessing multiple soil functions as described below.

**DNA extraction, PCR amplification, and amplicon sequencing**. For both the field and experimental studies, we assessed the diversity of soil archaea, bacteria, and fungi using Illumina-based sequencing. Genomic DNA was extracted from 0.5 g of each defrosted soil sample ($N = 864$ for the field study and $N = 30$ for the experimental study) using the PowerSoil® DNA Isolation Kit (MO BIO Laboratories, USA) according to the manufacturer's instructions. For our field study, extracted DNA was pooled at site level, ultimately resulting in 130 composite DNA samples under canopies of vegetation and in bare ground, respectively. Pooling DNA samples may outperform the commonly used method that extracts genomic DNA from mixed soil samples, which could remove large amounts of information on the diversity of soil microorganisms[14,22]. Negative controls (deionized $H_2O$ in place of soil) underwent identical procedures during the extraction to ensure zero contamination in downstream analyses.

The V3–V5 regions of the archaeal 16S rRNA gene were amplified using the primer pair Arch344F and Arch915R. Thermal conditions were composed of an initial denaturation of 3 min at 95 °C, ten cycles of touchdown PCR (95 °C for 30 s, annealing temperatures starting at 60 °C for 30 s then decreasing 0.5 °C per cycles, and 72 °C for 1 min), followed by 25 cycles at 95 °C for 30 s, 55 °C for 30 s, and 72 °C for 1 min, with a final extension at 72 °C for 10 min. The primer pair 338F and 806R was used for amplification of the V3–V4 regions of the bacterial 16S rRNA gene. Thermocycling conditions consisted of 3 min at 95 °C and then subjected to 30 amplification cycles of 30 s denaturation at 95 °C, 30 s annealing at 55 °C, followed by 72 °C for 45 s, and a final extension of 72 °C for 10 min. The fungal internal transcribed spacer (ITS) region 1 was amplified using the primer pair ITS1F and ITS2. The amplification conditions involved denaturation at 95 °C for 3 min, 35 cycles of 94 °C for 1 min, 51 °C for 1 min, and 72 °C for 1 min and a final extension at 72 °C for 10 min. Details of primers for each microbial taxa were given in Supplementary Table 10. These primers contained variable length error-correcting barcodes unique to each sample. All amplification reactions were performed in a total volume of 20 μl containing 4 μl of 5× FastPfu Buffer, 2 μl of 2.5 mM dNTPs, 0.8 μl of both the forward and reverse primers, 10 ng of template DNA, and 0.4 μl of FastPfu DNA Polymerase (TransGen Biotech., China). To mitigate individual PCR reaction biases each sample was amplified in triplicate and pooled together. All PCRs were done with the ABI GeneAmp® 9700 Thermal Cycler (Thermo Fisher Scientific, USA). PCR products were evaluated on 2.0% agarose gel with ethidium

bromide staining to ensure correct amplicon length, and were gel-purified using the AxyPrep DNA Gel Extraction Kit (Axygen Biosciences, USA). Purified amplicons were combined at equimolar concentrations and paired-end sequenced ($2 \times 300$ bp) on an Illumina MiSeq platform (Illumina, USA) at the Majorbio Bio-pharm Technology Co., Ltd. (Shanghai, China) according to standard protocols.

**Sequence processing.** Initial sequence processing was conducted with the QIIME pipeline[76]. Briefly, reads were quality-trimmed with a threshold of an average quality score higher than 20 over 10 bp moving-window sizes and a minimum length of 50 bp. Paired-end reads with at least 10 bp overlap and <5% mismatches were merged into full length sequences by using the FLASH program[77]. Sequences were de-multiplexed into samples based on their barcodes and a further round of quality control was performed to remove sequences containing any ambiguous bases or with a phred score <20 over the entire read length. 16S rDNA sequences were checked for chimeras by using the UCHIME algorithm[78] against "Gold" database (http://www.drive5.com/uchime/gold.fa). Chimeric ITS sequences were detected *de novo* by exploiting abundance data using UCHIME[78]. Identified chimeric sequences were eliminated from the dataset before downstream analyses, ultimately resulting in a total of 7,040,043/2,180,656 (field and experimental datasets, respectively), 7,141,265/2,349,035 and 9,045,735/3,831,302 high-quality chimaera-free sequences for archaea, bacteria, and fungi, respectively. Then, the sequences were clustered into OTUs according to 97% pairwise identity with the UCLUST algorithm in the USEARCH package[79]. Singleton OTUs were discarded. We determined the taxonomic identity of representative sequences from each OTU using the RDP Classifier[80] against the SILVA database v.128 release for prokaryotic OTUs or the UNITE database v.7.0 release for fungal OTUs. Sequences not classified at kingdom level or identified as non-microorganisms were removed. To correct sampling effort, the resultant OTU abundance tables were subsequently rarefied to the lowest number of sequences (10,707/27,265, 14,317/23,916, and 25,263/40,329 for archaea, bacteria, and fungi, respectively) found within an individual sample. Our resampled dataset included a total of 1,763/2,305, 14,060/5,604, and 9,463/1,786 OTUs for archaea, bacteria, and fungi, respectively.

**Measurement of individual soil functions.** For the field study, we measured in all soil samples the following seven variables related to nutrient pools: DNA concentration, soil organic C, total N, ammonium, nitrate, total P, and available P. These variables are key soil properties involving stocks of matter and energy that are representative of slow abiotic and biotic processes and can act as appropriate indicators of net process rates at long time scales such as soil C sequestration, soil nutrient storage capacity, and the build-up of soil fertility[2–4]. Variables such as these are the most commonly measured indicators for multifunctionality studies conducted in dryland ecosystems[4,13,14,49,81,82] and are considered to be critical determinants of soil functioning in natural drylands[13,33,38,83]. Collectively, these variables reflect multiple ecosystem functional categories including nutrient cycling, climate regulation, and soil properties and fertility which fall within the "supporting" and "regulating" ecosystem service categories[4,43]. In brief, organic C, total N, total P, and available P are good proxies of C, N, and P storage and also act as surrogates for C, N, and P availability for plants and microorganisms in drylands[13,38]. In particular, organic C is often used as an indicator of soil C sequestration[4], and N and P often limit the growth of plants and microorganisms, and ultimately food, fiber, and biomass production in drylands[38]. Ammonium and nitrate are the fraction of the soil N pool that is more readily available for plant and microbial uptake and are produced by microorganism-mediated processes such as nitrification, mineralization, and atmospheric-N fixation[84]. DNA concentration has been used as a powerful indicator of surface soil microbial biomass[14,45,67], which acts as a good substitute of microbial activity[4]. Across arid and semiarid regions in northern China, DNA concentration has recently been reported to be strongly related to soil microbial-biomass C and N (both Pearson's $r > 0.97$) estimated by the chloroform-fumigation extraction method[67]. Moreover, as a molecule rich in N and P, DNA could serve as a source of P, as well as C and energy, for soil microorganisms under nutrient-limiting conditions[85].

The determination of soil organic C was based on the Walkley–Black chromic acid wet oxidation method[86]. Total N was measured by the Kjeldahl sulfuric acid-digestion method[87]. Total soil P was determined by a molybdate colorimetric test after perchloric acid digestion[88]. Soil available P was assessed following a 0.5 M $NaHCO_3$ extraction[89]. Ammonium concentration was measured colorimetrically by the indophenol blue method[90]. Nitrate was first reduced to nitrite with hydrazine sulfate, and its concentration was determined from 2 M KCl extracts[91]. The concentration of DNA extracted as described above was measured with a NanoDrop™ 2000 UV-Vis spectrophotometer (Thermo Fisher Scientific, USA).

For the microcosm study, we measured in all soil samples seven variables: DNA concentration, activity of alkaline phosphatase (P mineralization), invertase (sucrose degradation), urease (N mineralization), β-glucosidase (starch degradation), catalase (hydrogen peroxide decomposition), and $CO_2$ fluxes (respiration). Extracellular enzymes such as those we measured are produced by soil microorganisms and the proximate agents of processes such as the stabilization and destabilization of soil organic matter, and measures of their activities are also considered a good indicator of soil fertility and microbial nutrient demand. In addition, respiration can be used as a proxy of soil microbial activity. Altogether, these variables are involved in microbial activity, the degradation and

mineralization of organic matter, and nutrient cycling in soil[13,46,81,92]. For our microcosm study, we did not measure those soil variables selected for our field study, except for DNA concentration, because those variables indicative of net process rates over long time periods were expected to be less sensitive to changes in soil moisture content and thus to be less affected during the short-term experiment (i.e., 30 days)[81]. Soil alkaline phosphatase, invertase, and urease activities were measured from air-dried soil as described in ref. [93]. The activity of β-glucosidase was assayed following the procedure described in ref. [13]. Soil catalase activity was determined by back-titrating residual $H_2O_2$ with $KMnO_4$ according to ref. [94]. Measurement of DNA concentration was done as described above. Soil $CO_2$ fluxes were measured at the end of experiment using a LI-COR 6400 portable $CO_2$ infrared gas analyzer with a 6400-09 soil respiration chamber (LI-COR Inc., Lincoln, NE, USA).

**Trade-offs and redundancy among soil functions.** For the field study, we evaluated potential trade-offs among individual soil functions that perform at high levels while others perform at low levels[3]. To do so, we calculated Pearson's correlation coefficients between each pair of individual soil functions. Among a total of 21 combinations we detected significant positive correlations in 18 of them and none showed a significant negative correlation (Supplementary Fig. 18a), suggesting no trade-offs between them. Additionally, the soil functions that were strongly correlated imply some degree of redundancy[3,19,82,95]. However, in only two cases (i.e., total N vs. DNA concentration and total N vs. organic C), correlations had $r$ values higher than 0.7 (refs. [19,82]), indicating that the degree of redundancy was not very high (Supplementary Fig. 18a).

**Assessment of soil multifunctionality.** We assessed soil multifunctionality using three basic methods: single-function, averaging, and multiple-threshold approaches, all of which have frequently been employed to measure multifunctionality in recent literature[13,14,22,26,45] and give complementary information for quantifying multifunctionality[3,10]. The averaging approach aims to combine a collection of soil functions into a single index that quantifies the average level of multiple soil functions. To obtain a quantitative multifunctionality index for each field site or experimental microcosm, we first normalized ($\log_{10}$-transformed when needed) and standardized each of the evaluated soil functions using the $Z$-score transformation. The $Z$-scores of the soil functions were then averaged to obtain a multifunctionality index for each field site or experimental microcosm[13]. For the field study, all selected individual soil functions had significant positive correlations with the multifunctionality index (Supplementary Fig. 18a). Although the averaged multifunctionality index has good statistical properties and provides an intuitively interpretable measure of the ability of an ecosystem to deliver multiple functions simultaneously, it ignores potential trade-offs among functions and assumes the substitutability of functions[3,10,13]. To address these limitations, we also quantified multifunctionality using the multiple-threshold approach for the field study. Multiple-threshold approach captures the number of functions that simultaneously exceed different thresholds of the maximum observed value of each function and evaluates whether more (or fewer) functions are performing simultaneously at high (or low) levels[10]. Following the recommendation as described in ref. [10], we determined the maximum level of functioning as the average of the top four values measured for the respective soil function among all field sites. Multiple-threshold multifunctionality was then calculated as the number of functions surpassing a series of consecutive thresholds (from 1 to 99% at 1% intervals) of the maximum of each function. This approach provides the following key indices to evaluate the relationships between biodiversity and multifunctionality: $T_{min}$ (the lowest threshold where biodiversity–multifunctionality relationships become significant), $T_{max}$ (the highest threshold beyond which biodiversity–multifunctionality relationships become nonsignificant) and $T_{mde}$ (the threshold where biodiversity shows a strongest positive or negative association with multifunctionality). Accordingly, $M_{min}$, $M_{max}$, and $M_{mde}$ indicate the number of functions (i.e., multiple-threshold multifunctionality) achieving at the respective thresholds[10]. Thus, it can be concluded that biodiversity exhibits a strong and positive association with multifunctionality if $T_{min}$ is low and the rest of the five indices are high; conversely, biodiversity exhibits a strong and negative association with multifunctionality if $T_{max}$ is high and the rest of the five indices are low[10]. Finally, we also realize that both multifunctionality metrics described above aggregate individual functions to characterize overall soil functioning, and thus may obscure relationships between biodiversity and key functions[3]. To address this problem, we incorporated the single-function approach for the field study, which facilitates mechanistic understanding of multifunctionality and the interpretations of results generated by using those aggregated metrics[10]. However, the relationships between soil multifunctionality and aridity or biodiversity evaluated by using both the single-function and multiple-threshold approaches were very similar to those obtained with the averaging approach, and hence we always used the averaged multifunctionality index as a metric of soil multifunctionality in the main text to make our results easier to compare.

**Metrics of plant and soil microbial diversity.** For the field study, we used the total number of plant species (plant species richness), archaeal OTUs (soil archaeal richness), bacterial OTUs (soil bacterial richness), and fungal OTUs (soil fungal

richness) tallied at each field site as a surrogate of plant, archaeal, bacterial, and fungal diversity, respectively. Given that fungi typically include several guilds (e.g., saprotrophs, pathogens, and symbionts), we further parsed fungal OTUs into these trophic modes based on their taxonomic assignments using the FUNGuild tool[96]. Following the recommendation as described in ref. [96], we only kept those fungal OTUs with mode assignments that are "highly probable" and "probable", so as not to overinterpret our results ecologically. We also calculated the diversity of each fungal trophic mode as the total number of OTUs within each mode tallied at each field site. Furthermore, for both the field and microcosm studies, we calculated a single index (i.e., soil microbial diversity index) to represent the overall changes in soil microbial diversity[20,22,45]. In brief, we first standardized all components of microbial diversity metrics (i.e., soil archaeal, bacterial, and fungal richness) by the $Z$-score transformation (overall mean of 0 and standard deviation of 1). The $Z$-scores of the diversity metrics were then averaged to obtain the soil microbial diversity index for each field site or experimental microcosm.

### Statistical analyses

*Field study.* Before statistical analyses, all the data were tested for normality. All soil functions, elevation and plant species richness were $\log_{10}$-transformed to improve the normality and homoscedasticity of residuals. We first evaluated the responses of each of the seven individual soil functions and multifunctionality to increasing aridity using linear and nonlinear [quadratic and generalized additive models (GAMs)] regressions (Fig. 2b–i), and the model with lower AIC value was selected in each case [differences in AIC ($\Delta$AIC) values >2 indicate that the models are different; Supplementary Table 2]. We further assessed whether soil multifunctionality responded more rapidly to aridity than did any individual soil functions. To this end, we explored the presence of aridity thresholds for those relationships that were better fitted by nonlinear regressions (Fig. 2b–i) using the standard protocols developed in ref. [33]. The presence of an aridity threshold means that once an aridity level is reached, a given variable either changes abruptly its value (i.e., discontinuous threshold) or its relationship with aridity (i.e., continuous threshold). Hence, a lower aridity threshold indicates that a given variable is more vulnerable to increasing aridity than are others[33]. We further fitted step (a linear regression that modifies only intercept at a given aridity level) and stegmented (showing changes both in intercept and slope at a given aridity level) regressions for the determination of discontinuous thresholds, and segmented (exhibiting changes only in slope at a given aridity level) regressions for continuous thresholds. Each of these models yields a change point (i.e., threshold) describing the aridity level that evidences the shift in a given non-linear relationship evaluated. We also used AIC to choose the best threshold model and the corresponding threshold in each case (Supplementary Table 2).

We then employed analysis of variance based on type-I sum of squares in a linear mixed-effects model (Eq. (2); Table 1) to test the relationships between the multiple biotic (BNPP, plant species richness, and the soil microbial diversity index) and abiotic (aridity, soil pH, and soil clay content) factors and soil multifunctionality:

$$
\begin{aligned}
\text{Soil multifunctionality} \sim{} & \text{Year} + \text{Plant species richness} \\
& + \text{Soil microbial diversity index} \\
& + \text{Plant species richness} \times \text{Soil microbial diversity index} + \text{Aridity} \\
& + \text{Aridity} \times \text{Plant species richness} \\
& + \text{Aridity} \times \text{Soil microbial diversity index} + \text{BNPP} + \text{Soil pH} + \text{Soil clay content} \\
& + \text{Elevation} + \text{Latitude} + \text{Longitude} + (1|\text{Soil type}) + (1|\text{Vegetation type})
\end{aligned}
\tag{2}
$$

where × indicates an interaction term. We obtained information on soil clay content (%) from the SoilGrids system (https://soilgrids.org/), and eliminated variation due to different sampling years by first entering the term "Year" into the statistical model[41]. The elevation, latitude, and longitude of the study sites were included to account for the spatial structure of our dataset[13,70]. To account for the similarities of soil and vegetation types among study sites we included "Soil type" and "Vegetation type" as random terms.

We further simplified the Eq. (2) to focus only on the relationships between aridity, biodiversity, and soil multifunctionality (Eq. (3); Supplementary Fig. 5). We did so because excluding additional biotic and abiotic factors did not change qualitatively the main results presented here (Table 1 and Supplementary Fig. 5), and therefore we used the simplest model to test our hypotheses more clearly. Our simplified model was:

$$
\begin{aligned}
\text{Soil multifunctionality} \sim{} & \text{Year} + \text{Plant species richness} \\
& + \text{Soil microbial diversity index} \\
& + \text{Aridity} + \text{Aridity} \times \text{Plant species richness} \\
& + \text{Aridity} \times \text{Soil microbial diversity index} \\
& + \text{Aridity} \times \text{Plant species richness} \times \text{Soil microbial diversity index} \\
& + (1|\text{Soil type}) + (1|\text{Vegetation type})
\end{aligned}
\tag{3}
$$

To evaluate how the biodiversity–multifunctionality relationships varied along aridity gradients, we conducted a moving-window analysis as detailed in ref. [69]. Briefly, we performed the linear mixed-effects model described in Eq. (3) for a subset window of 60 study sites with the lowest aridity values (this number of sites provided sufficient statistical power for our model), and repeated the same

calculations as many times as sites remained (i.e., 70). We then bootstrapped the standardized coefficients of each fixed term within each subset window, which was matched to the average value of aridity across the 60 sites. We fitted linear and nonlinear regressions to the bootstrapped coefficients of biodiversity and its interaction with aridity along aridity gradients (Fig. 3a, b and Supplementary Table 2), and identified the aridity thresholds for the changes in the coefficients of biodiversity (Fig. 3a and Supplementary Table 2) using the same procedure already described above. To provide further support for the aridity thresholds identified here, we also assessed the significance of the bootstrapped standardized coefficients of biodiversity and its interaction with aridity at 95% confidence intervals for each subset window (Fig. 3e). Before fitting threshold regressions, we evaluated whether the variables followed either a unimodal or bimodal distribution using the fitgmdist function in MATLAB (The MathWorks Inc., USA). Our results showed that all variables used for threshold detection presented unimodal distributions (Supplementary Table 11), suggesting that the three threshold regressions mentioned above (i.e., segmented, step, and stegmented) are appropriate in all cases[33]. We used the chngpt and gam packages in R (http://cran.r-project.org/) to fit segmented/step/stegmented and GAM regressions, respectively. To further check the validity of the thresholds identified, we bootstrapped linear regressions at both sides of each threshold for each variable. We then used the nonparametric Mann–Whitney $U$-test to compare the slope and the predicted value evaluated before and after each threshold. In all cases, we found significant differences in both of these two parameters (Fig. 3c, d and Supplementary Figs. 2, 3, 6).

Given a clear shift in the relationships between plant or microbial diversity and soil multifunctionality occurring at a threshold around an aridity level of 0.8 (Fig. 3), we further used OLS regressions to clarify the relationships between each component of plant or microbial diversity and soil multifunctionality in less and more arid regions separately, as well as across all sites (Fig. 4). To do so, we split our study sites into two groups: sites with aridity <0.8 (less arid regions; $N = 54$) and >0.8 (more arid regions; $N = 76$). Moreover, we fitted the mixed-effects model described in Eq. (2) for less and more arid regions separately to ensure the robustness of these bivariate correlations when accounting for multiple biotic and abiotic factors simultaneously, with the exception of using all components of microbial diversity metrics (i.e., soil archaeal, bacterial, and fungal richness) instead of the soil microbial diversity index in the models (Supplementary Table 4). All linear mixed-effects models were performed using the R package lme4. We used a variance inflation factor (VIF) to evaluate the risk of multicollinearity, and selected variables with VIF <10 in all cases[97]. Also, we evaluated whether a fitted mixed-effects model is singular (i.e., variance of any random term is close to zero) using the isSingular function. Moreover, we extracted the marginal (variance explained by fixed factors) and conditional (variance explained by fixed and random factors) $R^2$ values using the R package piecewiseSEM.

We next used SEMs to compare the hypothesized direct and indirect relationships between aridity, soil pH, soil clay content, BNPP, plant and microbial diversity, and soil multifunctionality in less and more arid regions (Fig. 5). The first step in SEM requires establishing an a priori model based on the hypothesized causal relationships among these variables (Supplementary Fig. 17a). Before modeling, bivariate correlations were checked between all variables to ensure that a linear model was appropriate (Supplementary Fig. 18b, c). We then parameterized our models using the grouped dataset and tested their respective goodness-of-fit statistics. Here we used the chi-squared test ($\chi^2$) and the root mean square error of approximation (RMSEA). Furthermore, as some of the variables didn't satisfy normality, the fit of the model was confirmed using the Bollen–Stine bootstrap test. All of these goodness-of-fit metrics revealed an acceptable fit of our a priori model, with the exception of removing the relationship between soil pH and BNPP with a coefficient close to zero for the SEM of more arid regions to improve its model fit (Fig. 5 and Supplementary Table 5).

We must note, however, that total soil P is typically considered to be controlled mostly by abiotic processes such as weathering of rocks rather than biotic processes in dryland ecosystems[38]. Also, total soil N was strongly correlated with DNA concentration and organic C in our dataset (Supplementary Fig. 18a), which may provide redundant information for the multifunctionality metrics used[3,95]. To ensure that these were not influencing our results, we repeated above analyses after excluding total soil N and P. These two sets of analyses provided very similar results (Supplementary Figs. 19–28 and Supplementary Tables 2, 3, 6–8), thus these issues do not affect the conclusions of this study.

Finally, we adopted a space-for-time substitution approach to quantitatively estimate the future changes in areas that are likely to cross the 0.8 aridity threshold identified in drylands across northern China (Fig. 6). To this end, we located both the expanding and shrinking areas with aridity levels crossing 0.8 projected for 2100 relative to 1970–2000 under two different scenarios (i.e., RCP 4.5 and 8.5, assuming saturated and exponential increases of $CO_2$ emissions, respectively) based on the aridity maps provided by Huang et al.[71]. Because the AI dataset (https://cgiarcsi.community/) has a spatial resolution of 30 arc-sec, we downscaled the data to a 0.5° × 0.5° resolution to match the aridity maps. And we excluded those areas that even will cross 0.8 aridity threshold by 2100 but that are not drylands today from our analyses to avoid overestimating our results[33]. All maps were visualized in ArcGIS 10.2 (ESRI, USA).

*Microcosm study.* Before analyses, soil respiration was $\log_{10}$-transformed to improve the normality and homoscedasticity of residuals. We first evaluated the

relationship between moisture content and soil multifunctionality using OLS regression (Supplementary Fig. 31a). We then used OLS regressions to assess the relationships of each component of microbial diversity (i.e., soil archaeal, bacterial, and fungal OTU richness) and the soil microbial diversity index with soil multifunctionality across the ten different moisture levels, as well as at both low and high moisture levels. Corresponding to the 0.8 aridity threshold identified in the field study, we selected 20% field capacity to split our dataset into two groups: 3−20% field capacity (low moisture levels) and 40−120% field capacity (high moisture levels). We did so because the value of moisture content (i.e., $6.09 \pm 0.39\%$) measured at 20% field capacity is within the maximum observed values of moisture content of field soil samples collected in arid regions (Supplementary Fig. 1). Thus, the selected moisture level could be closer to that of the lower boundary of arid regions (i.e., 0.8 aridity level) under field conditions. We only presented the result for soil bacterial richness (Supplementary Fig. 31b) because all of the other relationships were nonsignificant ($P > 0.05$). Finally, we also used SEMs to compare the hypothesized direct and indirect relationships between moisture content, microbial diversity, and soil multifunctionality at low and high moisture levels (see an a priori model in Supplementary Figs. 17b and 31c, d). Test of goodness-of-fit for SEMs were same as described above. All the SEM analyses were conducted using AMOS 21.0 (IBM SPSS Inc., USA). Data and code used to perform above analyses are available in figshare[98].

**Reporting Summary**. Further information on research design is available in the Nature Research Reporting Summary linked to this article.

## Data availability
The datasets that support the main findings of this study are publicly available on figshare [https://doi.org/10.6084/m9.figshare.15027561]. The raw archaeal sequence data generated in this study have been deposited in the NCBI SRA database under accession code PRJNA608843. The raw bacterial sequence data generated in this study have been deposited in the NCBI SRA database under accession code PRJNA609019. The raw fungal sequence data generated in this study have been deposited in the NCBI SRA database under accession code PRJNA609055. AI data are publicly available on the Global Aridity Index and Potential Evapotranspiration Climate Database [https://cgiarcsi.community/]. The remaining climate data reported in this study are publicly available on the WorldClim database [https://www.worldclim.org/]. Soil type and clay content data are publicly available on the Harmonized World Soil Database [https://iiasa.ac.at/] and SoilGrids system [https://soilgrids.org/], respectively. NDVI data are publicly available from the Moderate Resolution Imaging Spectroradiometer aboard NASA's Terra satellites [the MOD13Q1 product; https://neo.sci.gsfc.nasa.gov/].

## Code availability
The R code used to generate the main results of this study is publicly available on figshare [https://doi.org/10.6084/m9.figshare.15027561].

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

## Acknowledgements

We thank Duqing Zhu, Chuancong Dong, Miao Ye, and Fan Wu for their help with the field sampling, and Wim H. van der Putten, Cameron Wagg, Yunfeng Yang, Gaosen Zhang, Yongjun Liu, Qiang Yang, Dongshi Wan, and Qinfeng Guo for their helpful comments on earlier versions of the manuscript. Special thanks to Jianping Huang and Haipeng Yu for sharing and interpreting the projected future AI dataset. We also thank Guazhou Desert Ecosystem Field Observation Research Station and Core Facility of School of Life Sciences, Lanzhou University, for the convenience supplied during this work. This research was supported by the National Scientific and Technological Program on Basic Resources Investigation (no. 2019FY102002), National Natural Science Foundation of China (nos. 31770430 and 31700463), National Youth Top-notch Talent Support Program to J.D., China Postdoctoral Science Foundation (no. 2016M602890), Fundamental Research Funds for the Central Universities (no. lzujbky-2018-it05), and the Innovation Base Project of Gansu Province (no. 20190323). R.D.B. was supported by BBSRC GCRF grant BB/P022987/1 "Restoring soil function and resilience to degraded grasslands" and B.S. was supported by the University of Zurich Research Priority Program "Global Change and Biodiversity".

## Author contributions

J.D. conceived this study. Field data were collected by W.H., J.R., L.D., Q.D., M.J., S.Y., Y.S., Q.H., H.G., R.C., J.L., S.X., Z.W., H.H., X.L., Q.C. and J.D. Laboratory measurements were done by W.H., J.R., L.D., C.G., Q.H., H.G., J.L., Z.W., J.X., R.X., M.W., D.Z., Y.Z., J.L., H.Y., X.W., Y.D., Y.S., H.L., L.Z., X.L., M.A., A.M., M.A.A., X.L., R.L. and F.L. W.H., J.R. and L.D. performed data analyses with suggestion and help from S.Y., Y.S., Q.H., H.G., M.W., L.Z., C.H., J.L., J.-S.H., L.A., R.D.B., B.S. and J.D. The first draft was written

by W.H., J.R. and J.D. R.D.B. and B.S. contributed to the interpretation of data and the writing of the manuscript, and all authors contributed to the final version of the manuscript.

## Competing interests

The authors declare no competing interests.
