## [Peer Review File · Nature Communications]

Reviewer comments, first round -

Reviewer #1 (Remarks to the Author):

The manuscript tackles an important topic and presents new evidence on how the biodiversity–soil multifunctionality relationship changes across spatial aridity gradients. The novelty of the study is based on the finding that biodiversity of producers (plants) exhibits the strongest and positive association with multifunctionality at less arid ecosystems, while at more arid sites biodiversity of decomposers (soil microbes) shows the strongest association with multifunctionality. The observational field study is well design and has the benefit of including a very large spatial gradient of contrasting ecosystems and ranges in aridity, which is a strength. A microcosm experiment manipulating soil water availability complements the field study and confirms the finding that, in the absence of plants, the soil microbial diversity–multifunctionality relationship becomes tighter in lower compared to higher levels of water availability. This relationship is greatly important in terrestrial ecosystems, and the results in this manuscript suggest that a critical transition zone exists in biodiversity–multifunctionality relationships in drylands.

To my surprise, however, throughout the manuscript the authors present and interpret these findings as evidence for “stronger positive effects” of plant diversity on multifunctionality in less arid sites, and that in more arid sites “microbial diversity is the dominant driver of multifunctionality”. To me, this is a classic case of the language not fitting the evidence. The authors hypothesize that “microbial diversity is a dominant factor regulating soil multifunctionality in more arid regions, whereas plant diversity is of more importance for soil multifunctionality in less arid regions”. Although this language suggests that cause and effect between biodiversity and multifunctionality would be experimentally established in the study, only water availability is actually manipulated in the (microcosm) study. Therefore, neither the large field study nor the complementary microcosm experiment seems suitable to answer the postulated hypothesis. That would instead require the experimental manipulation of biodiversity of both plants and microbes. Given the absence of such experimental manipulation, any language that goes beyond the description and discussion of associations between biodiversity and multifunctionally and attempts to distinguish direction (i.e., biodiversity driving multifunctionality or vice-versa) is not supported by the results of this study. A causal relationship cannot be tested by correlation no matter how sophisticate it is. Until proven wrong, it may also be presumed that differences in soil nutrient status driven primarily by geochemical processes across the various ecosystems could control plant and microbial diversity...

To support the current description of results, discussion, and conclusions, the authors need to present the results of a manipulation experiment such as described above. Alternatively, the language of the manuscript must be deeply revised to properly reflect the study’s design and results. I comment on this and additional points below.

L 29 and 30-31: After reading the methods section, this would better change to “was associated with...” or similar.

L 50: or diversity of functions?

L 80: “As drivers”. After reading the methods, I suggest changing this to “was associated with...” or similar.

L 98-100: After reading the methods, the study’s design did not answer this hypothesis. I was hoping the microcosm experiment would involve manipulation of microbial diversity, but it did not. Water was the only variable manipulated.

L 113-114: This seems to be based on a presumed linear positive plant diversity–productivity relationship. As that is not always the cause, it would be helpful to show any references for these sites.

L 141-142: Why can’t the cause-effect here be the opposite (i.e. soil functions driving plants)?

L 182-183: Since this is based on SEM, not on a manipulation, it is not possible to say this.

L 190-192: Same as previous.

L 196-197: I do not think that the microcosm experiment confirmed this since only water availability was manipulated. The magnitude of microbes' roles in maintaining soil functions cannot be established from this experiment.

L 200: Sorry for being repetitive, but 'stronger positive effects' cannot be used here.

L 202: Same here.

L 217: Which key species? Why are they more important than others for multifunctionality?

L 233-239: How about nutrient immobilization by microbes? That seems like a very plausible mechanism explaining these patterns at less arid sites.

L 245-247: It is a stretch to say that the microcosm experiment confirmed a stronger effect of microbes on multifunctionality. Microbial diversity was never manipulated in either experiment and for that reason, it is not possible to establish cause-effect relationships. What would have happened if water was manipulated in soils devoid of microbes (sterile)? Wouldn't N, P, C change at all for pure geochemical reasons, not because of microbes?

L 262: Or, more accurately, associated with...

L 295: Clarify if each site was sampled once or repeatedly in these years.

L426-431: References needed for these methods.

L 522: Isn't NDVI accounting only for ANPP? If so, take into account that BNPP uses to be much higher than ANPP in arid sites.

Reviewer #2 (Remarks to the Author):

NCOMMS-21-00647 Hu et al.

This manuscript reports results from an extensive observational study across a large aridity gradient in China, coupled with an experiment manipulating water availability using soils from one of the sites. The aim is to assess shifts in the relationships between changes in the diversity of plants and soil microbes (archaea, bacteria and fungi) and those of soil nutrient stocks and transformation rates (the latter in the experiment). The main result is that the diversity of plants and fungi show strong relationships on soil functioning, but these relationships shifts with aridity. Specifically, at aridity levels ≤ 0.8 plants are major determinants of soil functioning, whereas at aridity levels ≥ 0.8 fungal diversity becomes an important driver of soil functioning.

I found the manuscript novel and of interest to the readership of Nature comm. It provides an uncommon assessment of the relative importance of changes in both plants and soil microbes for ecosystem functioning across aridity gradients in drylands, and shows a novel pattern explaining potential aridity thresholds affecting the functioning of dryland ecosystems, or at least the role of biodiversity in it. The Discussion is well written and clear, however, the Introduction, Methodology and Results could be clearer (see specific comments below). My main concerns are:

1) There are multiple statistical analyses, the rationale behind them, and complementarity amongst them is not well-explained. The interpolated map comes out of the blue, without any particular linkage to the initial objectives, and lacking a measure of the performance of these interpolations. I would prefer if the authors chose either the averaging or the multiple threshold approach to report changes in multifunctionality in the main text (of course, you can and should report the other set of results in the Suppl. Material), as using both is confusing. For example, I think results in Fig. 2 uses the multiple threshold approach to show the effects of aridity on multifunctionality (except for Fig. 2b perhaps), whereas Fig. 3 reports (I think, this wasn't clear) the averaging multifunctionality metric to evaluate relationships between diversity changes and functioning. The usage of different approaches makes it difficult to compare or be able to properly follow the results. More specific comments regarding the stats follows later on in this letter.

2) As the authors recognize, the experimental and observational studies are very hard to compare in any way. They measure different soil variables (nutrient concentrations in the observational, rates in the experiment), and the experiment uses only soil from one of the sites (very close to the aridity threshold of 0.8) and manipulate water availability but not microbial richness. The latter would be the necessary step to verify cause from correlation in the patterns found. I'm ok with joining both efforts in a single study, but the authors should be crystal clear of what the experiment is really contributing, or else, remove it from the study. This wasn't clear for me until

the very end of the manuscript.

3) The aridity threshold of 0.8 is a major part of the take-home message of this study, yet I found hard to find strong evidence in this regard, which is based on diversity x aridity interactions (which themselves do not show specifically such threshold), a seemingly arbitrary subsetting of their study sites in "low" and "high" aridity categories, and non-linear relationships (reported in the Suppl. Material, without a clear "breaking point at AI = 0.8). I think the authors should provide a stronger support for such threshold either using a moving window approach (fitting the effects of aridity, plant and microbe diversity on functioning across contrasting aridity levels; see Berdugo et al. 2019 cited in the text) or using standard tools designed specifically to evaluate the existence and place of potential thresholds (see Berdugo et al. 2020 cited in the text).

Specific comments

Throughout the manuscript: this is an observational study, and thus the authors should speak about relationships between diversity and functioning rather than effects of biodiversity on functioning.

L24-25: I would rather devote some space to briefly describe why would one expect plants and microbes to shift their importances as drivers of functioning across contrasting environments.

L33-34: Could the authors provide a quantitative estimation of the degree of this geographical expansion according to accepted future scenarios for aridity, please?

L63: According to Balvanera's and Lefcheck's et al. meta-analyses (2006 *ecology*, 2015 *nature comm.*), BEF relationships for plants are also variable. I would link this variability to a potential context-dependency of BEF (in general, including also plants), based on environmental conditions. You are dealing here with two major players of ecosystem functioning in drylands, and perhaps the most important environmental drivers, so this would make your study more appealing to readers.

L69: Provided that you define them before as detritivores and primary producers, these are just two trophic levels, not multiple. Furthermore, at least fungi include different trophic groups other than detritivores (pathogens, for example), how did the authors considered this in their analyses?

L58-79: This paragraph is a mix between two ideas that should be introduced more extensively and separately: 1) plants and microbes act in tandem to determine multifunctionality, hence we need to study them together, 2) BEF relationship are variable and potentially context-dependent, thus aridity can explain contrasting BEF relationships or shifts in the importance of plants and microbes as drivers of multifunctionality

L68-79: I found this text odd and confusing, rephrase for clarity, please. In addition, I think it would be good to mention more explicitly those studies on plant-microbe interactions in driving dryland multifunctionality (e.g., Delgado-Baquerizo et al. 2015, Jing et al. 2015, both in *Nature communications*), their main results, and the main gaps they left and that your study can help filling. The Introduction was a bit vague regarding what we know about the relationships between plant and soil biodiversity on the functioning of drylands, and the potential changes one might expect in these relationships when aridity changes.

L85: All of these are soil nutrient stocks, no rates of matter and energy, and not including aboveground processes. This should be acknowledged as a limitation in the Discussion. In addition, I was a bit surprised to find DNA concentration in the list of "functions". Since you are including soil microbes as predictors of soil functioning, I found a bit circular to include soil DNA as a response in this analysis too. Finally, a couple of these variables are highly correlated (L462; Suppl. Fig. 13), and therefore "double-counted" in the multifunctionality metrics. The authors should remove those variables providing redundant information as often recommended (see Manning et al. 2018 *Nature ecoevo*, for example)

L90: Agreed, but stocks may not reflect as well current rates of functioning and therefore could obscure the effect of current biodiversity on functioning. There are extensive discussions about the

importance of including rates in the measurement of ecosystem functioning that the authors seem to ignore.

L95: What is a "typical grassland" exactly? Which categories of vegetation types are these, could the authors cite a source?

L102-104: This is a bit circular: you expect that plants and microbes change their importance across aridity gradients (L98-100) because you assume that their importance will change with increasing aridity (L102-104)?

L104-110: This sentence is very hard to follow. Please, break into 3 and rewrite for clarity.

L109: I'm not really sure why the authors expect less plant productivity to drive stronger belowground competition exactly. Neither I am sure how aboveground competition for space and light would increase the dependency of soil functioning to plant inputs (L113-114). It would be good if the authors could elaborate more on this, indeed, this should be properly introduced before the "what we did" bit of the Introduction, so the reader understands the interest of what you did.

L110-114: Another long and convoluted sentence, break it into two and clarify.

L122: This is the first time an "interpolated map" is even mentioned. The authors should explain how this was obtained and what is it evaluating exactly. I found difficult to understand how an interpolated map is "demonstrating" anything, perhaps the best it can do is show some patterns.

L126: Instead of calling it a "sharp decline", it would be better to report the actual change (as a proportion of the values at the lowest AI, for example)

L131: Could you define what is T_{mde} exactly, please? I think it would help if you rewrote L128-129 for clarity. I couldn't really understand what T_{mde} is really informing about changes in multifunctionality.

L134: Increasing trends with what?

L171: The exact aridity levels dividing these less and more arid regions should be stated clearly early on in the Results section. I guess that the authors divided their data into two groups due to the contrasting plant richness x aridity and fungal diversity x aridity interactions, but this could be made clearer. The rationale behind each of the different analyses should also be explained. For example, it is not clear why the authors performed an SEM on top of the linear models described above.

L198 and elsewhere: the authors mention several times in their work the existence of some sort of threshold at an aridity level of 0.8. However, I failed to see a proper analysis regarding this potential threshold. I wonder if the authors could follow the approach in Berdugo et al. 2020 (Science, cited in the text) or a moving window analysis to test for this potential threshold in the relationships between plants vs fungal richness and soil multifunctionality.

L203-205: This refers to the stronger effects of plant richness under moderately arid conditions (i.e., plant richness x aridity interaction), not to the positive effects of plant richness on soil functioning per se.

L208-209: remove "from plant production and 208 declining influence on soil functions related to nutrient cycling and soil fertility", please. I think it will read much clearer.

L253: add "true" between "especially" and "for"

L283-284: This is a complicated way to explain that aridity index (AI) is counterintuitive in the sense that larger values mean wetter, not more arid, and that therefore you calculated the inverse (aridity level, as in the reference mentioned).

L284-286: And how was this aim achieved exactly, or how did you selected the site to achieve this aim?

L287-289: Are these different soil and vegetation types considered somehow in the analyses? They could confound part of the findings reported here

L296 This is a very large area and sampling effort to measure plant cover and richness, did you measured it visually walking around these 900 m² quadrats, or did you apply the point-intercept method in there somehow? I guess the different sites are represented by pH and clay content, but is this enough?

By the way, some of the predictors included in the linear models (Table 1) are very highly correlated with each other (Suppl. Figure 13). This could induce multicollinearity in the estimation of effect sizes, did the authors tested for this using VIF?

L308-309: You need to state here which soil samples are used to calculate multifunctionality. Information on L433-436 should be here.

L328: Do you think that the stronger importance of soil fungi for functioning when soil was fairly dry in the mesocosm experiment is related to the fact that soils came from a very arid site? Could the authors compare with other mesocosm experiments using different soils from contrasting environmental conditions from the published literature? Most importantly, the mesocosm experiment gives the false impression that you are able to causally link microbial diversity to soil functioning, whereas what you are measuring with this mesocosm experiment, as in the observational study, is that soil diversity and nutrients co-vary when water availability changes. This should be clearly mentioned in the text.

L472: You do not need to log-transform anything to obtain Z-scores

507-509: I don't really understand the link between this map and the objectives to be tested, so I suggest removing it. The 100+ sites are more than enough to evaluate relationships (not effects) between biodiversity, aridity and soil nutrients. If the authors insist in keeping it, I would like to see enough information to understand what exactly they did, or if it was any cross-validation applied to test how valid is this interpolation

L514-517: I don't really understand this analysis or how it complement the other ones. Could the authors develop more explanation to this, please? I think that the methods would be clearer if the Stats started with the main analysis (that on L518 onwards)

L534-535: Indicate that the effects of plant and soil microbial richness were tested simultaneously, please, this is one of the main points of the study. Did the authors included the 3- way interaction between plant richness x microbial richness x aridity?

L542: 0.5-0.8 is not equal to 0.8. Why did the authors divided their data according to the 0.8 level?

Figure 2: I would keep the map showing the sampling sites, panel b and panel e. I think the remaining contents are more confusing than anything else.

Fig. 4: Change to "Less arid" and "More arid" regions, please. Or even better, just state "sites with $AI \leq 0.8$ or ≥ 0.8 ", so the reader knows exactly what you mean.

Reviewer #3 (Remarks to the Author):

Ms. No: NCOMMS-21-00647

Title: Aridity-driven shift in biodiversity–soil multifunctionality relationships

This manuscript described results from an ambitious effort that explored how environmental

factors (in this case, aridity) affect biodiversity-soil multifunctionality relationship. The field experiment was very extensive, including 130 sites along a 4000-km gradient of aridity across Northern China. The research is original. The major result was that plant species richness was the dominant driver of soil multifunctionality in less arid regions, whereas microbial diversity (fungi in particular) played a greater role in more arid regions. It also identified the threshold aridity (0.8) for the shift in the relative role of plant and microbial diversity for soil multifunctionality. These results are interesting and deserve publication. The manuscript was in general well written. Still, I have a few concerns that may need to be addressed before the manuscript can be considered for publication.

First, the logic that seven nutrient stocks in soil were selected to represent the soil multifunctionality is not very clear. In other words, the justification was not convincing. This is not trivial because the number and type of "functions" selected likely affect the relationships between biodiversity and multifunctionality. The manuscript attempted to assert that biodiversity "drives" the multifunctionality (i.e., an integrated index of different labile and total C, N and P pools). However, this is debatable, at least. For example, it argued that plant and microbial diversity controls soil organic C. Yet, one can also argue that the opposite is right: soil organic C may dominate microbial diversity and to a lesser degree, plant diversity, as soil organic C dominates soil fertility and soil structure (niche availability for microbes). Moreover, taking total soil P as a soil biodiversity-driven function is highly questionable. Soil mineral composition (parental materials) and climate, rather than biological factors, are often believed to be the dominant determinants of total soil P.

Second, the linkages between microbial diversity at the DNA level and soil functions or the pool size of soil C, N and P have yet well established, although we know soil microbes (but not molecular diversity per se) exert major controls on soil C and N. Therefore, the relationships between diversity (both plant and microbial) and the most functions included in the multifunctionality index in the manuscript are largely correlational, not cause-effect ones.

Third, a study at the scale described in this manuscript is inherent with some issues with the timing and the scheme of sampling: Because of the extensive nature of field sampling, plants and soils were collected at different times: in different years and/or at different growing stages of plants when temperature and rainfall may also be significantly different. Some of soil functions as defined or selected in this manuscript, e.g., soil NO₃, are very sensitive to these temporal changes: plant uptake of NO₃ may quickly depletes soil NO₃ during the fast plant-growing season, but has little impact when plants are tiny. Similarly, a rainfall event may rapidly alter microbial transformations of soil N, or the microbial DNA concentration itself. It is unclear how field sampling was designed to deal with these issues. Also, the manuscript needs to better explain why integrating all the data obtained at different sampling times together is reasonable.

Minor comments

1. Sampling site selection: it is unclear how to select the soil type at each sampling site if multiple soil types exist at the location.

2. It is unclear why the microcosm experiment had high moisture treatments at 100% and 120% of field water holding capacity. I would think that with annual rainfall ranging from 21 to 453 mm, the soil moisture in any field site rarely reaches the field capacity, even just for a couple days. But the incubation experiment lasted for 30 days. Did results from these unrealistic high-water availability significantly skew the general trend of the relationship?

3. DNA extraction: Was relic DNA removed or considered? Did soils at the super dry sites have higher proportion of relic DNA?

4. The enzyme issue: Lns:437-441. The notion that soil enzyme activities and CO₂ fluxes in the incubation experiment "measure process rates directly and reflect the fluxes of matter and energy" is highly debatable. Linking soil enzyme activities with soil functions is always tricky. Most enzyme activities were estimated as the potential activities under the optimal conditions and often explain only a small proportion of variability of functions.

Replies to Reviewer 1

Reviewer #1 (Remarks to the Author):

We thank Reviewer 1 for the critical view on the manuscript and the suggestions for improvement. The most important changes related to comments by Reviewer 1 are:

- Because we recognize that including an additional experiment manipulating both plant and microbial diversity is not realistic, we have, as suggested, deeply revised the language of the manuscript to reflect the relationships between biodiversity and soil multifunctionality rather than the effects of biodiversity on multifunctionality.
- We have replaced ANPP with BNPP in all analyses of the revised manuscript (please see the revised Fig. 5, Table 1, Supplementary Fig. 28 and Supplementary Tables 4–8).

The manuscript tackles an important topic and presents new evidence on how the biodiversity–soil multifunctionality relationship changes across spatial aridity gradients. The novelty of the study is based on the finding that biodiversity of producers (plants) exhibits the strongest and positive association with multifunctionality at less arid ecosystems, while at more arid sites biodiversity of decomposers (soil microbes) shows the strongest association with multifunctionality. The observational field study is well design and has the benefit of including a very large spatial gradient of contrasting ecosystems and ranges in aridity, which is a strength. A microcosm experiment manipulating soil water availability complements the field study and confirms the finding that, in the absence of plants, the soil microbial diversity–multifunctionality relationship becomes tighter in lower compared to higher levels of water availability. This relationship is greatly important in terrestrial ecosystems, and the results in this manuscript suggest that a critical transition zone exists in biodiversity–multifunctionality relationships in drylands.

To my surprise, however, throughout the manuscript the authors present and interpret these findings as evidence for “stronger positive effects” of plant diversity on multifunctionality in less arid sites, and that in more arid sites “microbial diversity is the dominant driver of multifunctionality”. To me, this is a classic case of the language not fitting the evidence. The authors hypothesize that “microbial diversity is a dominant factor regulating soil multifunctionality in more arid regions, whereas plant diversity is of more importance for soil multifunctionality in less arid regions”. Although this language suggests that cause and effect between biodiversity and multifunctionality would be experimentally established in the study, only water availability is actually manipulated in the (microcosm) study. Therefore, neither the large field study nor the complementary microcosm experiment seems suitable to answer the postulated hypothesis. That would instead require the experimental manipulation of biodiversity of both plants and microbes. Given the absence of such experimental manipulation, any language that goes beyond the description and discussion of associations between biodiversity and multifunctionality and attempts to distinguish direction (i.e., biodiversity driving multifunctionality or vice-versa) is not supported by the results of this study. A causal relationship cannot be tested by correlation no matter how sophisticated it is. Until proven wrong, it may also be presumed that differences in soil nutrient status driven primarily by geochemical processes across the various ecosystems could control plant and microbial diversity...

To support the current description of results, discussion, and conclusions, the authors need to present the results of a manipulation experiment such as described above. Alternatively, the language of the manuscript must be deeply revised to properly reflect the study’s design and results. I comment on this and additional points below.

Response: We thank you for your appreciation of the large-scale dataset included in this work and your positive comment on the novelty and importance of the findings reported in our manuscript. Also, we thank you for highlighting the correlative nature of the field study, and, on reflection, agree that we overinterpreted our correlative data as evidence of causality. While we agree that a manipulation experiment of the kind

suggested would be needed to confirm the mechanistic basis of our observations, this wouldn't be realistic in the short term, as it would be a major undertaking. Therefore, and as suggested by you and the editor, we have toned down the language throughout the revised manuscript to reflect the relationships between biodiversity and soil multifunctionality rather than the effects of biodiversity on multifunctionality. While we are unable to include an additional experiment of this scale in this manuscript, we intend to take up the reviewers' great suggestion to perform a separate study to address this important issue in the future.

L 29 and 30-31: After reading the methods section, this would better change to “was associated with...” or similar.

Response: Thanks for your suggestion. We have revised as (Lines 29-32):

“Our results showed a strong positive association between plant species richness and soil multifunctionality in less arid regions, whereas microbial diversity, in particular of fungi, was positively associated with multifunctionality in more arid regions.”

L 50: or diversity of functions?

Response: Thanks. The sentence now reads (Lines 50-51):

“As a consequence, the diversity of species and functional traits within these communities may affect ecosystem functioning.”

L 80: “As drivers”. After reading the methods, I suggest changing this to “was associated with...” or similar.

Response: Thanks. We have revised as (Lines 115-116):

“To test our hypothesis, we evaluated how relationships between plant or soil microbial diversity and soil multifunctionality varied along a broad aridity gradient across northern China.”

L 98-100: After reading the methods, the study's design did not answer this hypothesis. I was hoping the microcosm experiment would involve manipulation of

microbial diversity, but it did not. Water was the only variable manipulated.

Response: We thank you for highlighting this issue. As discussed above, we are sorry that it would not be possible to include such experiment at this stage. We have revised the sentence, and it now reads (Lines 102-105):

“We predict that soil microbial diversity shows a stronger and positive association with ecosystem multifunctionality in more arid environments, whereas plant diversity exhibits a stronger and positive correlation with multifunctionality in less arid environments (Fig. 1).”

L 113-114: This seems to be based on a presumed linear positive plant diversity–productivity relationship. As that is not always the cause, it would be helpful to show any references for these sites.

Response: Thank you for pointing out this issue. Two studies reporting positive linear relationships between plant diversity and productivity in the same survey area have been cited here (Bai et al. 2007 and Ma et al. 2010; please see also Line 113).

Bai, Y. F. et al. Positive linear relationship between productivity and diversity: evidence from the Eurasian steppe. *J. Appl. Ecol.* **44**, 1023–1034 (2007).

Ma, W. H. et al. Environmental factors covary with plant diversity–productivity relationships among Chinese grassland sites. *Global Ecol. Biogeogr.* **19**, 233–243 (2010).

L 141-142: Why can't the cause-effect here be the opposite (i.e. soil functions driving plants)?

Response: We thank you for highlighting this issue, and sorry for overinterpreting the correlation-based analysis as evidence of causality. According to the comments of Reviewer 2 (L287-289), we have revised the Table 1 using a linear mixed-effects model method (soil and vegetation types were included as random terms). The sentence has now been revised as (Lines 163-166):

“We found that plant species richness, belowground net primary productivity (BNPP) and soil clay content were positively, whereas aridity and soil pH were negatively

associated with soil multifunctionality across the aridity gradient.”

L 182-183: Since this is based on SEM, not on a manipulation, it is not possible to say this.

Response: We agree with you. We have revised as (Lines 245-247):

“Most importantly, we detected a stronger positive relationship between soil microbial diversity (especially for bacteria and fungi) and multifunctionality.....”

L 190-192: Same as previous.

Response: Thanks. The sentence now reads (Lines 257-262):

“More importantly, aridity shifted the relationships between plant or microbial diversity and soil multifunctionality. Plant species richness was consistently and positively related to soil multifunctionality across the aridity gradient, and, as hypothesized, showed a stronger and more positive association with multifunctionality in less arid regions, whereas soil microbial diversity, in particular that of soil fungi, exhibited a stronger and positive association with multifunctionality in more arid regions.”

L 196-197: I do not think that the microcosm experiment confirmed this since only water availability was manipulated. The magnitude of microbes’ roles in maintaining soil functions cannot be established from this experiment.

Response: We agree with you. Indeed manipulating microbial diversity is needed to verify cause from correlational patterns observed in our field study. We have revised the sentence, and it now reads (Lines 263-267):

“Our microcosm experiment, which complemented the field study by experimentally manipulating soil water availability, confirmed that declining soil moisture content was a major driver of reduced soil multifunctionality, and that, in the absence of plants, the relationship between soil microbial diversity and multifunctionality became stronger and positive at low levels of soil moisture.”

L 200: Sorry for being repetitive, but ‘stronger positive effects’ cannot be used here.

Response: We thank you for pointing out this, and sorry again for the inappropriate expression. The sentence has now been revised as (Lines 267-270):

“Furthermore, we found that the shift in the relationships between plant or microbial diversity and soil multifunctionality occurred at an aridity level of ~ 0.8 in the field, corresponding to a soil moisture content of $\sim 6.09\%$ in the microcosm experiment in the absence of plants, below which soil microbial diversity was more strongly related to multifunctionality.”

L 202: Same here.

Response: Thanks. We have revised as (Lines 271-272):

“The consistent and positive relationship between plant species richness and soil multifunctionality across the aridity gradient aligned with previous reports.....”

L 217: Which key species? Why are they more important than others for multifunctionality?

Response: We thank you for pointing out this vague expression. Here, the key species refer to rare plant species. Our original intention is to discuss that rare plant species are typically more vulnerable to increasing aridity than common species (Tilman & Haddi 2002), and therefore the loss of rare species is more likely to account for the reduced plant diversity along the aridity gradients. Yet there is evidence suggesting that rare species can play a more important role in maintaining ecosystem multifunctionality than common species due to their lower degree of both functional trade-offs among species and redundancy in functional traits (Soliveres et al. 2016). Thus, we speculate that the loss of rare plant species could explain the weak relationship between plant diversity and soil multifunctionality under more arid conditions. However, due to their speculative nature we decided to remove these sentences from the revised manuscript. As an alternative, we have added additional explanations here; the added sentence reads (Lines 283-289):

“Alternatively, the dominant species in plant communities often change with

increasing aridity from diverse herbs to a few shrubs⁵². It is well known that the soil nutrient cycling or turnover rate is much slower for woody vegetation than for herbs, which largely depends on soil microbial activities and diversity, particularly in more arid environments⁵. Therefore, plant diversity may promote resource availability and soil functioning via increased resource turnover in less arid environments, whereas slower resource turnover could weaken the positive relationship between plant diversity and soil multifunctionality under more arid conditions.”

Tilman, D. & Haddi, A. Drought and biodiversity in grasslands. *Oecologia* **89**, 257–264 (1992).

Soliveres, S. et al. Locally rare species influence grassland ecosystem multifunctionality. *Phil. Trans. R. Soc. B* **371**, 20150269 (2016).

⁵²Yao, S. R. et al. Effects of water and energy on plant diversity along the aridity gradient across dryland in China. *Plants* **10**, 636 (2021).

⁵Wardle, D. A. et al. Ecological linkages between aboveground and belowground biota. *Science* **304**, 1629–1633 (2004).

L 233-239: How about nutrient immobilization by microbes? That seems like a very plausible mechanism explaining these patterns at less arid sites.

Response: We thank you for this great suggestion. We indeed overlooked this important point. This idea has now been incorporated as follows (Lines 308-310):

“....., or may reduce the available resources for plant uptake via nutrient immobilization⁵³ and therefore slow down nutrient supply and resource recirculation in soils^{14,21}.”

⁵³Li, Z. L. et al. Vital roles of soil microbes in driving terrestrial nitrogen immobilization. *Glob. Change Biol.* **27**, 1848–1858 (2021).

¹⁴Delgado-Baquerizo, M. et al. Microbial diversity drives multifunctionality in terrestrial ecosystems. *Nat. Commun.* **7**, 10541 (2016).

²¹Delgado-Baquerizo, M. et al. Multiple elements of soil biodiversity drive ecosystem functions across biomes. *Nat. Ecol. Evol.* **4**, 210–220 (2020).

L 245-247: It is a stretch to say that the microcosm experiment confirmed a stronger effect of microbes on multifunctionality. Microbial diversity was never manipulated in either experiment and for that reason, it is not possible to establish cause-effect relationships. What would have happened if water was manipulated in soils devoid of microbes (sterile)? Wouldn't N, P, C change at all for pure geochemical reasons, not because of microbes?

Response: We thank you for highlighting this issue, and sorry again for overinterpreting our correlative data as cause-effect relationships. We fully agree that manipulating soil microbial diversity is the necessary step to verify their effects on soil multifunctionality. This is a great direction that we will work on in the future. We have revised the sentence, and it now reads (Lines 318-320):

“As hypothesized, the relationship of soil microbial diversity with multifunctionality became tighter on the more arid side of the gradient, which was further confirmed by our microcosm experiment.”

Furthermore, although plants were absent in the microcosm experiment, our purpose of the manipulated experiment was to complement the field study and thus experimentally underpin the potential changes in the relationship between soil microbial diversity and multifunctionality with increasing aridity. Moreover, the soil was collected from the local environment in drylands under vegetation canopy (see also “Microcosm experiment” part in Methods). Thus, the microbes, soil nutrient pool and secondary metabolites of plants and microbes should reflect the situation in the original habitat. Although it is very difficult to manipulate soil microbial diversity and plant species richness for a microcosm experiment within one month, we could observe the non-linear relationships of both soil microbial diversity and multifunctionality with soil water content. However, the time span of the experiment was too short to detect changes in soil nutrient stocks. The empirical results, at least partially, can further prove the robustness of our field study.

L 262: Or, more accurately, associated with...

Response: Thanks again! To be more accurate, we have revised as (Lines 361-363):
“Beyond this aridity level, soil microbial diversity had a stronger and positive association with soil multifunctionality, whereas below it, plant diversity showed a stronger and more positive association with multifunctionality.”

L 295: Clarify if each site was sampled once or repeatedly in these years.

Response: We thank you for noticing this missing information. Each site indeed was sampled once during those years. This information has now been highlighted in Lines 403-405 as follows:

“Field sampling was conducted between June and September from 2015 to 2017 (each site was visited once over this period) following well-established standardized protocols as described in refs. 13 and 34.”

¹³Maestre, F. T. et al. Plant species richness and ecosystem multifunctionality in global drylands. *Science* **335**, 214–218 (2012).

³⁴Deng, J. M. et al. Plant mass–density relationship along a moisture gradient in north-west China. *J. Ecol.* **94**, 953–958 (2006).

L426-431: References needed for these methods.

Response: References for those methods have been added (Lines 569-574), thanks for spotting this missing information.

L 522: Isn't NDVI accounting only for ANPP? If so, take into account that BNPP uses to be much higher than ANPP in arid sites.

Response: We thank you for the suggestion. You are right that NDVI accounts only for ANPP. We also agree that BNPP can be more important than ANPP in dryland ecosystems. As suggested, we have replaced ANPP with BNPP in all analyses of the revised manuscript (please see revised Fig. 5, Table 1, Supplementary Fig. 28 and Supplementary Tables 4–8). These new results are qualitatively consistent with those derived using ANPP, and using one metric or the other does not change our

conclusions (see Figure 2 and Tables 1 and 2 below). Such consistency might be attributed to a certain proportional correlation between shoot growth rate and root growth rate for a given plant species under given environmental conditions (Deng et al. 2006; Chen et al. 2019). It is important to note, however, that BNPP is typically difficult to observe and measure in the field, especially over large spatial scales and environmental gradients. Therefore, we included additional measured data and tried our best to provide an approximate estimate of BNPP for each field site. Briefly, because approximately 77–98% of the precipitation occurs between June and September (during the peak-growing season) across our survey areas corresponding to the period of the highest plant aboveground and root biomass (Deng et al. 2006; Ma et al. 2010; Wang et al. 2018; Chen et al. 2019), we argue that BNPP can be estimated approximately at each site by the following equation:

$$\frac{\text{Aboveground biomass}}{\text{Root biomass}} \cong \frac{\text{ANPP}}{\text{BNPP}}$$

where both aboveground and root biomass are site-level measurements (g/m^2). We used NDVI as a metric for ANPP. For grasslands, aboveground biomass was measured by harvesting all plants, and root biomass was measured by the soil coring method. For deserts, we used the number of individuals per dominant shrub species and canopy cover and height of each individual to estimate site-level aboveground and root biomass according to the allometric models (please see Supplementary Table 9) developed in previous studies that were conducted in the same regions as sampled here.

We have further provided more detailed information on the approach in Lines 410-434.

Deng, J. M. et al. Plant mass–density relationship along a moisture gradient in north-west China. *J. Ecol.* **94**, 953–958 (2006).

Ma, W. H. et al. Environmental factors covary with plant diversity–productivity relationships among Chinese grassland sites. *Global Ecol. Biogeogr.* **19**, 233–243 (2010).

Wang, S., Wang, X. B., Han, X. G. & Deng, Y. Higher precipitation strengthens the microbial interactions in semi-arid grassland soils. *Global Ecol. Biogeogr.* **27**, 570–580 (2018).

Chen, R. F. et al. Life history strategies drive size-dependent biomass allocation patterns of dryland ephemerals and shrubs. *Ecosphere* **10**, e02709 (2019).

Figure 2. Structural equation models (SEMs) accounting for the hypothesized direct and indirect relationships between aridity, soil properties (pH and clay content), biodiversity (plant species richness and the soil microbial diversity index), ANPP and soil multifunctionality. a,b SEMs are shown for sites with aridity < 0.8 (a; $N = 54$) and > 0.8 (b; $N = 76$). Note that we only present significant relationships and their coefficients (numbers adjacent to arrows) for graphical simplicity. Latitude, longitude, and elevation of the field sites are included to account for the spatial structure of our dataset, and thus their coefficients are not included. Continuous and dashed arrows indicate positive and negative relationships, respectively. The thickness of the arrow is proportional to the magnitude of standardized path coefficients and indicative of the strength of the relationship. Asterisks indicate the significance level of each coefficient: * $P < 0.05$; ** $P < 0.01$; *** $P < 0.001$. R^2 is the proportion of variance explained by the model. Goodness-of-fit statistics for each SEM are given (d.o.f., degrees of freedom; RMSEA, root mean squared error of approximation).

Table 1. Linear mixed-effects model for the relationships between multiple biotic (ANPP, plant species richness and the soil microbial diversity index) and abiotic (aridity, soil pH and clay content) factors and soil multifunctionality with considering soil and vegetation types as random terms.

Term	df	ddf	MS	F	P	Estimate	VIF
Random terms are soil and vegetation types; Conditional R² 0.68; Marginal R² 0.59							
Year	1	44.0	3.53	9.79	0.003	-0.11	2.10
Plant species richness	1	12.2	11.15	30.91	< 0.001	0.02	2.28
Soil microbial diversity index	1	110.1	0	0	0.984	0.20	2.46
Aridity	1	29.0	7.95	22.06	< 0.001	-0.30	2.71
ANPP	1	106.1	0.06	0.15	0.696	0.08	1.94
Soil pH	1	108.9	4.38	12.16	< 0.001	-0.26	1.66
Soil clay content	1	114.5	6.01	16.67	< 0.001	0.19	2.10
Elevation	1	105.3	1.68	4.66	0.033	0.25	2.22
Latitude	1	115.2	4.95	13.72	< 0.001	0.36	4.11
Longitude	1	115.4	0.84	2.32	0.130	-0.17	1.82
Plant species richness × Soil microbial diversity index	1	108.1	0	0	0.955	0.20	4.48
Aridity × Plant species richness	1	102.9	1.46	4.06	0.047	-0.14	1.77
Aridity × Soil microbial diversity index	1	115.2	1.99	5.51	0.021	0.26	4.12

Fixed terms are fitted sequentially (type-I sum of squares) as indicated in Equation 2 in the main text, and × denotes an interaction term. Marginal (variance explained by fixed terms) and conditional (variance explained by fixed and random terms) R^2 values are given. The term “Year” is first introduced into the model to eliminate the variation due to different sampling years. Latitude, longitude, and elevation of the field sites are included to account for the spatial structure of our dataset. df, numerator degrees of freedom; ddf, denominator degrees of freedom; MS, mean squares; F , variance ratio; P , probability of type-I error; VIF, variance inflation factor.

Table 2. Linear mixed-effects models for the relationships between multiple abiotic (aridity, soil pH and clay content) and biotic (ANPP, plant species richness, and soil archaeal, bacterial, and fungal richness) factors and soil multifunctionality at sites with aridity < 0.8 and > 0.8.

Term	df	ddf	MS	F	P	Estimate	VIF
Sites with aridity < 0.8 (N = 54); Random terms are soil and vegetation types; Conditional R² 0.70; Marginal R² 0.65							
Plant species richness	1	17.7	13.58	43.31	< 0.001	0.27	6.10
Soil fungal richness	1	33.4	0.05	0.16	0.688	0.35	2.53
Soil archaeal richness	1	28.7	2.98	9.51	0.004	-0.41	6.27
Soil bacterial richness	1	34.5	7.10	22.64	< 0.001	-0.001	6.28
Aridity	1	34.9	0.04	0.14	0.714	-0.17	6.01
ANPP	1	32.0	0.14	0.45	0.507	0.13	5.93
Soil pH	1	36.9	0.001	0.002	0.969	-0.15	4.06
Soil clay content	1	36.8	0.44	1.40	0.245	0.18	1.93
Longitude	1	30.8	1.64	5.23	0.029	-0.37	3.75
Plant species richness × Soil fungal richness	1	31.2	0.65	2.07	0.160	0.13	4.12
Plant species richness × Soil archaeal richness	1	36.6	0.45	1.45	0.236	-0.07	5.94
Plant species richness × Soil bacterial richness	1	36.7	0.43	1.38	0.248	0.18	8.81
Aridity × Plant species richness	1	37.0	0.37	1.17	0.286	0.34	4.72
Aridity × Soil fungal richness	1	35.7	0.44	1.40	0.245	-0.07	3.54
Aridity × Soil archaeal richness	1	35.5	1.63	5.21	0.029	0.43	6.57
Aridity × Soil bacterial richness	1	33.8	0.09	0.29	0.592	0.10	7.26
Sites with aridity > 0.8 (N = 76); Random term is soil type; Conditional R² 0.3156; Marginal R² 0.3152							
Soil fungal richness	1	56.5	4.95	6.04	0.017	0.08	2.80
Soil archaeal richness	1	57.0	0.68	0.83	0.365	0.23	2.08
Soil bacterial richness	1	56.8	0.13	0.15	0.696	0.05	3.65
Plant species richness	1	56.9	0.44	0.54	0.465	-0.22	2.63
Aridity	1	53.3	4.23	5.16	0.027	-0.26	2.77
ANPP	1	48.3	0.67	0.82	0.369	0.14	2.31
Soil pH	1	55.1	1.77	2.17	0.147	-0.21	1.59
Soil clay content	1	56.9	7.40	9.04	0.004	0.24	3.36

Elevation	1	57.0	0.06	0.07	0.792	0.22	4.76
Latitude	1	56.7	4.17	5.09	0.028	0.36	6.38
Longitude	1	52.8	0.02	0.03	0.868	0.002	1.86
Plant species richness × Soil fungal richness	1	56.6	0.58	0.71	0.402	0.14	4.27
Plant species richness × Soil archaeal richness	1	55.1	0.05	0.06	0.812	0.07	2.82
Plant species richness × Soil bacterial richness	1	55.7	0.21	0.26	0.612	0.08	3.42
Aridity × Plant species richness	1	53.8	0.38	0.47	0.497	-0.06	2.09
Aridity × Soil fungal richness	1	57.0	1.22	1.49	0.227	0.15	3.45
Aridity × Soil archaeal richness	1	56.2	1.11	1.35	0.250	0.21	2.60
Aridity × Soil bacterial richness	1	56.5	0.20	0.24	0.625	0.13	5.27

Fixed terms are fitted sequentially (type-I sum of squares) as indicated in the table, and × denotes an interaction term. Soil and vegetation types are included as random terms. However, the term “vegetation type” is removed from the model fitted for sites with aridity > 0.8 because its variance is close to zero. To further address multicollinearity [the terms with VIF values > 10 (Hair et al. 1998)], we removed the terms “Year”, “Elevation”, and “Latitude” from the model fitted for sites with aridity < 0.8 and the term “Year” from the model fitted for sites with aridity > 0.8. Marginal (variance explained by fixed terms) and conditional (variance explained by fixed and random terms) R^2 values are shown. Latitude, longitude, and elevation of the field sites are included to account for the spatial structure of our dataset. df, numerator degrees of freedom; ddf, denominator degrees of freedom; MS, mean squares; F , variance ratio; P , probability of type-I error.

Hair, J. F., Black, W. C., Babin, B. J., Anderson, R. E. & Tatham, R. L. *Multivariate Data Analysis* (Prentice Hall, New Jersey, 1998).

Replies to Reviewer 2

Reviewer #2 (Remarks to the Author):

We thank Reviewer 2 for the thorough review of our manuscript and the critical points that were highlighted. We are addressing each point below. The most important changes to the manuscript are mentioned first.

- Parts of the Introduction have been expanded and reframed for clarity. Also, the section of hypothesis has been rewritten for clarity (Introduction, Lines 98-114).
- The rationale and complementarity among different statistical analyses are now explained thoroughly in the Results and Methods sections.
- We consistently used the averaging multifunctionality index to report our results in the main text, and moved the other set of results to the Supplementary Information. This information has further been highlighted in Lines 645-649 (Methods).
- The design and contribution of the microcosm experiment together with the caveats commented by the reviewers have been highlighted early on in the Introduction (Lines 133-141).
- We have used a moving-window approach combined with a standard procedure developed for the detection of ecological thresholds to provide stronger support for the existence of a threshold around 0.8 aridity level (please see new Fig. 3).
- We removed the interpolated map and moved the map showing field sites from the Supplementary Information to the main text. We have further revised Fig. 2 to be more informative about the nonlinear responses of individual soil functions and multifunctionality to increasing aridity and the aridity levels at which these responses showed abrupt changes (please see revised Fig. 2b-i). Also, we have added two maps providing a quantitative estimate of the future changes in areas with aridity crossing 0.8 under two different RCP scenarios (please see new Fig. 6).

This manuscript reports results from an extensive observational study across a large aridity gradient in China, coupled with an experiment manipulating water availability using soils from one of the sites. The aim is to assess shifts in the relationships between changes in the diversity of plants and soil microbes (archaea, bacteria and

fungi) and those of soil nutrient stocks and transformation rates (the latter in the experiment). The main result is that the diversity of plants and fungi show strong relationships on soil functioning, but these relationships shifts with aridity. Specifically, at aridity levels ≤ 0.8 plants are major determinants of soil functioning, whereas at aridity levels ≥ 0.8 fungal diversity becomes an important driver of soil functioning.

I found the manuscript novel and of interest to the readership of Nature comm. It provides an uncommon assessment of the relative importance of changes in both plants and soil microbes for ecosystem functioning across aridity gradients in drylands, and shows a novel pattern explaining potential aridity thresholds affecting the functioning of dryland ecosystems, or at least the role of biodiversity in it.

Response: We thank you for your positive comment about the originality and quality of this work, and your appreciation of the importance of the findings reported in our manuscript.

The Discussion is well written and clear, however, the Introduction, Methodology and Results could be clearer (see specific comments below). My main concerns are:

1) There are multiple statistical analyses, the rationale behind them, and complementarity amongst them is not well-explained. The interpolated map comes out of the blue, without any particular linkage to the initial objectives, and lacking a measure of the performance of these interpolations. I would prefer is the authors chose either the averaging or the multiple threshold approach to report changes in multifunctionality in the main text (of course, you can and should report the other set of results in the Suppl. Material), as using both is confusing. For example, I think results in Fig. 2 uses the multiple threshold approach to show the effects of aridity on multifunctionality (except for Fig. 2b perhaps), whereas Fig. 3 reports (I think, this wasn't clear) the averaging multifunctionality metric to evaluate relationships between diversity changes and functioning. The usage of different approaches makes it difficult to compare or be able to properly follow the results. More specific comments regarding the stats follows later on in this letter.

Response: We thank you for these valuable comments and suggestions.

- We have added additional analyses and results to the revised manuscript. Accordingly, parts of the Results and Methods have been expanded and revised.

Also, the rationale behind different statistical analyses and the complementarity among them have now been clarified thoroughly in the Results and Methods.

- We agree that the interpolated map didn't contribute substantially to the theme of this study, and thus we removed it from the revised manuscript (please see also responses to L122, L507-509 and Figure 2 below).
- As requested, we have used the averaging multifunctionality index to report our results throughout the main text and moved the other set of results to the Supplementary Information. The justification for doing so has also been highlighted in Methods as follows (Lines 645-649):

“However, the relationships between soil multifunctionality and aridity or biodiversity evaluated by using both the single-function and multiple-threshold approaches were very similar to those obtained with the averaging approach, and hence we always used the averaged multifunctionality index as a metric of soil multifunctionality in the main text to make our results easier to compare.”

2) As the authors recognize, the experimental and observational studies are very hard to compare in any way. They measure different soil variables (nutrient concentrations in the observational, rates in the experiment), and the experiment uses only soil from one of the sites (very close to the aridity threshold of 0.8) and manipulate water availability but not microbial richness. The latter would be the necessary step to verify cause from correlation in the patterns found. I'm ok with joining both efforts in a single study, but the authors should be crystal clear of what the experiment is really contributing, or else, remove it from the study. This wasn't clear for me until the very end of the manuscript.

Response: We thank you for highlighting this issue. We fully agree that manipulating soil microbial diversity (even both plant and soil microbial diversity) is needed to verify the causality from correlational patterns observed in our field study. We acknowledge this comment and will conduct such experiment in the future. However, as we mentioned in the originally submitted version, our intention was not to directly compare the two studies, which indeed have great differences in many ways as you point out. Instead, we argue that the microcosm experiment manipulating soil water availability can, at least partially, complement the field study and thus experimentally underpin the potential changes in the relationship (not effects) between soil microbial diversity and multifunctionality with increasing aridity in the absence of plants. In

light of this, we decided to keep the experiment in the revised manuscript. To further clarify this, the design and contribution of the microcosm experiment along with the caveats mentioned by you and other reviewers (please see also the response to L 245-247 raised by Reviewer 1) have now been highlighted in the Introduction (Lines 133-141), Results (Lines 236-239), Discussion (Lines 263-267) and Methods (Lines 453-463):

Lines 133-141: “In addition to the field study, we manipulated soil water availability in a microcosm experiment to experimentally test for linkages between moisture content, microbial diversity and soil multifunctionality by simulating differences in moisture conditions among our field sites (Supplementary Fig. 1; see also Methods for more details of experimental design). The purpose of the microcosm experiment was to complement the field study and thus experimentally underpin the potential changes in the relationship between soil microbial diversity and multifunctionality with increasing aridity in the absence of plants. Here we did not have the resources to take into account plant diversity and we assessed soil multifunctionality by measuring process rates because the time span of the experiment was too short to detect changes in soil nutrient stocks.”

Lines 236-239: “To complement the field study and further confirm the potential changes in the soil microbial diversity–multifunctionality relationship with increasing aridity in the absence of plants, we evaluated the linkages between moisture content, microbial diversity and soil multifunctionality by experimentally manipulating water availability in soil microcosms.”

Lines 263-267: “Our microcosm experiment, which complemented the field study by experimentally manipulating soil water availability, confirmed that declining soil moisture content was a major driver of reduced soil multifunctionality, and that, in the absence of plants, the relationship between soil microbial diversity and multifunctionality became stronger and positive at low levels of soil moisture.”

Lines 453-463: “In addition to the large-scale field study described above, we manipulated soil water availability in a microcosm experiment to evaluate the linkages between moisture content, soil microbial diversity and multifunctionality. It is important to note that our intention is not to directly compare results between these

two different approaches [i.e., in the field, measures of soil functions are related to nutrient pools, which we use to associate soil multifunctionality with both plant and soil microbial diversity, whereas in the microcosm experiment the measures of soil functions are related to process rates such as respiration rate and key enzyme activities (see below), which we use to associate soil multifunctionality with microbial diversity in the absence of plants]. Rather, by using an experimental microcosm approach, we aimed to complement the field study and thus further verify the potential increases in aridity to alter the relationship between soil microbial diversity and multifunctionality in the absence of plants.”

3) The aridity threshold of 0.8 is a major part of the take-home message of this study, yet I found hard to find strong evidence in this regard, which is based on diversity x aridity interactions (which themselves do not show specifically such threshold), a seemingly arbitrary subsetting of their study sites in “low” and “high” aridity categories, and non-linear relationships (reported in the Suppl. Material, without a clear “breaking point at AI = 0.8). I think the authors should provide a stronger support for such threshold either using a moving window approach (fitting the effects of aridity, plant and microbe diversity on functioning across contrasting aridity levels; see Berdugo et al. 2019 cited in the text) or using standard tools designed specifically to evaluate the existence and place of potential thresholds (see Berdugo et al. 2020 cited in the text).

Response: We thank you so much for these suggestions, which help us to provide stronger support for the existence of a 0.8 aridity threshold. As suggested, we first used a moving-window approach (Berdugo et al. 2019) to determine the changes in relationships between plant or microbial diversity and soil multifunctionality along the aridity gradient, on top of which we further detected their respective change point (i.e., threshold) using the standard procedure developed by Berdugo et al. (2020). Our new results support a clear shift in relationships between plant or microbial diversity and soil multifunctionality occurring at an aridity level around 0.8 (please see new Fig. 3). These results and the detailed approach have now been highlighted in Results (Lines 175-191) and Methods (Lines 713-726) sections.

Berdugo, M. et al. Aridity preferences alter the relative importance of abiotic and biotic drivers on plant species abundance in global drylands. *J. Ecol.* **107**, 190–202 (2019).

Berdugo, M. et al. Global ecosystem thresholds driven by aridity. *Science* **367**, 787–790 (2020).

Specific comments

Throughout the manuscript: this is an observational study, and thus the authors should speak about relationships between diversity and functioning rather than effects of biodiversity on functioning.

Response: Thank you for highlighting this issue. Indeed, it is problematic to imply causal relationships from the correlation we have found. In the revised version, we now only speak of “associated with” (or similar) throughout the manuscript (please see also the response to the main concern of Reviewer 1).

L24-25: I would rather devote some space to briefly describe why would one expect plants and microbes to shift their importance as drivers of functioning across contrasting environments.

Response: We thank you for the comment. The sentence has been revised, and it now reads (Lines 24-26):

“Relationships between biodiversity and multiple ecosystem functions (that is, ecosystem multifunctionality) are context-dependent and therefore may differ across contrasting environments.”

L33-34: Could the authors provide a quantitative estimation of the degree of this geographical expansion according to accepted future scenarios for aridity, please?

Response: We thank you for this interesting suggestion. We have added two maps (please see new Fig. 6) depicting both the expanding and shrinking areas with aridity levels crossing 0.8 projected for 2100 relative to 1970–2000 under RCP 4.5 and 8.5 scenarios based on the future aridity maps provided by Huang *et al.* (2016). Our new results indicate that the areas will expand by 11.5% and 28.3% until 2100 under RCP 4.5 and 8.5, respectively. Accordingly, this sentence has been revised as (Lines 32-35):

“This shift in the relationships between plant or microbial diversity and soil multifunctionality occurred at an aridity level of ~ 0.8 , the boundary between semiarid and arid climates, which is expected to advance geographically $\sim 28\%$ by the end of the current century.”

Further, the detailed information has also been highlighted in Results (Lines 228-234) and Methods (Lines 781-790).

Huang, J. P., Yu, H. P., Guan, X. D., Wang, G. Y. & Guo, R. X. Accelerated dryland expansion under climate change. *Nat. Clim. Change* **6**, 166–171 (2016).

L63: According to Balvanera’s and Lefcheck’s et al. meta-analyses (2006 *ecolett*, 2015 *nature comm.*), BEF relationships for plants are also variable. I would link this variability to a potential context-dependency of BEF (in general, including also plants), based on environmental conditions. You are dealing here with two major players of ecosystem functioning in drylands, and perhaps the most important environmental drivers, so this would make your study more appealing to readers.

Response: We thank you for highlighting this missing point, and your appreciation of the research topic. We agree that the relationship between plant diversity and ecosystem multifunctionality can vary with the environmental context. According to your later comments (see responses to L58-79 and L68-79 below), this sentence, and this section of the Introduction, have been rewritten and expanded as follows (Lines 77-79):

Lines 77-79: “It is likely, however, that relationships between biodiversity and ecosystem multifunctionality also depend on the environmental context and therefore may change along environmental gradients^{12,16,24,25}.”

¹²Isbell, F. et al. High plant diversity is needed to maintain ecosystem services. *Nature* **477**, 199–202 (2011).

¹⁶Fanin, N. et al. Consistent effects of biodiversity loss on multifunctionality across contrasting ecosystems. *Nat. Ecol. Evol.* **2**, 269–278 (2018).

²⁴Jucker, T. & Coomes, D. A. Comment on “Plant species richness and ecosystem multifunctionality in global drylands”. *Science* **337**, 155 (2012).

²⁵Perkins, D. M. et al. Higher biodiversity is required to sustain multiple ecosystem processes across temperature regimes. *Glob. Change Biol.* **21**, 396–406 (2015).

L69: Provided that you define them before as detritivores and primary producers, these are just two trophic levels, not multiple. Furthermore, at least fungi include different trophic groups other than detritivores (pathogens, for example), how did the

authors considered this in their analyses?

Response:

- We thank you for noticing this error. However, this section of the Introduction has been expanded and reframed for clarity according to your later comments (see responses to L58-79 and L68-79 below). This sentence has now been removed, and the idea has been incorporated into a separate section (Lines 58-76).
- Furthermore, we also thank you for highlighting the fact that fungi include different trophic modes. In light of this, we parsed fungal OTUs into three different trophic groups (i.e., fungal saprotrophs, pathogens, and symbionts) based on their taxonomic assignments using the FunGuild database. These new data were further used to evaluate the relationships between each component of microbial diversity and soil multifunctionality in less (aridity < 0.8) and more (aridity > 0.8) arid regions separately, as well as across all sites (please see the revised Fig. 4 and Supplementary Figs. 12–16). This information has now been highlighted in Results (Lines 192-207) and Methods (Lines 655-661). Interestingly, and as hypothesized, saprotrophic fungi were the only fungal guild whose diversity showed a significant and positive association with soil multifunctionality in more arid regions (please see the revised Fig. 4). This finding is now further discussed in Lines 333-342 as follows:

“Saprotrophic fungi are the primary decomposers that promote decomposition, mineralization and soil nutrient acquisition processes, as well as other above- and belowground functions linked to these processes^{6,28}. For example, higher diversity of fungal saprotrophs boosts the rapid break down of organic matter from complex and recalcitrant polymers into simple and labile materials⁶⁰. This process may contribute to multiple soil functions under infertile and more arid conditions where a large proportion of the primary productivity is returned to soil as recalcitrant plant litter⁵. Similarly, diverse saprotrophic fungal communities may facilitate niche sharing among plant species and greater use of limiting nutrients by altering soil nutrient supply rates and resource partitioning, thereby increasing plant productivity and associated soil functions^{23,54}.”

⁶Bardgett, R. D. & van der Putten, W. H. Belowground biodiversity and ecosystem functioning. *Nature* **515**, 505–511 (2014).

²⁸Li, J. et al. Fungal richness contributes to multifunctionality in boreal forest soil. *Soil Biol.*

Biochem. **136**, 107526 (2019).

⁶⁰Eastwood, D. C. et al. The plant cell wall–decomposing machinery underlies the functional diversity of forest fungi. *Science* **333**, 762–765 (2011).

⁵Wardle, D. A. et al. Ecological linkages between aboveground and belowground biota. *Science* **304**, 1629–1633 (2004).

²³Wang, X. Y. et al. High ecosystem multifunctionality under moderate grazing is associated with high plant but low bacterial diversity in a semi-arid steppe grassland. *Plant Soil* **448**, 265–276 (2020).

⁵⁴van der Heijden, M. G. A., Bardgett, R. D. & van Straalen, N. M. The unseen majority: soil microbes as drivers of plant diversity and productivity in terrestrial ecosystems. *Ecol. Lett.* **11**, 296–310 (2008).

L58-79: This paragraph is a mix between two ideas that should be introduced more extensively and separately: 1) plants and microbes act in tandem to determine multifunctionality, hence we need to study them together, 2) BEF relationships are variable and potentially context-dependent, thus aridity can explain contrasting BEF relationships or shifts in the importance of plants and microbes as drivers of multifunctionality.

Response: Thank you for these important suggestions, which make the Introduction more substantial and clearer. This section of the Introduction has been expanded and reframed for clarity, and further divided into two separate paragraphs. The first part (idea 1) has now been rewritten and highlighted in Lines 58-76, and the second part (idea 2) has also been substantially revised and highlighted in Lines 77-97.

L68-79: I found this text odd and confusing, rephrase for clarity, please. In addition, I think it would be good to mention more explicitly those studies on plant-microbe interactions in driving dryland multifunctionality (e.g., Delgado-Baquerizo et al. 2015, Jing et al. 2015, both in Nature communications), their main results, and the main gaps they left and that your study can help filling. The Introduction was a bit vague regarding what we know about the relationships between plant and soil biodiversity on the functioning of drylands, and the potential changes one might expect in these relationships when aridity changes.

Response: We thank you for these suggestions.

- As mentioned earlier, this section of the Introduction has been divided into two separate paragraphs for clarity (Lines 58-97) and reframed to include more of the

previous studies. Also, the two studies on the relationships between plant or soil microbial diversity and ecosystem multifunctionality that you highlighted have been introduced more extensively and explicitly (Lines 62-67, 69-72, 90-91). Furthermore, the main gaps relevant to our study have been highlighted, on top of which the objective and rationale of our study have further been clarified as follows (Lines 87-97):

“However, empirical data are still lacking for the linkages among both plant or soil microbial diversity and ecosystem multifunctionality along extended environmental gradients at large spatial scales. Importantly, although Jing *et al.*²² reported that regional-scale change in climate could mediate the relationships between plant or soil microbial diversity and ecosystem multifunctionality, the extent to which these relationships vary, and whether their relative strength shifts along environmental gradients, remains largely untested. This limits our predictive understanding of the potential ecological consequences of biodiversity change of both plants and soil microorganisms under different environmental conditions. This knowledge may be of particular importance if areas of conservation priority are to be identified and attempts to alleviate the effects of environmental change are made.”

²²Jing, X. et al. The links between ecosystem multifunctionality and above- and belowground biodiversity are mediated by climate. *Nat. Commun.* **6**, 8159 (2015).

- According to your later comments (see responses to L104-110 to L110-114 below), we have rewritten the hypothesis section and moved it to a new paragraph before “what we did” of the Introduction (Lines 98-114), where we elaborated why we expect a shift in relationships between plant or microbial diversity and soil multifunctionality along the environmental gradients.

L85: All of these are soil nutrient stocks, no rates of matter and energy, and not including aboveground processes. This should be acknowledged as a limitation in the Discussion. In addition, I was a bit surprised to find DNA concentration in the list of “functions”. Since you are including soil microbes as predictors of soil functioning, I found a bit circular to include soil DNA as a response in this analysis too. Finally, a couple of these variables are highly correlated (L462; Suppl. Fig. 13), and therefore

“double-counted” in the multifunctionality metrics. The authors should remove those variables providing redundant information as often recommended (see Manning et al. 2018 Nature ecoevo, for example)

Response: We thank you for highlighting these important issues.

- First, we agree on the issues related to the identity of soil “functions” measured in the field study. Indeed process rates may be as important as stocks of energy or matter when assessing the relationships between biodiversity and ecosystem functions and multifunctionality. As suggested, this point is now being acknowledged as a limitation in the Discussion (Lines 343-352; see also the response to L90 below).
- Second, there is a growing concern about the relationship between soil microbial diversity and biomass in recent years (Bastida et al. 2021). DNA concentration has been used as a powerful indicator of surface soil microbial biomass in many previous studies (e.g., Kuske et al. 2002; Johnson et al. 2012; Wagg et al. 2014; Delgado-Baquerizo et al. 2016) because it corresponds well with other methods that reflect microbial biomass. For example, we recently found that DNA concentration was strongly related to the soil microbial biomass carbon and nitrogen (both Pearson’s $r > 0.97$) estimated by classic chloroform-fumigation extraction method across arid and semiarid regions in northern China (Gong et al. 2021). In addition, as a molecule rich in N and P, DNA could be an important source of microbial nutrition (Pinchuk et al. 2008). In light of this, we decided to keep DNA concentration in our analyses. Also, the rationale for including DNA concentration as a soil “function” has further been highlighted in Methods as follows (Lines 562-568):

“DNA concentration has been used as a powerful indicator of surface soil microbial biomass^{14,62,80}, which acts as a good substitute of microbial activity⁴. Across arid and semiarid regions in northern China, DNA concentration has recently been reported to be strongly related to soil microbial-biomass C and N (both Pearson’s $r > 0.97$) estimated by the chloroform-fumigation extraction method⁶². Moreover, as a molecule rich in N and P, DNA could serve as a source of P, as well as C and energy, for soil microorganisms under nutrient-limiting conditions⁸¹.”

Bastida, F., Eldridge, D. J., García, C., Kenny Png, G., Bardgett, R. D. & Delgado-Baquerizo, M. Soil microbial diversity–biomass relationships are driven by soil carbon content across global

biomes. *ISME J.* <https://doi.org/10.1038/s41396-021-00906-0> (2021).

Kuske, C. R. et al. Comparison of soil bacterial communities in rhizospheres of three plant species and the interspaces in an arid grassland. *Appl. Environ. Microbiol.* **68**, 1854–1863 (2002).

Johnson, S. L., Kuske, C. R., Carney, T. D., Housman, D. C., Gallegos-Graves, L. V. & Belnap, J. Increased temperature and altered summer precipitation have differential effects on biological soil crusts in a dryland ecosystem. *Glob. Change Biol.* **18**, 2583–2593 (2012).

⁸⁰Wagg, C., Bender, S. F., Widmer, F. & van der Heijden, M. G. A. Soil biodiversity and soil community composition determine ecosystem multifunctionality. *Proc. Natl Acad. Sci. USA* **111**, 5266–5270 (2014).

¹⁴Delgado-Baquerizo, M. et al. Microbial diversity drives multifunctionality in terrestrial ecosystems. *Nat. Commun.* **7**, 10541 (2016).

⁶²Gong, H. Y. et al. Soil microbial DNA concentration is a powerful indicator for estimating soil microbial biomass C and N across arid and semi-arid regions in northern China. *Appl. Soil Ecol.* **160**, 103869 (2021).

⁸¹Pinchuk, G. E. et al. Utilization of DNA as a sole source of phosphorus, carbon, and energy by *Shewanella* spp.: ecological and physiological implications for dissimilatory metal reduction. *Appl. Environ. Microbiol.* **74**, 1198–1208 (2008).

⁴Garland, G. et al. A closer look at the functions behind ecosystem multifunctionality: a review. *J. Ecol.* **109**, 600–613 (2021).

- Finally, we thank you for highlighting the issue related to possible redundancy caused by highly correlated “functions”. We noticed that two pairs of individual functions (i.e., total nitrogen vs. DNA concentration and total nitrogen vs. organic carbon) had Pearson’s $r > 0.7$ (a value often used to evaluate the degree of redundancy among individual functions; Valencia et al. 2015, 2018; please see Supplementary Fig. 18a). To address this issue and the fact that total phosphorus is mostly driven by abiotic processes (this point was highlighted by Reviewer 3 and the editor), we removed total nitrogen and phosphorus and then repeated all analyses (please see Supplementary Figs. 19–28 and Supplementary Tables 2,3,6–8). Removing these two functions didn’t change our results in any way and thereby confirmed the robustness of our findings. This information has now been highlighted in Results (Lines 222-227) and Methods (Lines 773-780). In light of this, we still kept these two functions in the analyses of the main text.

Valencia, E. et al. Functional diversity enhances the resistance of ecosystem multifunctionality to aridity in Mediterranean drylands. *New Phytol.* **206**, 660–671 (2015).

Valencia, E. et al. Cascading effects from plants to soil microorganisms explain how plant species richness and simulated climate change affect soil multifunctionality. *Glob. Change Biol.* **24**, 5642–5654 (2018).

L90: Agreed, but stocks may not reflect as well current rates of functioning and therefore could obscure the effect of current biodiversity on functioning. There are extensive discussions about the importance of including rates in the measurement of ecosystem functioning that the authors seem to ignore.

Response: Thanks so much for highlighting this missing idea. Together with your comment above, these defects have now been highlighted as limitations in the Discussion as follows (Lines 343-352):

“This study, which presents new evidence on shifts in the biodiversity–soil multifunctionality relationships along a broad aridity gradient, has some limitations that should be addressed in future research. For instance, our field study measured soil functions that are representative of nutrient stocks, but did not include variables related to soil process rates and aboveground processes. While nutrient stocks could be considered indicators of longer-term net process rates that are too slow to be measured directly under natural conditions, the inclusion of actual process rates would better reflect the current status of ecosystem multifunctionality²⁻⁴. Therefore, focusing only on nutrient stocks could obscure the relationships between current biodiversity and soil multifunctionality. Acknowledging this, future studies should consider both nutrient pools and process rates to deepen our understanding of biodiversity–multifunctionality relationships.”

²Liu, X. J. et al. Tree species richness increases ecosystem carbon storage in subtropical forests. *Proc. R. Soc. B.* **285**, 20181240 (2018).

³Manning, P. et al. Redefining ecosystem multifunctionality. *Nat. Ecol. Evol.* **2**, 427–436 (2018).

⁴Garland, G. et al. A closer look at the functions behind ecosystem multifunctionality: a review. *J. Ecol.* **109**, 600–613 (2021).

L95: What is a “typical grassland” exactly? Which categories of vegetation types are these, could the authors cite a source?

Response: Typical grassland is one of the major vegetation types in dry-subhumid and semiarid regions of northern China according to the China’s vegetation atlas at a scale of 1:1,000,000 (Chinese Academy of Sciences, 2001). This reference has now been added (Line 130). To further clarify, the dominant plant species found for each of the four main vegetation types in drylands across northern China have also been

highlighted in Methods as follows (Lines 394-399):

“.....and the four main vegetation types⁴⁴, i.e., typical grassland (dominated by *Stipa* spp., *Leymus* spp., *Cleistogenes* spp. and *Agropyron* spp.), desert grassland (dominated by *Stipa* spp., *Cleistogenes* spp., *Suaeda* spp. and *Artemisia* spp.), alpine grassland (dominated by *Stipa* spp., *Leymus* spp., *Carex* spp. and *Festuca* spp.), and desert (dominated by *Reaumuria* spp., *Salsola* spp., *Calligonum* spp. and *Nitraria* spp.).”

⁴⁴Chinese Academy of Sciences. *Vegetation Atlas of China* (Science Press, Beijing, 2001).

L102-104: This is a bit circular: you expect that plants and microbes change their importance across aridity gradients (L98-100) because you assume that their importance will change with increasing aridity (L102-104)?

Response: We thank you for noticing this error. According to your later comments (see responses to L104-110 to L110-114 below), we have rewritten this section of the Introduction for clarity, and moved it to a new paragraph before “what we did” of the Introduction (Lines 98-114). The sentence now reads (Lines 98-105):

“Given the context-dependency of biodiversity–multifunctionality relationships and that plants and soil microorganisms may have different roles in maintaining ecosystem multifunctionality^{5–7,29}, we hypothesize that the relationships between plant or soil microbial diversity and ecosystem multifunctionality may shift along an aridity gradient due to changes in the net effects of interactions across trophic levels. We predict that soil microbial diversity shows a stronger and positive association with ecosystem multifunctionality in more arid environments, whereas plant diversity exhibits a stronger and positive correlation with multifunctionality in less arid environments (Fig. 1).”

⁵Wardle, D. A. et al. Ecological linkages between aboveground and belowground biota. *Science* **304**, 1629–1633 (2004).

⁶Bardgett, R. D. & van der Putten, W. H. Belowground biodiversity and ecosystem functioning. *Nature* **515**, 505–511 (2014).

⁷Naeem, S., Hahn, D. R. & Schuurman, G. Producer–decomposer co-dependency influences biodiversity effects. *Nature* **403**, 762–764 (2000).

²⁹Bardgett, R. D. & Wardle, D. A. *Aboveground–Belowground Linkages: Biotic Interactions, Ecosystem Processes, and Global Change* (Oxford University Press, Oxford, 2010).

L104-110: This sentence is very hard to follow. Please, break into 3 and rewrite for clarity.

Response: We thank you for pointing out this issue. This sentence has been revised for clarity, and it now reads (Lines 105-110):

“Specifically, we expect top-down effects of soil microbial decomposer diversity on ecosystem multifunctionality (via increasing organic matter decomposition and nutrient transformation) to be of more importance under more arid conditions^{30–32}. Here plants are scarce and primary productivity and consequent resource inputs to soils are limited³³, which increases belowground competition for limiting resources such as water and nutrients^{34–37}, and thus enhances dependency on soil microbial decomposers for ecosystem functioning^{14,21} (Fig. 1).”

³⁰Pointing, S. B. & Belnap, J. Microbial colonization and controls in dryland systems. *Nat. Rev. Microbiol.* **10**, 551–562 (2012).

³¹Wang, C. et al. Aridity threshold in controlling ecosystem nitrogen cycling in arid and semi-arid grasslands. *Nat. Commun.* **5**, 4799 (2014).

³²Makhalanyane, T. P. et al. Microbial ecology of hot desert edaphic systems. *FEMS Microbiol. Rev.* **39**, 203–221 (2015).

³³Berdugo, M. et al. Global ecosystem thresholds driven by aridity. *Science* **367**, 787–790 (2020).

³⁴Deng, J. M. et al. Plant mass–density relationship along a moisture gradient in north-west China. *J. Ecol.* **94**, 953–958 (2006).

³⁵Chen, R. F. et al. Life history strategies drive size-dependent biomass allocation patterns of dryland ephemerals and shrubs. *Ecosphere* **10**, e02709 (2019).

³⁶Deng, J. M. et al. Trade-offs between the metabolic rate and population density of plants. *PLoS ONE* **3**, e1799 (2008).

³⁷Chen, R. F. et al. Effects of biotic and abiotic factors on forest biomass fractions. *Natl. Sci. Rev.* **0**, nwab02 (2021).

¹⁴Delgado-Baquerizo, M. et al. Microbial diversity drives multifunctionality in terrestrial ecosystems. *Nat. Commun.* **7**, 10541 (2016).

²¹Delgado-Baquerizo, M. et al. Multiple elements of soil biodiversity drive ecosystem functions across biomes. *Nat. Ecol. Evol.* **4**, 210–220 (2020).

L109: I’m not really sure why the authors expect less plant productivity to drive stronger belowground competition exactly. Neither I am sure how aboveground competition for space and light would increase the dependency of soil functioning to

plant inputs (L113-114). It would be good if the authors could elaborate more on this, indeed, this should be properly introduced before the “what we did” bit of the Introduction, so the reader understands the interest of what you did.

Response: We thank you for these comments and suggestions. These sentences have now been rewritten for clarity (please see also responses to L104-110 and L110-114).

- First, we argue that plants are typically scarce in more arid environments where plant-derived resource inputs to soil are limited by low primary productivity. This would increase belowground competition for available resources and therefore dependency of ecosystem functioning on nutrient supply via microorganisms-mediated organic matter decomposition and nutrient transformation. Thus, we expected that net top-down effects driven by the diversity of soil microbial decomposers would be of more importance for ecosystem multifunctionality under more arid conditions by controlling resource outputs and by feedback effects to plant communities (see also the response to L104-110 above).
- Second, we argue that primary productivity is less restricted by water shortage and that larger available biotope space may increase the potential for niche complementarity among plant species (Pfisterer & Schmid 2002; Dimitrakopoulos & Schmid 2004; Jousset et al. 2011), and thus plant diversity–productivity relationships are expected to develop to a greater extent in less arid environments. Therefore, we expected that net bottom-up effects driven by the diversity of plant producers would be of more importance for ecosystem multifunctionality under less arid conditions by controlling resource inputs and by cascading effects on soil microbial communities (see also the response to L110-114 below).

Pfisterer, A. B. & Schmid, B. Diversity-dependent production can decrease the stability of ecosystem functioning. *Nature* **416**, 84–86 (2002).

Dimitrakopoulos, P. G. & Schmid, B. Biodiversity effects increase linearly with biotope space. *Ecol. Lett.* **7**, 574–583 (2004).

Jousset, A., Schmid, B., Scheu, S. & Eisenhauer, N. Genotypic richness and dissimilarity opposingly affect ecosystem functioning. *Ecol. Lett.* **14**, 537–545 (2011).

- Finally, according to your suggestion, this section of the Introduction has now

been moved before the “what we did” of the Introduction (Lines 98-114). Furthermore, we have also revised this section for clarity (Lines 98-114; please see also the responses to L104-110 and L110-114).

L110-114: Another long and convoluted sentence, break it into two and clarify.

Response: Thank you for the suggestion. This sentence now reads (Lines 111-114): “In contrast, we expect bottom-up effects driven by the diversity of plant producers (via controlling resource inputs) to be of more importance under less arid conditions^{33,38}. Here primary productivity is less restricted by water shortage and thus plant diversity–productivity^{39–41} and –multifunctionality relationships are expected to develop to a greater extent (Fig. 1).”

³³Berdugo, M. et al. Global ecosystem thresholds driven by aridity. *Science* **367**, 787–790 (2020).

³⁸Delgado-Baquerizo, M. et al. Decoupling of soil nutrient cycles as a function of aridity in global drylands. *Nature* **502**, 672–676 (2013).

³⁹Pfisterer, A. B. & Schmid, B. Diversity-dependent production can decrease the stability of ecosystem functioning. *Nature* **416**, 84–86 (2002).

⁴⁰Bai, Y. F. et al. Positive linear relationship between productivity and diversity: evidence from the Eurasian steppe. *J. Appl. Ecol.* **44**, 1023–1034 (2007).

⁴¹Ma, W. H. et al. Environmental factors covary with plant diversity–productivity relationships among Chinese grassland sites. *Global Ecol. Biogeogr.* **19**, 233–243 (2010).

L122: This is the first time an “interpolated map” is even mentioned. The authors should explain how this was obtained and what is it evaluating exactly. I found difficult to understand how an interpolated map is “demonstrating” anything, perhaps the best it can do is show some patterns.

Response: We thank you for highlighting this issue. As you commented elsewhere (see responses to your main concern 1 above and to L507-509 and Figure 2 below), the interpolated map indeed had little to do with the objectives of the study. We have now therefore removed it from the revised manuscript.

L126: Instead of calling it a “sharp decline”, it would be better to report the actual change (as a proportion of the values at the lowest AI, for example)

Response: Thank you for the suggestion. According to your comments below (please see response to L514-517 below), we have revised the Fig. 2, and reported the

nonlinear responses of individual soil functions and multifunctionality to increasing aridity as well as the aridity levels (i.e., aridity thresholds) at which these responses showed abrupt changes (please see the revised Fig. 2b-i). Therefore, the results (i.e., the original Supplementary Fig. 2) reporting the linear relationships between aridity and each of the seven individual soil functions measured have now been removed. Accordingly, this sentence has also been removed.

L131: Could you define what is T_{mde} exactly, please? I think it would help if you rewrote L128-129 for clarity. I couldn't really understand what T_{mde} is really informing about changes in multifunctionality.

Response: We thank you for pointing out this vague expression. According to a comment by you above (please see response to your first main concern), these results have been moved to Supplementary Information (please see Supplementary Fig. 4). These results reported the relationship between aridity and multiple-threshold multifunctionality. T_{mde} denotes the threshold (a certain percentage within a 1 to 99% range) where a variable shows the strongest positive or negative association with multifunctionality. Note that T_{mde} alone is not sufficient to reflect the strength of the relationship between a variable and multifunctionality. Rather, a combination of T_{mde} and the other five indices (i.e., T_{min} , T_{max} , M_{min} , M_{max} , and M_{mde}) is needed to comprehensively understanding how a variable influences (or relates to) multifunctionality. For example, we expect that aridity has strong and negative effects on soil multifunctionality if T_{max} is high and the rest of the five indices are low. To be clearer, we have now rewritten the sentence for clarity (Lines 154-161) and provide more explanations for the six indices in Methods as follows (Lines 630-640):

Lines 154-161: "Similarly, the multiple-threshold approach provided additional evidence that aridity had strong and negative effects on soil multifunctionality (Supplementary Fig. 4 and Supplementary Table 3). Aridity was negatively related to the number of soil functions surpassing a broad threshold spectrum from 1% (T_{min}) to 99% (T_{max}), with the negative relationship peaking at a low threshold of 10% (T_{mde} ; Supplementary Fig. 4 and Supplementary Table 3). On average, three soil functions were performing at T_{mde} and none at T_{max} , which together indicated that multiple soil functions were suppressed to low performance levels by increasing aridity (Supplementary Fig. 4a and Supplementary Table 3)."

Lines 630-640: “This approach provides the following key indices to evaluate the relationships between biodiversity and multifunctionality: T_{\min} (the lowest threshold where biodiversity–multifunctionality relationships become significant), T_{\max} (the highest threshold beyond which biodiversity–multifunctionality relationships become non-significant) and T_{mde} (the threshold where biodiversity shows a strongest positive or negative association with multifunctionality). Accordingly, M_{\min} , M_{\max} and M_{mde} indicate the number of functions (i.e., multiple-threshold multifunctionality) achieving at the respective thresholds¹⁰. Thus, it can be concluded that biodiversity exhibits a strong and positive association with multifunctionality if T_{\min} is low and the rest of the five indices are high; conversely, biodiversity exhibits a strong and negative association with multifunctionality if T_{\max} is high and the rest of the five indices are low¹⁰.”

¹⁰Byrnes, J. E. K. et al. Investigating the relationship between biodiversity and ecosystem multifunctionality: challenges and solutions. *Methods Ecol. Evol.* **5**, 111–124 (2014).

L134: Increasing trends with what?

Response: We thank you for noticing this vague expression. According to your comments below (see response to L514-517 below), we have removed the original Fig. 2e from the revised manuscript. Therefore, this sentence has now been removed accordingly.

L171: The exact aridity levels dividing these less and more arid regions should be stated clearly early on in the Results section. I guess that the authors divided their data into two groups due to the contrasting plant richness x aridity and fungal diversity x aridity interactions, but this could be made clearer. The rationale behind each of the different analyses should also be explained. For example, it is not clear why the authors performed an SEM on top of the linear models described above.

Response: We thank you for highlighting this important issue. As mentioned earlier, we have used a moving-window approach (the relationships between plant or microbial diversity and soil multifunctionality were tested simultaneously; please see also the response to L534-535 below) and a standard procedure for the detection of ecological thresholds based on your suggestions to better support the existence of a 0.8 aridity threshold (please see the new Fig. 3). This information has now been

highlighted explicitly early on in the Results as follows (Lines 192-196):

“Given the clear shift in biodiversity–soil multifunctionality relationships detected at an aridity level of around 0.8 (Fig. 3), we further divided the study sites into two groups, namely sites with aridity < 0.8 and > 0.8 , representing less and more arid regions respectively, to examine whether there was a significant linear relationship between each component of biodiversity and soil multifunctionality in less and more arid regions.”

Furthermore, we have added a number of new analyses and results to the revised manuscript. Accordingly, sections of the Results and Methods have further been expanded and revised. Also, the rationale behind each of the different analyses and the complementarity among them have now been clarified thoroughly in the Results and Methods (please see also the response to your first main concern above).

Finally, we fitted the linear mixed-effects models for less and more arid regions separately (see the revised Supplementary Table 4) to ensure the robustness of the bivariate correlations (see the revised Fig. 4) when accounting for multiple biotic and abiotic factors simultaneously, while as a complement, we further performed SEMs to infer the hypothesized direct and indirect relationships between multiple biotic and abiotic factors and soil multifunctionality, and to test whether different indirect pathways may drive the aridity–biodiversity–multifunctionality relationships in less and more arid regions (see the revised Fig. 5). This information has now also been highlighted in the Results and Methods.

L198 and elsewhere: the authors mention several times in their work the existence of some sort of threshold at an aridity level of 0.8. However, I failed to see a proper analysis regarding this potential threshold. I wonder if the authors could follow the approach in Berdugo et al. 2020 (Science, cited in the text) or a moving window analysis to test for this potential threshold in the relationships between plants vs fungal richness and soil multifunctionality.

Response: Thank you for these very useful suggestions. As mentioned earlier, we have confirmed and provided stronger support for the existence of a threshold around 0.8 aridity level by using the approaches you recommended (see the new Fig. 3; see also the response to the third main concern proposed by you above).

L203-205: This refers to the stronger effects of plant richness under moderately arid conditions (i.e., plant richness x aridity interaction), not to the positive effects of plant richness on soil functioning per se.

Response: Thanks. The sentence now reads (Lines 271-275):

“The consistent and positive relationship between plant species richness and soil multifunctionality across the aridity gradient aligned with previous reports^{11–13,22,26} and can be explained by increased litter inputs into the soil due to increased net primary production or complementarity resource use among species^{45,46}. As predicted, the positive relationship between plant diversity and soil multifunctionality became weaker from less towards more arid regions.”

¹¹Hector, A. & Bagchi, R. Biodiversity and ecosystem multifunctionality. *Nature* **448**, 188–190 (2007).

¹²Isbell, F. et al. High plant diversity is needed to maintain ecosystem services. *Nature* **477**, 199–202 (2011).

¹³Maestre, F. T. et al. Plant species richness and ecosystem multifunctionality in global drylands. *Science* **335**, 214–218 (2012).

²⁶Eisenhauer, N. et al. Plant diversity maintains multiple soil functions in future environments. *eLife* **7**, e41228 (2018).

²²Jing, X. et al. The links between ecosystem multifunctionality and above- and belowground biodiversity are mediated by climate. *Nat. Commun.* **6**, 8159 (2015).

⁴⁵Hooper, D. U. & Vitousek, P. M. Effects of plant composition and diversity on nutrient cycling. *Ecol. Monogr.* **68**, 121–149 (1998).

⁴⁶Cardinale, B. J. et al. The functional role of producer diversity in ecosystems. *Am. J. Bot.* **98**, 572–592 (2011).

L208-209: remove “from plant production and 208 declining influence on soil functions related to nutrient cycling and soil fertility”, please. I think it will read much clearer.

Response: Thank you for the suggestion. The sentence has been amended accordingly (Line 278).

L253: add “true” between “especially” and “for”

Response: Thanks. The word has been added accordingly (Line 328).

L283-284: This is a complicated way to explain that aridity index (AI) is counterintuitive in the sense that larger values mean wetter, not more arid, and that therefore you calculated the inverse (aridity level, as in the reference mentioned).

Response: We thank you for highlighting this point. The sentence now reads (Lines 382-385):

“The aridity level of each site was calculated as $1 - \text{aridity index (AI)}$, where AI is the ratio of precipitation to potential evapotranspiration³⁸. We obtained AI from the Global Aridity Index and Potential Evapotranspiration Climate database (<https://cgiarcsi.community/>).”

³⁸Delgado-Baquerizo, M. et al. Decoupling of soil nutrient cycles as a function of aridity in global drylands. *Nature* **502**, 672–676 (2013).

L284-286: And how was this aim achieved exactly, or how did you selected the site to achieve this aim?

Response: We thank you for noticing this missing information. We have now added the information. The sentence now reads (Lines 385-391):

“The selection of the field sites aimed to minimize the potential impacts of human activity and other disturbances on soil, vegetation and geomorphological characteristics based on the following three criteria: (i) sites were at least 1 km away from major roads and > 50 km from human habitations; (ii) sites were under pristine or unmanaged conditions without visible signs of domestic animal grazing, grass/wood collection, engineering restoration plantings, and infrastructure construction; and (iii) the soil was dry without experiencing rainfall events for at least three days prior to sampling.”

L287-289: Are these different soil and vegetation types considered somehow in the analyses? They could confound part of the findings reported here

Response: We thank you for highlighting this issue. Indeed, considering this point in the analyses made our results more robust. To account for the similarities of soil and vegetation types among study sites, we have now replaced the original general linear model with a linear mixed-effects model (including “Soil type” and “Vegetation type” as random terms) throughout the revised manuscript (see the revised Table 1 and

Supplementary Table 4,6,7).

L296 This is a very large area and sampling effort to measure plant cover and richness, did you measured it visually walking around these 900 m² quadrats, or did you apply the point-intercept method in there somehow? I guess the different sites are represented by pH and clay content, but is this enough?

By the way, some of the predictors included in the linear models (Table 1) are very highly correlated with each other (Suppl. Figure 13). This could induce multicollinearity in the estimation of effect sizes, did the authors tested for this using VIF?

Response: We thank you for highlighting these important issues.

- Actually, we employed the belt transect method (Grant et al. 2004) to measure plant cover and species richness. Briefly, we established four 1.5 × 30 m parallel transects (spaced 8 m apart) within each 30 × 30 m quadrats at each site. The investigator walks steadily along each transect while visually estimating vegetation cover and listing all plant species. The process is rapid because the vegetation types often are similar along each transect. The investigator must only note the appearance of new species. This information is now provided in Lines 405-409 (Methods).

Grant, T. A., Madden, E. M., Murphy, R. K., Smith, K. A. & Nenneman, M. P. Monitoring native prairie vegetation: the belt transect method. *Ecol. Restor.* **22**, 106–111 (2004).

- We agree and acknowledge that some unmeasured and perhaps important soil variables were not included in our analyses. However, we also admit that it is typically not realistic to consider all possible variables. We chose soil pH and clay content in our statistical analyses because these two soil variables are among the most important drivers of structure and functioning in dryland ecosystems. For instance, soil pH is a major driver of plant and soil microbial diversity in drylands (e.g., Maestre et al. 2015; Palpurina et al. 2017). Also, soil pH has been reported to strongly influence ecosystem multifunctionality in global drylands (Delgado-Baquerizo et al. 2016). Similarly, the content of clay has been found to play key roles in controlling water availability, community structure and

biogeochemical processes in drylands (Maestre et al. 2012; Delgado-Baquerizo et al. 2013). Specifically, clay has an important role on the soil water retention and nutrients/fertility, thereby influencing plant and soil microbial diversity and biomass, and can also modify local soil pH.

Maestre, F. T. et al. Increasing aridity reduces soil microbial diversity and abundance in global drylands. *Proc. Natl Acad. Sci. USA* **112**, 15684–15689 (2015).

Palpurina, S. et al. The relationship between plant species richness and soil pH vanishes with increasing aridity across Eurasian dry grasslands. *Global Ecol. Biogeogr.* **26**, 425–434 (2017).

Delgado-Baquerizo, M. et al. Microbial diversity drives multifunctionality in terrestrial ecosystems. *Nat. Commun.* **7**, 10541 (2016).

Maestre, F. T. et al. Plant species richness and ecosystem multifunctionality in global drylands. *Science* **335**, 214–218 (2012).

Delgado-Baquerizo, M. et al. Decoupling of soil nutrient cycles as a function of aridity in global drylands. *Nature* **502**, 672–676 (2013).

- A test for potential multicollinearity using VIF has now been added into the tables of all linear mixed-effects models (please see the revised Table 1 and Supplementary Tables 4,6,7). Accordingly, this information has further been highlighted in Methods as follows (Lines 748-749):

“We used a variance inflation factor (VIF) to evaluate the risk of multicollinearity, and selected variables with $VIF < 10$ in all cases⁹⁴.”

⁹⁴Hair, J. F., Black, W. C., Babin, B. J., Anderson, R. E. & Tatham, R. L. *Multivariate Data Analysis* (Prentice Hall, New Jersey, 1998).

L308-309: You need to state here which soil samples are used to calculate multifunctionality. Information on L433-436 should be here.

Response: We thank you for the suggestion. The sentence now reads (Lines 441-445): “Collectively, 6–21 soil samples per site were collected, and in total 864 samples were taken and analyzed for each of the seven individual soil functions (see below) and multifunctionality. All soil functions evaluated in the field study were calculated at site level by using a weighted average of the mean values observed in vegetated areas and bare ground by their respective cover^{13,14,38}.”

¹³Maestre, F. T. et al. Plant species richness and ecosystem multifunctionality in global drylands. *Science* **335**, 214–218 (2012).

¹⁴Delgado-Baquerizo, M. et al. Microbial diversity drives multifunctionality in terrestrial ecosystems. *Nat. Commun.* **7**, 10541 (2016).

³⁸Delgado-Baquerizo, M. et al. Decoupling of soil nutrient cycles as a function of aridity in global drylands. *Nature* **502**, 672–676 (2013).

L328: Do you think that the stronger importance of soil fungi for functioning when soil was fairly dry in the microcosm experiment is related to the fact that soils came from a very arid site? Could the authors compare with other microcosm experiments using different soils from contrasting environmental conditions from the published literature? Most importantly, the microcosm experiment gives the false impression that you are able to causally link microbial diversity to soil functioning, whereas what you are measuring with this microcosm experiment, as in the observational study, is that soil diversity and nutrients co-vary when water availability changes. This should be clearly mentioned in the text.

Response: We thank you for highlighting these issues.

- Despite the fact that only one soil sample was used in the microcosm experiment and it was collected from a relatively arid site (i.e., ~ 0.8 aridity level), we do not believe that these issues are significantly affecting the general trend of the observed relationship between soil microbial diversity and multifunctionality. The legacy effects of aridity on soil microbial communities, process rates and multifunctionality should be identical across different treatments of soil water availability. In our microcosm experiment, therefore, the manipulated moisture content is the only determinant of any observed changes and irrelevant to the background of soils used.
- Furthermore, we agree on the point that our microcosm experiment does not establish causality between soil microbial diversity and multifunctionality. As mentioned earlier, the design and contribution of the microcosm experiment along with the caveats mentioned by you and other reviewers have now further been highlighted in the Introduction (Lines 133-141), Results (Lines 236-239) and Discussion (Lines 263-267) for clarity.

L472: You do not need to log-transform anything to obtain Z-scores.

Response: We thank you for the comment. We are sorry that the previous description is not clear and confusing you. The sentence should read (Lines 613-617):

“To obtain a quantitative multifunctionality index for each field site or experimental microcosm, we first normalized (log₁₀-transformed when needed) and standardized each of the evaluated soil functions using the Z-score transformation. The Z-scores of the soil functions were then averaged to obtain a multifunctionality index for each field site or experimental microcosm¹³.”

¹³Maestre, F. T. et al. Plant species richness and ecosystem multifunctionality in global drylands. *Science* **335**, 214–218 (2012).

507-509: I don't really understand the link between this map and the objectives to be tested, so I suggest removing it. The 100+ sites are more than enough to evaluate relationships (not effects) between biodiversity, aridity and soil nutrients. If the authors insist in keeping it, I would like to see enough information to understand what exactly they did, or if it was any cross-validation applied to test how valid is this interpolation.

Response: We thank you for the comment and suggestion. The interpolated map indeed is not very relevant to the objectives of our study. As mentioned earlier, the interpolated map has now been replaced with a distribution map of field site in the revised manuscript (please see the revised Fig. 2a). Furthermore, we have also toned down the language and now only speak of “relationship between” (or similar) throughout the revised manuscript (please see also the response to the main concern of Reviewer 1).

L514-517: I don't really understand this analysis or how it complement the other ones. Could the authors develop more explanation to this, please? I think that the methods would be clearer if the Stats started with the main analysis (that on L518 onwards).

Response: We thank you for highlighting these issues. Actually, the original Fig. 2e, which is based on the multiple threshold approach, describes the response of the threshold (a certain percentage within a 1 to 99% range) of the maximum observed value that each soil function achieves to increasing aridity. Our original intention was to present the nonlinear response of such threshold to increasing aridity for each of

individual soil functions measured.

However, because the threshold of the maximum observed value that each soil function achieves at a certain aridity level is proportional to the observed value of each function, we argue that the nonlinear responses determined by using the threshold should be very similar with those determined by using the observed value per se. To consistently employ the averaging multifunctionality index throughout the manuscript (as you suggested above) and make this analysis more intuitive and clearer, we now decide to remove the original Fig. 2e.

Instead, we now report the nonlinear responses of individual soil functions and multifunctionality to increasing aridity by fitting either quadratic or GAM regressions, on top of which we have further determined the aridity thresholds at which those responses showed drastic or abrupt changes using the standard procedure for the detection of ecological thresholds recommended by you (please see the revised Fig. 2b-i). We believe that our new results are more straightforward and informative, and allow us to address a key scientific question that whether the negative effects of aridity are stronger when multiple soil functions are considered simultaneously or whether soil multifunctionality is more susceptible to increasing aridity than are any individual soil functions. This information has now been highlighted in Results (Lines 145-154), Discussion (Lines 254-257) and Methods (Lines 671-682).

L534-535: Indicate that the effects of plant and soil microbial richness were tested simultaneously, please, this is one of the main points of the study. Did the authors included the 3- way interaction between plant richness x microbial richness x aridity?

Response: We thank you for highlighting this important point. The linear mixed-effects model used for the moving-window analysis has now been revised as follows (Lines 710-712):

“Soil multifunctionality \sim Year + Plant species richness + Soil microbial diversity index + Aridity + Aridity \times Plant species richness + Aridity \times Soil microbial diversity index + Aridity \times Plant species richness \times Soil microbial diversity index + (1|Soil type) + (1|Vegetation type)”

where we eliminated variation due to different sampling years by first entering the term “Year” into the model according to the comments by Reviewer 3. Based on your

suggestions, we included soil and vegetation types as random terms. Furthermore, plant species richness and soil microbial diversity were tested simultaneously and the 3- way interaction term was also included in the model.

L542: 0.5-0.8 is not equal to 0.8. Why did the authors divided their data according to the 0.8 level?

Response: We thank you for pointing out this issue. In the previous version of the manuscript, selecting the 0.8 aridity level to divide our dataset was indeed a bit of arbitrary. In the revised version, we have now provided stronger support for the existence of the 0.8 aridity threshold (please see the response to your third main concern and also the new Fig. 3a). Therefore, the reason that we use the 0.8 aridity level to split our data is now more solid. This information has now further been highlighted early on in the Results (Lines 192-196).

Figure 2: I would keep the map showing the sampling sites, panel b and panel e. I think the remaining contents are more confusing than anything else.

Response: Thank you for these suggestions. As mentioned earlier, we have now removed the interpolated map and moved the distribution map of sampling sites to the main text (please see the revised Fig. 2a). To be more straightforward and informative, panel b and panel e have also been replaced (please see the revised Fig. 2b-i; see also the response to L514-517 above).

Fig. 4: Change to “Less arid” and “More arid” regions, please. Or even better, just state “sites with $AI \leq 0.8$ or ≥ 0.8 ”, so the reader knows exactly what you mean.

Response: We thank you for the suggestion. We have now changed the expressions to “sites with aridity < 0.8 or > 0.8 ” in both the revised Figs. 4 and 5.

Replies to Reviewer 3

Reviewer #3 (Remarks to the Author):

We thank Reviewer 3 for the inspiring suggestions to improve the manuscript. The most important changes to the manuscript in relation to the comments by Reviewer 3 are the following:

- We have checked the robustness of our findings to the exclusion of total soil phosphorus.
- We have toned down the language and now only speak of “relationships between biodiversity and soil multifunctionality” (or similar) instead of “the effects of biodiversity on multifunctionality” throughout the revised manuscript.
- The reasonability of integrating all data obtained at different sampling times has now further been explained.

This manuscript described results from an ambitious effort that explored how environmental factors (in this case, aridity) affect biodiversity-soil multifunctionality relationship. The field experiment was very extensive, including 130 sites along a 4000-km gradient of aridity across Northern China. The research is original. The major result was that plant species richness was the dominant driver of soil multifunctionality in less arid regions, whereas microbial diversity (fungi in particular) played a greater role in more arid regions. It also identified the threshold aridity (0.8) for the shift in the relative role of plant and microbial diversity for soil multifunctionality. These results are interesting and deserve publication. The manuscript was in general well written. Still, I have a few concerns that may need to be addressed before the manuscript can be considered for publication.

Response: We thank you for your appreciation of the large-scale field work, and your positive comment on the originality and importance of the findings reported in our manuscript.

First, the logic that seven nutrient stocks in soil were selected to represent the soil multifunctionality is not very clear. In other words, the justification was not convincing. This is not trivial because the number and type of ‘functions’ selected likely affect the relationships between biodiversity and multifunctionality. The

manuscript attempted to assert that biodiversity “drives” the multifunctionality (i.e., an integrated index of different labile and total C, N and P pools). However, this is debatable, at least. For example, it argued that plant and microbial diversity controls soil organic C. Yet, one can also argue that the opposite is right: soil organic C may dominate microbial diversity and to a lesser degree, plant diversity, as soil organic C dominates soil fertility and soil structure (niche availability for microbes). Moreover, taking total soil P as a soil biodiversity-driven function is high questionable. Soil mineral composition (parental materials) and climate, rather than biological factors, are often believed to be the dominant determinants of total soil P.

Response: We thank you for highlighting these important issues.

- As you stressed, it indeed is problematic to imply causality using correlative data. According to the comments by you and other reviewers, we have toned down the language to avoid distinguishing direction between biodiversity and soil multifunctionality (for example, “drives” or similar) and now only speak of “correlated with” (or similar) throughout the manuscript.
- We fully agree that the number and identity of “functions” selected may affect the observed relationships between biodiversity and soil multifunctionality. We also thank you for pointing out the fact that total soil phosphorus is mostly driven by abiotic rather than biotic processes. To address these issues, we removed total soil phosphorus and nitrogen (highlighted by Reviewer 2 and the editor, because total soil nitrogen is highly related to several individual soil functions and therefore may provide redundant information when assessing soil multifunctionality) and then repeated all analyses (please see Supplementary Figs. 19–28 and Supplementary Tables 2,3,6–8). We found that removing these two functions did not change our results in any way and thereby confirmed the robustness of our analyses. Given that soil phosphorus is, to a lesser extent, derived from the biological decomposition of organic matter, we therefore decide to keep total soil phosphorus in our analyses of the main text. Also, this information has now further been highlighted in the Results (Lines 222-227) and Methods (Lines 773-780) as follows:

Lines 222-227: “To address the potential redundancy between total soil N and other individual soil functions (Supplementary Fig. 18a), and the fact that total soil P is more closely related to abiotic rather than biotic processes, we removed these two soil functions and then repeated the above analyses. Consistent results

were found for the simplified version of soil multifunctionality including five soil functions (i.e., simplified soil multifunctionality) (Supplementary Figs. 19–28 and Supplementary Tables 2,3,6–8).”

Line 773-780: “We must note, however, that total soil P is typically considered to be controlled mostly by abiotic processes such as weathering of rocks rather than biotic processes in dryland ecosystems³⁸. Also, total soil N was strongly correlated with DNA concentration and organic C in our dataset (Supplementary Fig. 18a), which may provide redundant information for the multifunctionality metrics used^{3,92}. To ensure that these were not influencing our results, we repeated above analyses after excluding total soil N and P. These two sets of analyses provided very similar results (Supplementary Figs. 19–28 and Supplementary Tables 2,3,6–8), thus these issues do not affect the conclusions of this study.”

³⁸Delgado-Baquerizo, M. et al. Decoupling of soil nutrient cycles as a function of aridity in global drylands. *Nature* **502**, 672–676 (2013).

³Manning, P. et al. Redefining ecosystem multifunctionality. *Nat. Ecol. Evol.* **2**, 427–436 (2018)

⁹²Gamfeldt, L., Hillebrand, H. & Jonsson, P. R. Multiple functions increase the importance of biodiversity for overall ecosystem functioning. *Ecology* **89**, 1223–1231 (2008).

Second, the linkages between microbial diversity at the DNA level and soil functions or the pool size of soil C, N and P have yet well established, although we know soil microbes (but not molecular diversity per se) exert major controls on soil C and N. Therefore, the relationships between diversity (both plant and microbial) and the most functions included in the multifunctionality index in the manuscript are largely correlational, not cause-effect ones.

Response: We appreciate your opinions and agree on this point. As discussed above, we now try to use more cautious wording and now only speak of “relationships between biodiversity and soil multifunctionality” (or similar) rather than “the effects of biodiversity on multifunctionality” throughout the revised manuscript.

Third, a study at the scale described in this manuscript is inherent with some issues with the timing and the scheme of sampling: Because of the extensive nature of field

sampling, plants and soils were collected at different times: in different years and/or at different growing stages of plants when temperature and rainfall may also be significantly different. Some of soil functions as defined or selected in this manuscript, e.g., soil NO₃, are very sensitive to these temporal changes: plant uptake of NO₃ may quickly depletes soil NO₃ during the fast plant-growing season, but has little impact when plants are tiny. Similarly, a rainfall event may rapidly alter microbial transformations of soil N, or the microbial DNA concentration itself. It is unclear how field sampling was designed to deal with these issues. Also, the manuscript needs to better explain why integrating all the data obtained at different sampling times together is reasonable.

Response: We thank you for highlighting these important points. While we recognize the issues related to different sampling times and years, it was not possible to complete such extensive field sampling efforts within a single year. Nevertheless, we attempted to minimize the potential impact of these issues in three ways. First, in each year we conducted field sampling between June and September (during the peak-growing season), when approximately 77–98% of the precipitation occurs across our survey areas corresponding to the highest biological activity and productivity. Second, we tried to exclude potential effects of recent rainfall events on microorganisms and certain soil functions. For example, the selection of sites for the field study was based on the criterion that the soil was dry and without rainfall for at least three days prior to sampling. In light of this, we expect that the effects of different sampling times and years or seasonality should be minimal. Third, we accounted for the potential effects of different sampling years in our statistical analyses. To do so, we eliminated the variation in soil multifunctionality due to different sampling years by first entering the term “Year” into the all revised linear mixed-effects models (please see the new Fig. 3 and the revised Table 1 and Supplementary Table 4). Our new results are fully compatible with the previous ones, suggesting that integrating all the data obtained at different sampling years together is reasonable. Collectively, this information has now further been highlighted in the Methods as follows (Lines 390-391, 423-426, 439-441, and 700-701):

Lines 390-391: “and (iii) the soil was dry without experiencing rainfall events for at least three days prior to sampling.”

Line 423-426: “Across our survey areas, approximately 77–98% of the precipitation occurs between June and September (during the peak-growing season) corresponding to the period of the highest plant above- and belowground biomass^{34,35,41,69}.”

Line 439-441: “All vegetation and soil surveys were carried out during the wet season (June to September) when biological activity and productivity are maximal; as such, we do not expect the different sampling times and years or seasonality to be a major factor influencing our conclusions.”

Lines 700-701: “.....and eliminated variation due to different sampling years by first entering the term “Year” into the statistical model⁴¹.”

³⁴Deng, J. M. et al. Plant mass–density relationship along a moisture gradient in north-west China. *J. Ecol.* **94**, 953–958 (2006).

³⁵Chen, R. F. et al. Life history strategies drive size-dependent biomass allocation patterns of dryland ephemerals and shrubs. *Ecosphere* **10**, e02709 (2019).

⁴¹Ma, W. H. et al. Environmental factors covary with plant diversity–productivity relationships among Chinese grassland sites. *Global Ecol. Biogeogr.* **19**, 233–243 (2010).

⁶⁹Wang, S., Wang, X. B., Han, X. G. & Deng, Y. Higher precipitation strengthens the microbial interactions in semi-arid grassland soils. *Global Ecol. Biogeogr.* **27**, 570–580 (2018).

Minor comments

1. Sampling site selection: it is unclear how to select the soil type at each sampling site if multiple soil types exist at the location.

Response: We thank you for highlighting this missing information. Actually, the field sampling sites were selected based on a key criterion, that is, sites represented the local flora and soil types covering an area of no less than 10,000 m². This point has now been highlighted in Lines 405-406 (Methods) as follows:

“In brief, three 30 × 30 m quadrats were established at each site to represent the local vegetation and soil types that covered an area of no less than 10,000 m².”

2. It is unclear why the microcosm experiment had high moisture treatments at 100% and 120% of field water holding capacity. I would think that with annual rainfall ranging from 21 to 453 mm, the soil moisture in any field site rarely reaches the field capacity, even just for a couple days. But the incubation experiment lasted for 30 days.

Did results from these unrealistic high-water availability significantly skew the general trend of the relationship?

Response: We thank you for noticing this issue. Actually, the moisture treatments in our microcosm experiment were designed to match with differences in moisture conditions among the field sites. And, the moisture contents corresponding to 100% and 120% of field water holding capacity are within the scope of moisture conditions among the field sites (please see the revised Supplementary Fig. 1). Therefore, we believe that the manipulated moisture treatments at 100% and 120% of field holding capacity are reasonable and can reflect the realistic conditions of soil water availability in the field. However, and to further address whether the high moisture treatments at 100% and 120% field holding capacity significantly affect the general trend of the relationship between soil microbial diversity and multifunctionality, we also checked the robustness of our analyses to the exclusion of these two high moisture treatments. We found that removing these two moisture treatments did not affect our conclusions in any way (please see Figures 3 and 4 below). Therefore, we still include the moisture treatments at 100% and 120% field holding capacity in our analyses of the revised manuscript (please see Supplementary Figs. 30 and 31). Furthermore, the design of the microcosm experiment has now further been highlighted early on in the Introduction as follows (Lines 133-136):

“In addition to the field study, we manipulated soil water availability in a microcosm experiment to experimentally test for linkages between moisture content, microbial diversity and soil multifunctionality by simulating differences in moisture conditions among our field sites (Supplementary Fig. 1; see also Methods for more details of experimental design).”

Figure 3 Relationships between moisture content and soil archaeal richness (a), soil bacterial richness (b), soil fungal richness (c), and the soil microbial diversity index (d). These results are shown for the microcosm experiment without including the moisture treatments at 100% and 120% field holding capacity. The red lines represent the fitted linear or quadratic OLS model. Model choice was based on AIC value. Differences in AIC (Δ AIC) values > 2 indicate that the models are different. Linear model was chosen when the Δ AIC values between linear and quadratic models were < 2 . Solid and dashed lines denote statistically significant ($P \leq 0.05$) and non-significant ($P > 0.05$) relationships, respectively. Dots represent means \pm SE ($N = 3$).

Figure 4 Relationships between moisture content, microbial diversity, and soil multifunctionality shown for the microcosm experiment without including the moisture treatments at 100% and 120% field holding capacity. a Bivariate correlation between moisture content and soil multifunctionality. The red line represents the fitted linear OLS model. Rest of legend as in Figure 3. b Relationship between soil bacterial richness and multifunctionality at high (40-80% field capacity; $N = 9$) and low (3-20% field capacity; $N = 15$) moisture levels, as well as across all experimental microcosms ($N = 24$; the black line). Lines represent the fitted linear OLS model. Rest of legend as in Figure 3. c,d SEMs accounting for the hypothesized direct and indirect relationships between moisture content, soil microbial diversity and multifunctionality at high (c; 40-80% field capacity; $N = 9$) and low (d; 3-20% field capacity; $N = 15$) moisture levels. Black and gray arrows denote significant and non-significant relationships, respectively. Continuous and dashed arrows indicate positive and negative relationships, respectively. The thickness of the arrow is proportional to the magnitude of standardized path coefficients and indicative of the strength of the relationship. Asterisks indicate the significant level of each coefficient: $*P < 0.05$; $***P < 0.001$. R^2 is the proportion of variance explained by the model. Goodness-of-fit statistics for each SEM are given (d.o.f., degrees of freedom; RMSEA, root mean squared error of approximation).

3. DNA extraction: Was relic DNA removed or considered? Did soils at the super dry sites have higher proportion of relic DNA?

Response: We thank you for highlighting this issue. The DNA extraction method used indeed did not distinguish the active fraction from the total microbial communities. However, it is possible that the active microbial community differs from the total community and is more important for soil multifunctionality. Despite this caveat, we expect that the effects of relic DNA on our results should be minimal, because our study system (i.e., drylands) is characterized by relatively high summer temperatures (although also relatively low water availability) which is the most important factor that limits the long-term preservation of relic DNA. According to your comments here, this potential limitation has now further been highlighted in the Discussion (Lines 353-357) as follows:

“Furthermore, the DNA extraction method used to characterize soil microbial diversity focuses on the total microbial communities and fail to discriminate its active fraction, which may be related more closely to soil multifunctionality⁶¹. Despite this caveat, we expect the effects of the DNA extraction method to be minor because the studied dryland regions are characterized by high summer temperatures that fasten the degradation of relic DNA^{62,63}.”

⁶¹Bastida, F. et al. The active microbial diversity drives ecosystem multifunctionality and is physiologically related to carbon availability in Mediterranean semi-arid soils. *Mol. Ecol.* **25**, 4660–4673 (2016).

⁶²Gong, H. Y. et al. Soil microbial DNA concentration is a powerful indicator for estimating soil microbial biomass C and N across arid and semi-arid regions in northern China. *Appl. Soil Ecol.* **160**, 103869 (2021).

⁶³Willerslev, E., Hansen, A. J. & Poinar, H. N. Isolation of nucleic acids and cultures from fossil ice and permafrost. *Trends Ecol. Evol.* **19**, 141–147 (2004).

4. The enzyme issue: Lns:437-441. The notion that soil enzyme activities and CO₂ fluxes in the incubation experiment “measure process rates directly and reflect the fluxes of matter and energy” is highly debatable. Linking soil enzyme activities with soil functions is always tricky. Most enzyme activities were estimated as the potential activities under the optimal conditions and often explain only a small proportion of variability of functions.

Response: We thank you for these comments and agree on the issue related the direct

and strong linkage between soil enzyme activities and process rates. Based on the comments by you, we have now revised the sentence to avoid such strong expression. The sentence now reads (Lines 580-584):

“Extracellular enzymes such as those we measured are produced by soil microorganisms and the proximate agents of processes such as the stabilization and destabilization of soil organic matter, and measures of their activities are also considered a good indicator of soil fertility and microbial nutrient demand. In addition, respiration can be used as a proxy of soil microbial activity.”

We hope that you find our revision satisfactory. Thank you very much!

Reviewer comments, second round –

Reviewer #1 (Remarks to the Author):

The authors have addressed the majority of my concerns, although they are not able to include data from a biodiversity manipulation experiment, which are not readily available. Although I still think these data would have added to the paper, it is overall strengthened in both the rationale and interpretation of data, which were my main concerns.

Reviewer #2 (Remarks to the Author):

The authors have done an excellent job in addressing my previous concerns. I think this is a much more solid and more clearly written work. I only have a few minor concerns before recommending this manuscript for publication.

L46-50 This is a very convoluted couple of sentences. I would suggest rewriting to "Ecosystem functioning (fluxes of matter and energy between trophic levels, and nutrient stocks and transformation rates) are affected by the collective activities of producer, consumer and decomposer communities¹⁻⁴." I would also remove L50-51, it doesn't add too much.

L79-81. This sentence says nothing about environmental gradients, which is the topic within this paragraph. Remove or rephrase to accommodate it to the main line of argumentation, please. Same with sentence in L81-83, the relationships between these negative effects of plant richness on functioning and environmental gradients is unclear

L85-86 The negative effect of soil bacterial and fungal richness on the functioning of semiarid grassland seem to directly contradict your initial hypothesis, doesn't it? Could the authors elaborate a little bit more this part?

L154-161. I found these results rather confusing. Since these are largely supporting those of the averaging approach, I would just say that "These results were consistent when using the multiple-threshold approach to calculate multifunctionality (Supplementary Fig. 4 and Supplementary Table 3)".

L210-211 rewrite to "As a complement to the analyses of bivariate correlations (Fig. 4; Supplementary Table 4), we used structural equation models (SEMs) to..."

L278-279 Your data does not provide any evidence of aridity increasing competition between plants. Indeed, several studies on this topic shows rather the opposite.

Mean annual precipitation of the study site from which the microcosm soils comes from is different in L468 (7.9 °C) and in L473 (18.5 °C), clarify this, please.

L475 Field capacity, by definition, is that maximum amount of water that a given soil can hold. Therefore, it seems impossible to me to maintain a soil in > 100% of its field capacity for over a month.

Fig. 5 Goodness-of-fit tests for the SEM of high aridity sites are not great. Chi² is borderline, despite having only 1 degree of freedom, Bollen-Stine is non-significant, and RMSEA's value is well above 0.05. This is not too dramatic, considering that is one amongst the many tests performed, and that it is just to assess indirect vs direct effects, but should be acknowledged in the main text and revised to improve model fit if possible.

Fig 6 (and related main text in L232-233), I do not understand what the authors mean by "shrinking areas" in this context, are those areas currently with AI > 0.8 that will have a lower AI

in the future? If so, why is that these are included within the first category in the maps (green + grey)? Clarification needed here

Reviewer #3 (Remarks to the Author):

I reviewed the previous version of this manuscript. The authors addressed all of my concerns in this revised version. Indeed, additional analyses by excluding total N and total P have strengthened the major argument of the manuscript. Changing "effects of diversity on ecosystem functions" to "relationships between diversity and ecosystem functions" across the manuscript has better reflected what the dataset and the experimental designs can substantiate. I feel that this manuscript is now acceptable for publication. Congrats to the authors for this very nice piece of work.

Shuijin Hu

P.S.

Ln. 34: I would feel more comfortable to replace "expected" with "predicted"

Ln. 347: considered "as" indicators...

Responses to Reviewers

Replies to Reviewer 1

Reviewer #1 (Remarks to the Author):

The authors have addressed the majority of my concerns, although they are not able to include data from a biodiversity manipulation experiment, which are not readily available. Although I still think these data would have added to the paper, it is overall strengthened in both the rationale and interpretation of data, which were my main concerns.

Response: We thank the reviewer for his/her positive feedback on our revisions and for the understanding of being unable to include a biodiversity manipulation experiment in this manuscript. However, we really appreciate the reviewer's great suggestion and intend to perform such experiment in a separate study in the future.

Replies to Reviewer 2

Reviewer #2 (Remarks to the Author):

The authors have done an excellent job in addressing my previous concerns. I think this is a much more solid and more clearly written work. I only have a few minor concerns before recommending this manuscript for publication.

Response: We thank the reviewer for his/her very thorough review of our revised manuscript again. The additional comments and suggestions helped us a lot to further improve the manuscript and we really appreciate the positive feedback on our revisions.

L46-50 This is a very convoluted couple of sentences. I would suggest rewriting to “Ecosystem functioning (fluxes of matter and energy between trophic levels, and nutrient stocks and transformation rates) are affected by the collective activities of producer, consumer and decomposer communities¹⁻⁴.” I would also remove L50-51, it doesn’t add too much.

Response: Thank you for the suggestions.

- The sentence has been revised accordingly for clarity, and it now reads (Lines 46-48):

“Ecosystem functioning (fluxes of matter and energy between trophic levels, and nutrient stocks and transformation rates) are affected by the collective activities of producer, consumer and decomposer communities¹⁻⁴.”

¹Srivastava, D. S. & Vellend, M. Biodiversity–ecosystem function research: is it relevant to conservation? *Annu. Rev. Ecol. Evol. Syst.* **36**, 267–294 (2005).

²Liu, X. J. et al. Tree species richness increases ecosystem carbon storage in subtropical forests. *Proc. R. Soc. B.* **285**, 20181240 (2018).

³Manning, P. et al. Redefining ecosystem multifunctionality. *Nat. Ecol. Evol.* **2**, 427–436 (2018).

⁴Garland, G. et al. A closer look at the functions behind ecosystem multifunctionality: a review. *J. Ecol.* **109**, 600–613 (2021).

- Furthermore, the latter sentence has been removed accordingly.

L79-81. This sentence says nothing about environmental gradients, which is the topic

within this paragraph. Remove or rephrase to accommodate it to the main line of argumentation, please. Same with sentence in L81-83, the relationships between these negative effects of plant richness on functioning and environmental gradients is unclear.

Response: We thank you for pointing out this vague expression. Our intention is to say that different relationship patterns between plant species richness and ecosystem multifunctionality have been reported in several previous studies, which could be attributed to the context-dependency of biodiversity–ecosystem multifunctionality relationships. However, these sentences indeed neglect to highlight the topic about environmental context. To further clarify, we have revised the sentences as (Lines 76-81):

“For instance, plant species richness has been shown to enhance ecosystem multifunctionality in small-scale plant diversity-manipulation experiments^{11,12,16,26} and large-scale studies of dryland and grassland ecosystems across different environmental conditions^{13,22}. Furthermore, a meta-analysis of 94 experimental manipulations of plant species richness across aquatic and terrestrial habitats revealed that plant diversity sometimes has negative effects on ecosystem multifunctionality²⁷.”

¹¹Hector, A. & Bagchi, R. Biodiversity and ecosystem multifunctionality. *Nature* **448**, 188–190 (2007).

¹²Isbell, F. et al. High plant diversity is needed to maintain ecosystem services. *Nature* **477**, 199–202 (2011).

¹⁶Fanin, N. et al. Consistent effects of biodiversity loss on multifunctionality across contrasting ecosystems. *Nat. Ecol. Evol.* **2**, 269–278 (2018).

²⁶Eisenhauer, N. et al. Plant diversity maintains multiple soil functions in future environments. *eLife* **7**, e41228 (2018).

¹³Maestre, F. T. et al. Plant species richness and ecosystem multifunctionality in global drylands. *Science* **335**, 214–218 (2012).

²²Jing, X. et al. The links between ecosystem multifunctionality and above- and belowground biodiversity are mediated by climate. *Nat. Commun.* **6**, 8159 (2015).

²⁷Lefcheck, J. S. et al. Biodiversity enhances ecosystem multifunctionality across trophic levels and habitats. *Nat. Commun.* **6**, 6936 (2015).

L85-86 The negative effect of soil bacterial and fungal richness on the functioning of

semiarid grassland seem to directly contradict your initial hypothesis, doesn't it? Could the authors elaborate a little bit more this part?

Response: We thank you for highlighting this point.

- We checked the two studies cited here, which reported negative effects of soil bacterial and saprophytic fungal richness on ecosystem multifunctionality in semiarid grassland and subtropical forest, respectively. To be more accurate, the sentence has now been revised as (Lines 83-84):

“..... negative effects of soil bacterial and saprophytic fungal richness have been reported in semiarid grassland and subtropical forest, respectively^{17,23},.....”

¹⁷Schuldt, A. et al. Biodiversity across trophic levels drives multifunctionality in highly diverse forests. *Nat. Commun.* **9**, 2989 (2018).

²³Wang, X. Y. et al. High ecosystem multifunctionality under moderate grazing is associated with high plant but low bacterial diversity in a semi-arid steppe grassland. *Plant Soil* **448**, 265–276 (2020).

- In light of the main topic within this paragraph (Lines 74-95), we do not believe that these results directly contradict our initial hypothesis (Lines 96-103). First, the main line of argumentation within this paragraph is the context-dependency of the relationships between plant or soil microbial diversity and ecosystem multifunctionality, which implies that biodiversity–ecosystem multifunctionality relationships may vary across different environmental conditions or ecosystem types. Thus, it is reasonable that different relationship patterns between soil microbial diversity and ecosystem multifunctionality were reported in different studies. Second, we consider the linkages among plant species richness, microbial diversity and soil multifunctionality in this manuscript, and hypothesize that the relative strength of the relationships between plant or microbial diversity and soil multifunctionality may shift along an aridity gradient. Thus, the main topic of this manuscript is very different from those of the two studies cited here. Finally, our results show a negative relationship between soil bacterial richness and multifunctionality in less arid regions (i.e., semiarid and dry-subhumid regions; please see also Fig. 4d), which is similar with the reported negative effect of soil bacterial richness on ecosystem multifunctionality in semiarid grassland (Wang et al. 2020).

Wang, X. Y. et al. High ecosystem multifunctionality under moderate grazing is associated with high plant but low bacterial diversity in a semi-arid steppe grassland. *Plant Soil* **448**, 265–276 (2020).

L154-161. I found these results rather confusing. Since these are largely supporting those of the averaging approach, I would just say that “These results were consistent when using the multiple-threshold approach to calculate multifunctionality (Supplementary Fig. 4 and Supplementary Table 3)”.

Response: Thanks. As suggested, we have now simplified this text as follows (Lines 159-161):

“Similarly, strong and negative effects of aridity on soil multifunctionality were observed when using the multiple-threshold approach (Supplementary Fig. 4 and Supplementary Table 3).”

L210-211 rewrite to “As a complement to the analyses of bivariate correlations (Fig. 4; Supplementary Table 4), we used structural equation models (SEMs) to...”

Response: Thank you for the suggestion. The sentence has been amended accordingly (Lines 210-211).

L278-279 Your data does not provide any evidence of aridity increasing competition between plants. Indeed, several studies on this topic shows rather the opposite.

Response: We thank you for highlighting this issue and agree that our data does not provide such evidence of aridity increasing competition between plants, which is not the main topic of this manuscript. To avoid the speculative and controversial phrase, we have removed the words “and increasing competition”. The sentence now reads (Lines 286-287):

“Aridity thus acts as an environmental filter, selecting plant species with similar niches⁴⁹. Therefore,”

⁴⁹Le Bagousse-Pinguet, Y. et al. Testing the environmental filtering concept in global drylands. *J. Ecol.* **105**, 1058–1069 (2017).

Mean annual precipitation of the study site from which the microcosm soils comes

from is different in L468 (7.9 °C) and in L473 (18.5 °C), clarify this, please.

Response: We thank you for noticing this. We referred to 7.9 °C as the mean annual temperature of the sampling site (please see also Line 476), but to 18.5 °C as the mean annual soil surface temperature (please see also Line 481). Thus, we incubated soil microcosms at 18.5 °C to simulate the *in situ* soil conditions of the sampling site (please see also Line 481).

L475 Field capacity, by definition, is that maximum amount of water that a given soil can hold. Therefore, it seems impossible to me to maintain a soil in > 100% of its field capacity for over a month.

Response: We thank you for highlighting this vague expression and fully agree on the definition of field capacity. Actually, the ten moisture levels in our microcosm experiment were designed based on different percentages of field capacity and aimed to match with differences in moisture conditions among the field sites (please see also Lines 131-134, Lines 484-487 and Supplementary Fig. 1). Therefore, our intention is to say that the moisture contents of our soil microcosms were adjusted and artificially maintained at the ten moisture levels respectively equivalent to 3–120% field capacity during the duration of the experiment for a month. Because the soil used for the microcosm experiment has a relatively low field capacity (i.e., 27.6%; please see also Line 484), the moisture content measured at 120% field capacity at the end of experiment was $33.57\% \pm 1.94$, which falls within the scope of moisture conditions among the field sites (please see also Supplementary Fig. 1). In light of this, we believe that the manipulated moisture treatment at 120% of field holding capacity is reasonable and can reflect the realistic conditions of soil water availability in the field. To further clarify, we have revised the sentence, and it now reads (Lines 482-484):

“....., and moisture contents were adjusted and artificially maintained at the ten levels respectively equivalent to 3%, 5%, 8%, 10%, 20%, 40%, 60%, 80%, 100% and 120% field capacity (27.6%) during the duration of the experiment for 30 days.”

Fig. 5 Goodness-of-fit tests for the SEM of high aridity sites are not great. Chi2 is borderline, despite having only 1 degree of freedom, Bollen-Stine is non-significant, and RMSEA's value is well above 0.05. This is not too dramatic, considering that is one amongst the many tests performed, and that it is just to assess indirect vs direct effects, but should be acknowledged in the main text and revised to improve model fit

if possible.

Response: We thank you for highlighting this issue, of which we were not aware. As suggested, we have revised to improve the overall goodness of fit of the SEM of more arid regions for both soil multifunctionality (please see Figure 1 below) and simplified soil multifunctionality (please see Figure 2 below). For both the revised models, we released 1 additional degree of freedom by removing the relationship between soil pH and BNPP with a path coefficient close to zero. The goodness-of-fit statistics of these two revised models have now been improved greatly and reveal an acceptable fit of our *a priori* model (please see also Supplementary Fig. 17a). Also, these new results did not change the previous findings reported in the manuscript. Therefore, we present these two new models in this revised version (please see also the revised Fig. 5b; Supplementary Fig. 28b and Supplementary Tables 5,8). Furthermore, this information has also been highlighted in the main text (Lines 768-771) and the legend of the Fig. 5 (Lines 1134-1135) as follows:

Lines 768-771: “All of these goodness-of-fit metrics revealed an acceptable fit of our *a priori* model, with the exception of removing the relationship between soil pH and BNPP with a coefficient close to zero for the SEM of more arid regions to improve its model fit (Fig. 5; Supplementary Table 5).”

Lines 1134-1135: “For the SEM of sites with aridity > 0.8 , we remove the relationship between soil pH and BNPP with a coefficient close to zero to improve its overall goodness of fit.”

Figure 1 Structural equation model (SEM) accounting for the hypothesized direct and indirect relationships between aridity, soil properties (pH and clay content), biodiversity (plant species richness and the soil microbial diversity index), BNPP and soil multifunctionality. SEM is shown for sites with aridity > 0.8 ($N = 76$). Note that we only present significant relationships (two-sided $P < 0.05$) and their coefficients (numbers adjacent to arrows) for graphical simplicity. Latitude, longitude, and elevation of the field sites are included to account for the spatial structure of our dataset, and thus their coefficients are not included. *A priori* model including all hypothesized causal relationships is available in Supplementary Fig. 17a, and all the rest of coefficients and their significance levels are available in Supplementary Table 5. Note that we remove the relationship between soil pH and BNPP with a coefficient close to zero to improve its overall goodness of fit. Continuous and dashed arrows indicate positive and negative relationships, respectively. The thickness of the arrow is proportional to the magnitude of standardized path coefficients and indicative of the strength of the relationship. Asterisks indicate the significance level of each coefficient: $*P < 0.05$; $**P < 0.01$; $***P < 0.001$. R^2 is the proportion of variance explained by the model. Goodness-of-fit statistics for the SEM are given (d.o.f., degrees of freedom; RMSEA, root mean squared error of approximation).

Figure 2 SEM accounting for the hypothesized causal relationships between aridity, soil properties (pH and clay content), biodiversity (plant species richness and the soil microbial diversity index), BNPP and simplified soil multifunctionality. SEM is shown for sites with aridity > 0.8 ($N = 76$).

We only present significant relationships (two-sided $P < 0.05$) and their coefficients (numbers adjacent to arrows) for graphical simplicity. Latitude, longitude, and elevation of the field sites are included to account for the spatial structure of our dataset, and thus their coefficients are not included. An *a priori* model including all hypothesized causal relationships is available in Supplementary Fig. 17a, and all the rest of coefficients and their significance levels are available in Supplementary Table 8. Note that we remove the relationship between soil pH and BNPP with a coefficient close to zero to improve its overall goodness of fit. Continuous and dashed arrows indicate positive and negative relationships, respectively. The thickness of the arrow is proportional to the magnitude of standardized path coefficients and indicative of the strength of the relationship. Asterisks indicate the significant level of each coefficient: * $P < 0.05$; ** $P < 0.01$; *** $P < 0.001$. R^2 is the proportion of variance explained by the model. Goodness-of-fit statistics for the SEM are given (d.o.f., degrees of freedom; RMSEA, root mean squared error of approximation).

Fig 6 (and related main text in L232-233), I do not understand what the authors mean by “shrinking areas” in this context, are those areas currently with AI > 0.8 that will have a lower AI in the future? If so, why is that these are included within the first category in the maps (green + grey)? Clarification needed here.

Response: We thank you for noticing this point. Yes, you are exactly right. We referred to “shrinking areas” as those areas currently (1970-2000) with an aridity level > 0.8 that will have a lower aridity level (< 0.8) predicted for 2100. We used the Fig. 6 to show both the predicted expanding and shrinking areas with an aridity level crossing 0.8 relative to 1970-2000 for 2100 in the main text. Based on these maps, we can quantitatively estimate the percentage of expanding areas between 1970-2000 and 2100 as:

Percentage of expanding areas (%) = (Expanding areas between 1970-2000 and 2100 – Shrinking areas between 1970-2000 and 2100)/(Baseline areas in 1970-2000)

where the baseline areas in 1970-2000 should include both the unchanged (i.e., the grey shading) and shrinking (i.e., the green shading) areas between 1970-2000 and 2100.

However, we recognize that the previous legends of the Fig. 6 are a bit vague and not readily understood. To avoid confusion, we have revised the legends of the Fig. 6 to make it more accurate and clearer (please see also Figure 3 below).

Figure 3 Predicted future changes in areas crossing 0.8 aridity level in drylands across northern China. a,b Predictions of future changes in areas that will cross 0.8 aridity level are shown for between 1970–2000 and 2100 by the IPCC’s RCP 4.5 (i.e., assuming saturated increase in CO₂ emissions; a) and 8.5 (i.e., assuming sustained increase in CO₂ emissions; b) scenarios in drylands across northern China, respectively. The blank areas are outside of the range considered for this study (i.e., areas that are dry-subhumid regions, semiarid regions and non-drylands today).

Replies to Reviewer 3

Reviewer #3 (Remarks to the Author):

I reviewed the previous version of this manuscript. The authors addressed all of my concerns in this revised version. Indeed, additional analyses by excluding total N and total P have strengthened the major argument of the manuscript. Changing "effects of diversity on ecosystem functions" to "relationships between diversity and ecosystem functions" across the manuscript has better reflected what the dataset and the experimental designs can substantiate. I feel that this manuscript is now acceptable for publication. Congrats to the authors for this very nice piece of work.

Response: We thank the reviewer for his positive feedback on our revisions and for his great suggestions/comments to improve the manuscript.

P.S.

Ln. 34: I would feel more comfortable to replace “expected” with “predicted”

Response: Thank you for the suggestion. The word has been replaced accordingly (Line 33).

Ln. 347: considered “as” indicators...

Response: Thanks. The word has now been added accordingly (Line 355).

=====

We hope that you find our revision satisfactory. Thank you very much!